# Exogenous Isomorphism for Counterfactual Identifiability

**Yikang Chen** [1]   **Dehui Du** [1]

## Abstract

This paper investigates $\sim_{\mathcal{L}_3}$-identifiability, a form of complete counterfactual identifiability within the Pearl Causal Hierarchy (PCH) framework, ensuring that all Structural Causal Models (SCMs) satisfying the given assumptions provide consistent answers to all causal questions. To simplify this problem, we introduce exogenous isomorphism and propose $\sim_{\mathrm{EI}}$-identifiability, reflecting the strength of model identifiability required for $\sim_{\mathcal{L}_3}$-identifiability. We explore sufficient assumptions for achieving $\sim_{\mathrm{EI}}$-identifiability in two special classes of SCMs: Bijective SCMs (BSCMs), based on counterfactual transport, and Triangular Monotonic SCMs (TM-SCMs), which extend $\sim_{\mathcal{L}_2}$-identifiability. Our results unify and generalize existing theories, providing theoretical guarantees for practical applications. Finally, we leverage neural TM-SCMs to address the consistency problem in counterfactual reasoning, with experiments validating both the effectiveness of our method and the correctness of the theory.

## 1. Introduction

The purpose of counterfactual reasoning is to answer questions about hypothetical, unobserved worlds. It has been applied in tasks such as fairness evaluation (Kusner et al., 2017), explanation generation (Karimi et al., 2020), harm quantification (Richens et al., 2022), and policy optimization (Tsirtsis & Rodriguez, 2023). Counterfactual identification is a subtask of counterfactual reasoning, aiming to determine whether all models satisfying certain reasonable assumptions yield the same answer to a specific counterfactual question. It is critical to ensure the dependability of counterfactual reasoning results, as the reasoning outcomes can only be consistent with the assumptions — and thus confirm the reliability of the constructed model — if counterfactual identifiability is guaranteed.

(Pearl & Mackenzie, 2018) categorized causal questions into three levels of human cognition, termed the Pearl Causal Hierarchy (PCH) (Bareinboim et al., 2022). Counterfactual reasoning corresponds to human imagination, situated at the highest level $\mathcal{L}_3$ of the PCH, encoding the most intricate and nuanced information. Structural Causal Models (SCMs) provide specific semantics for addressing counterfactual questions. Counterfactual identifiability on SCMs guarantees that all SCMs adhering to the assumptions yield consistent results for $\mathcal{L}_3$ quantities. Prior research has constrained causal structures to identify specific counterfactual effects (Shpitser & Pearl, 2008; Correa et al., 2021; Xia et al., 2023), or limited causal mechanisms to determine counterfactual outcomes (Lu et al., 2020; Nasr-Esfahany et al., 2023; Scetbon et al., 2024). Further studies investigate model identifiability for counterfactuals, offering empirical evidence that counterfactual outcomes are identifiable (Khemakhem et al., 2021; Javaloy et al., 2023).

This work introduces a novel identification target: identification across the entire counterfactual layer of the PCH. This requires that all SCMs satisfying the assumptions yield consistent results for any counterfactual statement. Within the PCH, since the counterfactual layer $\mathcal{L}_3$ encodes all causal information, the identifiability of the SCMs in $\mathcal{L}_3$ implies that these SCMs provide consistent answers to all causal questions, rendering them indistinguishable for any causal statements. Thus, identifiability over the counterfactual layer represents the most stringent and comprehensive goal for causal quantity identifiability within the PCH.

To achieve this goal, we first examine the identifiability problem in causal inference from the perspective of model identifiability in Section 2. In this context, identifiability over the counterfactual layer is denoted as $\sim_{\mathcal{L}_3}$-identifiability. To simplify the $\sim_{\mathcal{L}_3}$-identifiability problem, we establish an alternative form of identifiability in Section 3, denoted as $\sim_{\mathrm{EI}}$-identifiability, which is induced by an equivalence relation termed exogenous isomorphism. This form of identifiability demonstrates that fully recovering the exogenous variables of SCMs is unnecessary to ensure consistent results for any counterfactual quantities, clarifying the strength of assumptions required for counterfactual layer identification.

---

[1]Shanghai Key Laboratory of Trustworthy Computing, East China Normal University. Correspondence to: Dehui Du <dhdu@sei.ecnu.edu.cn>.

*Proceedings of the 42$^{nd}$ International Conference on Machine Learning*, Vancouver, Canada. PMLR 267, 2025. Copyright 2025 by the author(s).

We then explore sets of assumptions that induce $\sim_{\text{EI}}$-identifiability, focusing on two special classes of SCMs. The first class, termed Bijective SCMs (BSCMs), is studied in Section 4, where we induce exogenous isomorphism between BSCMs from the perspective of counterfactual transport (Theorem 4.6), offering a novel interpretation for counterfactual identifiability. The second class, termed Triangular Monotonic SCMs (TM-SCMs), is examined in Section 5. We identify a simple method to induce exogenous isomorphism between TM-SCMs (Corollary 5.4), which can be viewed as a strengthened version of $\sim_{\mathcal{L}_2}$-identifiability for Markovian Causal Bayesian Networks. This corollary unifies and generalizes prior theories for proving counterfactual outcome identifiability, indirectly demonstrating that several models used in previous works for counterfactual outcome identifiability are theoretically $\sim_{\mathcal{L}_3}$-identifiable. This provides theoretical guarantees for their safe application in practice. Finally, leveraging this theory, we utilize TM-SCMs parameterized by neural networks in Section 6 to address counterfactual identification and estimation problems. Experimental results in Section 8 on these models empirically support the validity of our theory.

## 2. Preliminaries

In this section, we provide the background for understanding the problem and describe the problem setting. We begin by reviewing the relevant concepts of SCMs. In Section 2.1, we present the definition of SCM and introduce key notions such as do-intervention and causal order. Subsequently, in Section 2.2, we leverage these concepts to logically characterize $\mathcal{L}_1, \mathcal{L}_2, \mathcal{L}_3$ in PCH and define $\mathcal{L}_3$-consistency. Next, in Section 2.3, we introduce counterfactual identification problem. Finally, in Section 2.4, we elaborate on model identifiability, establish the connection between causal identifiability and model identifiability, and motivate the research objective of this work: $\sim_{\mathcal{L}_3}$-identifiability.

**Notation**  We study the problem from a more general measure-theoretic perspective. Let the probability space be $(\Omega, \mathcal{F}, P)$, where the set $\Omega$ is the sample space, the $\sigma$-algebra $\mathcal{F}$ on $\Omega$ is the event space, and $P$ is the probability measure. We assume all measurable spaces are standard Borel. We denote by uppercase $X$ a random variable taking values from the measurable space $(\Omega_X, \mathcal{F}_X)$, which is a measurable function from $\Omega$ to $\Omega_X$, where $\Omega_X$ is also called the domain of $X$. For a given finite index set $\mathcal{I}$, we use uppercase bold $\mathbf{X}$ to denote a collection of random variables $(X_i)_{i \in \mathcal{I}}$ with the associated space $(\Omega_{\mathbf{X}}, \mathcal{F}_{\mathbf{X}})$, where $\Omega_{\mathbf{X}} = \prod_{i \in \mathcal{I}} \Omega_{X_i}$ and $\mathcal{F}_{\mathbf{X}} = \bigotimes_{i \in \mathcal{I}} \mathcal{F}_{X_i}$ are the product space and the product $\sigma$-algebra, respectively. For a subset of indices $I \subseteq \mathcal{I}$, $(X_i)_{i \in I}$ is abbreviated as $\mathbf{X}_I$ or renamed as $\mathbf{Y}$ with $\mathbf{Y} \subseteq \mathbf{X}$, where the corresponding index set is denoted by $I_{\mathbf{Y}}$. The distribution of $\mathbf{X}$ is represented by

$P_{\mathbf{X}}$, defined as the pushforward of $P$ under $\mathbf{X}$, such that $P_{\mathbf{X}} = \mathbf{X}_{\sharp} P$. A set $\mathcal{X} \in \mathcal{F}_{\mathbf{X}}$ represents a random event, with the associated probability term $\mathbb{P}(\mathbf{X} \in \mathcal{X}) = P_{\mathbf{X}}(\mathcal{X})$.

### 2.1. Structural Causal Model

**Definition 2.1** (Structural Causal Model (SCM)). An SCM is a tuple $\mathcal{M} = \langle \mathcal{I}, \Omega_{\mathbf{V}}, \Omega_{\mathbf{U}}, \boldsymbol{f}, P_{\mathbf{U}} \rangle$, where $\mathcal{I}$ is a finite index set. The product space $\Omega_{\mathbf{V}} = \prod_{i \in \mathcal{I}} \Omega_{V_i}$ and $\Omega_{\mathbf{U}} = \prod_{i \in \mathcal{I}} \Omega_{U_i}$ denote the domain of endogenous and exogenous variables, respectively. The measurable function $\boldsymbol{f} : \Omega_{\mathbf{V}} \times \Omega_{\mathbf{U}} \to \Omega_{\mathbf{V}} = (f_i)_{i \in \mathcal{I}}$ specifies the causal mechanisms, where each $f_i : \Omega_{\mathbf{V}_{\text{pa}(i)}} \times \Omega_{U_i} \to \Omega_{V_i}$ is associated with a set of indices $\text{pa}(i) \subseteq \mathcal{I}$. The probability measure $P_{\mathbf{U}}$ is called the exogenous distribution. If $P_{\mathbf{U}} = \prod_{i \in \mathcal{I}} P_{U_i}$ is a product measure, the SCM is said to be Markovian.

In this work, we primarily focus on recursive SCMs, which are defined such that a partial order $\preceq$ exists on the index set $\mathcal{I}$, and for any pair of indices $i, j \in \mathcal{I}$, if $j \preceq i$, then $i \notin \text{pa}(j)$. If the partial order $\preceq$ admits a linear extension $\leq$, we call it the causal order of the SCM, which can be constructed using a topological sorting algorithm and is not necessarily unique. Recursiveness ensure that the solution of the SCM exists and is unique. According to (Bongers et al., 2016), if an SCM has a unique solution, then there exists a function $\boldsymbol{\Gamma}$ such that for almost all $\mathbf{u} \in \Omega_{\mathbf{U}}$ and $\mathbf{v} \in \Omega_{\mathbf{V}}$, $\mathbf{v} = \boldsymbol{\Gamma}(\mathbf{u})$ if and only if $\mathbf{v} = \boldsymbol{f}(\mathbf{v}, \mathbf{u})$. We refer to $\boldsymbol{\Gamma}$ as the solution mapping of the recursive SCM.

SCMs also allow for operations called do-interventions. For a subset of indices $I_{\mathbf{X}} \subseteq \mathcal{I}$, performing an intervention to set $\mathbf{X} = \mathbf{V}_{I_{\mathbf{X}}}$ to a specific value $\mathbf{x}$ corresponds to deriving a new SCM $\mathcal{M}_{[\mathbf{x}]} = \langle \mathcal{I}, \Omega_{\mathbf{V}}, \Omega_{\mathbf{U}}, \boldsymbol{f}_{[\mathbf{x}]}, P_{\mathbf{U}} \rangle$, called a submodel. In $\mathcal{M}_{[\mathbf{x}]}$, $\boldsymbol{f}_{[\mathbf{x}]} = (f_{i[\mathbf{x}]})_{i \in \mathcal{I}}$ is defined such that

$$f_{i[\mathbf{x}]} = \begin{cases} x_i & i \in I_{\mathbf{X}}, \\ f_i & i \in \mathcal{I} \setminus I_{\mathbf{X}}. \end{cases}$$

Endogenous variables in $\mathcal{M}_{[\mathbf{x}]}$ will be denoted as $\mathbf{V}_{[\mathbf{x}]}$.

Another property of recursive SCMs is that any submodel is also a recursive SCM. This implies that, in the submodel, given a value $\mathbf{u}$, the endogenous variables $\mathbf{V}_{[\mathbf{x}]}$ are determined as $\boldsymbol{\Gamma}_{[\mathbf{x}]}(\mathbf{u})$. Under this premise, the value of the endogenous variables $\mathbf{Y}_{[\mathbf{x}]}$ under a given $\mathbf{u}$ is represented as $\mathbf{Y}_{\mathcal{M}_{[\mathbf{x}]}}(\mathbf{u}) = (\boldsymbol{\Gamma}_{[\mathbf{x}]}(\mathbf{u}))_{I_{\mathbf{Y}}}$, called the potential response.

### 2.2. Pearl Causal Hierarchy

(Pearl & Mackenzie, 2018) categorized causal questions into three levels of human cognition: seeing, doing, and imagining. To formalize these questions and delineate the logical differences between the three levels, symbolic languages $\mathcal{L}_1, \mathcal{L}_2, \mathcal{L}_3$ are introduced. Each language consists of $\mathcal{L}_i$-statements $\varphi$, where every $\varphi \in \mathcal{L}_i$ is a Boolean combination

of inequalities between polynomials over probability terms $\mathbb{P}(\alpha_i)$. For $\mathcal{L}_1$, $\mathbb{P}(\alpha_1)$ takes the form $\mathbb{P}(\mathbf{Y} \in \boldsymbol{\mathcal{Y}})$, representing the probability of $\boldsymbol{\mathcal{Y}}$ occurring, making $\mathcal{L}_1$ a standard probabilistic logic. $\mathcal{L}_2$ introduces interventions, with $\mathbb{P}(\alpha_2)$ in the form $\mathbb{P}(\mathbf{Y}_{[\mathbf{x}]} \in \boldsymbol{\mathcal{Y}})$, denoting the probability of $\boldsymbol{\mathcal{Y}}$ occurring when $\mathbf{X}$ is intervened to x. $\mathcal{L}_3$ further encodes conjunctions under different interventions, making $\mathbb{P}(\alpha_3)$ take the form $\mathbb{P}(\mathbf{Y}_* \in \boldsymbol{\mathcal{Y}}_*)$, where $\mathbf{Y}_* = (\mathbf{Y}_{j[\mathbf{x}_j]})_{j \in 1:n}$ represents the collection of endogenous variables $\mathbf{Y}_{j[\mathbf{x}_j]}$ from $n$ submodels $\mathcal{M}_{[\mathbf{x}_1]}, \mathcal{M}_{[\mathbf{x}_2]}, \ldots, \mathcal{M}_{[\mathbf{x}_n]}$. Thus, $\mathbf{Y}_* \in \boldsymbol{\mathcal{Y}}_*$ is logically equivalent to the conjunction $\bigwedge_{j \in 1:n} \mathbf{Y}_{j[\mathbf{x}_j]} \in \boldsymbol{\mathcal{Y}}_j$.

An SCM $\mathcal{M}$ provides semantics for the symbolic languages $\mathcal{L}_1, \mathcal{L}_2, \mathcal{L}_3$. For $\mathcal{L}_3$, any term of the form $\mathbb{P}(\mathbf{Y}_* \in \boldsymbol{\mathcal{Y}}_*)$ can be evaluated by $\mathcal{M}$ as $P^{\mathcal{M}}_{\mathbf{Y}_*}(\boldsymbol{\mathcal{Y}}_*)$, where $P^{\mathcal{M}}_{\mathbf{Y}_*}$ is called the counterfactual distribution. Let $\mathbf{Y}_{\mathcal{M}_*} = (\mathbf{Y}_{\mathcal{M}_{j[\mathbf{x}_j]}})_{j \in 1:n}$. Since each $\mathbf{Y}_{j[\mathbf{x}_j]} = \mathbf{Y}_{\mathcal{M}_{j[\mathbf{x}_j]}} \circ \mathbf{U}$ almost surely, the $P^{\mathcal{M}}_{\mathbf{Y}_*}$ is the pushforward measure $(\mathbf{Y}_{\mathcal{M}_*})_\sharp P_{\mathbf{U}}$, yielding:

$$P^{\mathcal{M}}_{\mathbf{Y}_*}(\boldsymbol{\mathcal{Y}}_*) = P_{\mathbf{U}}\left( (\mathbf{Y}_{\mathcal{M}_*})^{-1}[\boldsymbol{\mathcal{Y}}_*] \right). \quad (1)$$

Equation 1 is called $\mathcal{L}_3$-valuation. After the $\mathcal{L}_3$-valuation, a statement $\varphi \in \mathcal{L}_3$ evaluated by $\mathcal{M}$, denoted as $\varphi(\mathcal{M})$, can be checked for validity. If $\varphi(\mathcal{M})$ holds, we write $\mathcal{M} \models \varphi$.

The Pearl Causal Hierarchy (PCH) is defined as the combination of the above syntax and semantics. Specifically, when an SCM $\mathcal{M}$ is given, the collection of observational, interventional, and counterfactual distributions defined syntactically by $\mathcal{L}_1, \mathcal{L}_2, \mathcal{L}_3$ and semantically by $\mathcal{L}_1, \mathcal{L}_2, \mathcal{L}_3$-valuations constitutes the PCH. In this sense, the PCH encapsulates all causal information entailed by an SCM.

Furthermore, the PCH exhibits a strict hierarchy. In terms of syntax, the representations of the terms $\mathbb{P}(\alpha_i)$ imply that $\mathcal{L}_1 \subsetneq \mathcal{L}_2 \subsetneq \mathcal{L}_3$. In terms of logical expressiveness, similar theorems have been established, known as the Causal Hierarchy Theorem (CHT) (Bareinboim et al., 2022).

To illustrate the expressiveness of $\mathcal{L}_i$, consistency is introduced. Let $\mathcal{L}_i(\mathcal{M}) = \{\varphi(\mathcal{M}) \mid \varphi \in \mathcal{L}_i\}$ denote the $\mathcal{L}_i$-theory of $\mathcal{M}$, then $\mathcal{L}_i$-consistency is defined as follows:

**Definition 2.2** ($\mathcal{L}_i$-Consistency). SCMs $\mathcal{M}^{(1)}$ and $\mathcal{M}^{(2)}$ are $\mathcal{L}_i$-consistent, denoted $\mathcal{M}^{(1)} \sim_{\mathcal{L}_i} \mathcal{M}^{(2)}$, if their $\mathcal{L}_i$-theories are identical, i.e., $\mathcal{L}_i(\mathcal{M}^{(1)}) = \mathcal{L}_i(\mathcal{M}^{(2)})$.

The CHT demonstrates that in the PCH, $\sim_{\mathcal{L}_i}$ generally does not imply $\sim_{\mathcal{L}_j}$ for $i < j$, formally indicating that higher levels possess greater expressiveness than lower levels.

### 2.3. Counterfactual Identification

In practice, the underlying true SCM $\mathcal{M}^*$ is almost never fully specified, making direct reasoning on $\mathcal{M}^*$ impractical. Consequently, researchers rely on partial knowledge about $\mathcal{M}^*$ to address causal questions, which necessitates

constructing a proxy $\mathcal{M}'$ that satisfies this partial knowledge and performing equivalent reasoning on $\mathcal{M}'$. Causal identification is the task of proving or reasoning whether any proxy constructed from partial knowledge provides consistent answers to specific causal questions.

For example, when the true SCM $\mathcal{M}^*$ is assumed to be Markovian and both observational distribution and the causal graph are available, constructing a Markovian Causal Bayesian Network (CBN) suffices to answer any causal question in $\mathcal{L}_2$ via truncated factorization (Pearl, 2009). Thus, Markovian CBNs can be regarded as identifiable at the intervention level. When the Markovian assumption fails, semi-Markovian CBNs can still answer a subset of causal questions in $\mathcal{L}_2$ using do-calculus.

Regarding counterfactuals, as they represent the highest level in the PCH, achieving counterfactual identifiability requires stricter conditions and more complex methodologies. For instance, even Markovian CBNs cannot ensure consistency when answering certain $\mathcal{L}_3$ questions (Nasr-Esfahany & Kiciman, 2023). Prior research on counterfactual identification often focuses on identifying a subset of counterfactual statements $\boldsymbol{\varphi} \subseteq \mathcal{L}_3$. These efforts can be categorized into constraining causal structures to identify counterfactual effects, such as (Shpitser & Pearl, 2008), or constraining causal mechanisms to identify counterfactual outcomes, such as (Nasr-Esfahany et al., 2023).

### 2.4. Model Identifiability

SCMs are a special class of generative models, where the exogenous distribution and causal mechanisms can be interpreted as the latent distribution and generative function of the model, respectively. Identifiability has been a continuously discussed topic in the literature, appearing in representation learning (Bengio et al., 2013), causal discovery (Glymour et al., 2019), causal representation learning (Schölkopf et al., 2021), and causal quantity inference (Pearl, 2009). Providing a broad definition of identifiability may lead to ambiguity. Borrowing from the definition of identifiability in (Khemakhem et al., 2020), these notions of identifiability can be unified through equivalence classes:

**Definition 2.3** ($\sim$-Identifiability). Let $\mathcal{A}$ denote a set of assumptions, $[\mathcal{A}] = \{\mathcal{M} \mid \mathcal{M} \models \mathcal{A}\}$ includes all models satisfying the assumptions, and $\sim$ denote an equivalence relation between models. Then a model is said to be $\sim$-identifiable from $\mathcal{A}$, or identifiable up to $\sim$-equivalence class, if for any $\mathcal{M}^{(1)}, \mathcal{M}^{(2)} \in [\mathcal{A}]$, $\mathcal{M}^{(1)} \sim \mathcal{M}^{(2)}$.

We now interpret the problem of causal identifiability from the perspective of $\sim$-identifiability. Suppose the causal question involves a set of $\mathcal{L}_i$-statements $\boldsymbol{\varphi} \subseteq \mathcal{L}_i$, and two SCMs $\mathcal{M}^{(1)}$ and $\mathcal{M}^{(2)}$ are said to be $\boldsymbol{\varphi}$-consistent if $\boldsymbol{\varphi}(\mathcal{M}^{(1)}) = \boldsymbol{\varphi}(\mathcal{M}^{(2)})$. Since $\boldsymbol{\varphi}$-consistency is an equiva-

lence relation between SCMs, denoted as $\mathcal{M}^{(1)} \sim_{\boldsymbol{\varphi}} \mathcal{M}^{(2)}$, the problem of causal identification with respect to $\boldsymbol{\varphi}$ can be reframed as a problem of $\sim_{\boldsymbol{\varphi}}$-identifiability.

The CHT states that when $\mathcal{A}$ contains only low-level knowledge, $\sim_{\boldsymbol{\varphi}}$-identifiability is often unattainable. Achieving this requires assuming higher-level information. For instance, under the assumptions of known $\mathcal{L}_1$ observational distribution, causal graph, and Markovianity, the causal graph encodes structural constraints on $\mathcal{L}_2$, enabling the constructed CBN to answer all questions in $\mathcal{L}_2$. Since the union of all statements $\boldsymbol{\varphi} \subseteq \mathcal{L}_2$ equals $\mathcal{L}_2$, this property is also referred to as $\sim_{\mathcal{L}_2}$-identifiability.

This paper focuses on $\sim_{\mathcal{L}_3}$-identifiability, which is an enhanced version of $\sim_{\mathcal{L}_2}$-identifiability, requiring $\sim_{\boldsymbol{\varphi}}$-identifiability for any $\boldsymbol{\varphi} \subseteq \mathcal{L}_3$. If $\mathcal{A}$ satisfies $\sim_{\mathcal{L}_3}$-identifiability, then by the definition of $\mathcal{L}_3$-consistency, any $\mathcal{M}^{(1)}, \mathcal{M}^{(2)} \in [\mathcal{A}]$ satisfy $\mathcal{M}^{(1)} \sim_{\mathcal{L}_3} \mathcal{M}^{(2)}$. Since $\mathcal{L}_3$ represents the highest level in the PCH and encodes all causal information of the SCM, if $\mathcal{M}^{(1)} \sim_{\mathcal{L}_3} \mathcal{M}^{(2)}$, the two models are indistinguishable under any causal statement. Therefore, $\sim_{\mathcal{L}_3}$-identifiability is the ultimate goal for causal identifiability within the PCH.

## 3. Exogenous Isomorphism

$\sim_{\mathcal{L}_3}$-identifiability, by definition, is not straightforward to handle, as it is indirectly defined through the PCH rather than directly based on the model itself. Therefore, we aim to find a simpler model-based identifiability notion that implies $\sim_{\mathcal{L}_3}$-identifiability, thereby simplifying the problem.

One possible choice is $=$-identifiability, which requires uniquely identifying the generative model. Some prior works have achieved $=$-identifiability (Xi & Bloem-Reddy, 2023), which indirectly induces $\sim_{\mathcal{L}_3}$-identifiability. This serves as one approach to addressing $\sim_{\mathcal{L}_3}$-identifiability, but the required assumptions are often too strong. An alternative property is defined via counterfactual equivalence (Peters et al., 2017, Proposition 6.49), which—while not insisting on uniqueness of the causal mechanisms—does demand full recovery of the exogenous distribution.

Between $=$-identifiability, counterfactual equivalence and $\sim_{\mathcal{L}_3}$-identifiability, there should exist other forms of $\sim$-identifiability, as counterfactual reasoning does not require a completely fixed latent representation. This is akin to humans being able to answer "what-if" questions without fully understanding the underlying factors of the physical world. Identifying such a form of $\sim$-identifiability is a goal worth exploring. Motivated by this, we have identified an equivalence relation between SCMs, denoted as $\sim_{\text{EI}}$ and referred to as exogenous isomorphism, for which $\sim_{\text{EI}}$-identifiability strictly implies $\sim_{\mathcal{L}_3}$-identifiability, yet it is weaker than the previous two. This characterization more precisely reflects

the strength of model identifiability required to achieve complete counterfactual identifiability.

Let the recursive SCMs $\mathcal{M}^{(k)} = \langle \mathcal{I}, \Omega_{\mathbf{V}}, \Omega_{\mathbf{U}}^{(k)}, \boldsymbol{f}^{(k)}, P_{\mathbf{U}}^{(k)} \rangle$ share the index set $\mathcal{I}$ and the domain of endogenous variables $\Omega_{\mathbf{V}}$. For a given endogenous value $\mathbf{v} \in \Omega_{\mathbf{V}}$, we abbreviate $f_i^{(k)}(\mathbf{v}_{\text{pa}^{(k)}(i)}, \cdot)$ as $f_i^{(k)}(\mathbf{v}, \cdot)$. Exogenous isomorphism is then defined as:

**Definition 3.1** (Exogenous Isomorphism). Recursive SCMs $\mathcal{M}^{(1)}$ and $\mathcal{M}^{(2)}$ are said to be exogenously isomorphic, denoted $\mathcal{M}^{(1)} \sim_{\text{EI}} \mathcal{M}^{(2)}$, if there exists a shared causal ordering $\leq$ and function $\mathbf{h} : \Omega_{\mathbf{U}}^{(1)} \to \Omega_{\mathbf{U}}^{(2)}$ satisfying:

- Component-wise Bijection: For each $i \in \mathcal{I}$, $\mathbf{h} = (h_i)_{i \in \mathcal{I}}$, where $h_i : \Omega_{U_i}^{(1)} \to \Omega_{U_i}^{(2)}$ is a bijection;
- Exogenous Distribution Isomorphism: $P_{\mathbf{U}}^{(2)} = \mathbf{h}_\sharp P_{\mathbf{U}}^{(1)}$;
- Causal Mechanism Isomorphism: For each $i \in \mathcal{I}$, for almost every $u_i^{(1)} \in \Omega_{U_i}^{(1)}$ and all $\mathbf{v} \in \Omega_{\mathbf{V}}$,

$$f_i^{(2)}(\mathbf{v}, h_i(u_i^{(1)})) = f_i^{(1)}(\mathbf{v}, u_i^{(1)}).$$

The implication of $\sim_{\text{EI}}$-identifiability for $\sim_{\mathcal{L}_3}$-identifiability is formally stated in the following theorem:

**Theorem 3.2** ($\sim_{\text{EI}}$ Implies $\sim_{\mathcal{L}_3}$). *For recursive SCMs $\mathcal{M}^{(1)}$ and $\mathcal{M}^{(2)}$, if $\mathcal{M}^{(1)} \sim_{\text{EI}} \mathcal{M}^{(2)}$, then $\mathcal{M}^{(1)} \sim_{\mathcal{L}_3} \mathcal{M}^{(2)}$.*

**Sketch of proof** The mechanism isomorphism of each $f_i$ can be progressively deduced into the potential responses, ensuring $\mathbf{V}_{\mathcal{M}_*}^{(1)} = \mathbf{V}_{\mathcal{M}_*}^{(2)} \circ \mathbf{h}$. Given $P_{\mathbf{U}}^{(2)} = \mathbf{h}_\sharp P_{\mathbf{U}}^{(1)}$, we have $(\mathbf{V}_{\mathcal{M}_*}^{(2)} \circ \mathbf{h})_\sharp P_{\mathbf{U}}^{(1)} = (\mathbf{V}_{\mathcal{M}_*}^{(2)})_\sharp (\mathbf{h}_\sharp P_{\mathbf{U}}^{(1)})$, i.e., $(\mathbf{V}_{\mathcal{M}_*}^{(1)})_\sharp P_{\mathbf{U}}^{(1)} = (\mathbf{V}_{\mathcal{M}_*}^{(2)})_\sharp P_{\mathbf{U}}^{(2)}$. Thus, all $\mathcal{L}_3$-evaluations for $\mathcal{M}^{(1)}$ and $\mathcal{M}^{(2)}$ are identical, and any statement $\varphi \in \mathcal{L}_3$ yields the same result. Therefore, $\mathcal{M}^{(1)} \sim_{\mathcal{L}_3} \mathcal{M}^{(2)}$. For a detailed proof, see Appendix A.2.

## 4. Bijective SCM

Once identifiability is defined, the next step is to explore the assumption sets that induce identifiability. In this work, for simplicity, we target some special settings of recursive SCMs. Throughout, the set of positive integers $\{i \mid l \leq i \leq r, i \in \mathbb{N}^+\}$ is abbreviated as $l:r$ for $l, r \in \mathbb{N}^+$. Moreover, we assume that for each $i \in \mathcal{I}$, the domain of exogenous variables $\Omega_{U_i}$ and the domain of endogenous variables $\Omega_{V_i}$ are both indexed $\mathbb{R}^{d_i}$ with $1:d_i$ as the index set. That is, both exogenous and endogenous variables are random vectors, and the nested index set $\boldsymbol{\mathcal{I}} = \{(i,j) \mid i \in \mathcal{I}, j \in 1:d_i\}$ indexes the individual dimensions of $\Omega_{\mathbf{U}}$ and $\Omega_{\mathbf{V}}$.

Since the cardinalities of the exogenous and endogenous domains are equal, a bijection can be established between these domains. A bijection implies no information loss. If the causal mechanism is also bijective, then for any observation in the endogenous domain, a distinguishable latent

encoding can be identified in the exogenous domain. Below, we formally define this specific setting of SCMs:

**Definition 4.1** (Bijective SCM (BSCM)). A recursive SCM $\mathcal{M}$ is called a bijective SCM (BSCM) if its solution mapping $\mathbf{\Gamma}$ is a bijection.

**Proposition 4.2.** *A recursive SCM $\mathcal{M}$ is a BSCM if and only if $f_i(\mathbf{v}, \cdot)$ is a bijection for every $i \in \mathcal{I}$ and all $\mathbf{v} \in \Omega_{\mathbf{V}}$.*

For BSCMs $\mathcal{M}^{(1)}$ and $\mathcal{M}^{(2)}$, since their solution mappings $\mathbf{\Gamma}^{(k)} : \Omega_{\mathbf{U}}^{(k)} \to \Omega_{\mathbf{V}}$ are bijections, it is evident that the composition $(\mathbf{\Gamma}^{(2)})^{-1} \circ \mathbf{\Gamma}^{(1)} : \Omega_{\mathbf{U}}^{(1)} \to \Omega_{\mathbf{U}}^{(2)}$ is also a bijection. Given that exogenous isomorphism requires a bijection between $\Omega_{\mathbf{U}}^{(1)}$ and $\Omega_{\mathbf{U}}^{(2)}$, a natural question arises: can $(\mathbf{\Gamma}^{(2)})^{-1} \circ \mathbf{\Gamma}^{(1)}$ directly serve as an exogenous isomorphism? The following theorem provides a precise answer.

**Theorem 4.3** (BSCM-EI). *If two BSCMs $\mathcal{M}^{(1)}$ and $\mathcal{M}^{(2)}$ share a common causal order $\leq$ and the same observational distribution $P_{\mathbf{V}}$, then $\mathcal{M}^{(1)} \sim_{\mathrm{EI}} \mathcal{M}^{(2)}$ if and only if for every $i \in \mathcal{I}$, there exists a bijection $h_i : \Omega_{U_i}^{(1)} \to \Omega_{U_i}^{(2)}$ such that for all $\mathbf{v} \in \Omega_{\mathbf{V}}$,*

$$(f_i^{(2)}(\mathbf{v}, \cdot))^{-1} \circ (f_i^{(1)}(\mathbf{v}, \cdot)) = h_i$$

*almost surely.*[1]

### 4.1. $\sim_{\mathrm{EI}}$-Identification for BSCM

To derive assumption sets that imply $\sim_{\mathrm{EI}}$-identifiability, we need to restrict our perspective to a single SCM $\mathcal{M}$. Consider the conditions in Theorem 4.3, where there exists a function $h_i : \Omega_{U_i}^{(1)} \to \Omega_{U_i}^{(2)}$ such that for all $\mathbf{v} \in \Omega_{\mathbf{V}}$, $(f_i^{(2)}(\mathbf{v}, \cdot))^{-1} \circ (f_i^{(1)}(\mathbf{v}, \cdot)) = h_i$ holds almost surely. Now, consider different $\mathbf{v}, \mathbf{v}' \in \Omega_{\mathbf{V}}$. Since both $f_i^{(1)}(\mathbf{v}, \cdot)$ and $f_i^{(2)}(\mathbf{v}, \cdot)$ are bijections, by the associativity of composition,

$$(f_i^{(1)}(\mathbf{v}', \cdot)) \circ (f_i^{(1)}(\mathbf{v}, \cdot))^{-1} = (f_i^{(2)}(\mathbf{v}', \cdot)) \circ (f_i^{(2)}(\mathbf{v}, \cdot))^{-1}$$

almost surely, where both sides come from the SCM $\mathcal{M}$, and we explicitly name this concept as follows:

**Definition 4.4** (Counterfactual Transport). For a BSCM $\mathcal{M}$, the function $\mathbf{K}_{\mathcal{M}} : \Omega_{\mathbf{V}} \times \Omega_{\mathbf{V}} \times \Omega_{\mathbf{V}} \to \Omega_{\mathbf{V}}$ is called the counterfactual transport if $\mathbf{K}_{\mathcal{M}} = (K_{\mathcal{M},i})_{i \in \mathcal{I}}$ and for every $i \in \mathcal{I}$ and all $\mathbf{v}, \mathbf{v}' \in \Omega_{\mathbf{V}}$, the component $K_{\mathcal{M},i}(\cdot, \mathbf{v}, \mathbf{v}') = (f_i(\mathbf{v}', \cdot) \circ (f_i(\mathbf{v}, \cdot))^{-1}$.

Under Markovianity, the practical meaning of counterfactual transport is the transport between conditional distributions:

**Proposition 4.5.** *If the BSCM $\mathcal{M}$ is Markovian, then for almost all $\mathbf{v}, \mathbf{v}' \in \Omega_{\mathbf{V}}$, the conditional distributions satisfy $P_{V_i | \mathbf{V}_{pa(i)}}(\cdot, \mathbf{v}') = (K_{\mathcal{M},i}(\cdot, \mathbf{v}, \mathbf{v}'))_\sharp P_{V_i | \mathbf{V}_{pa(i)}}(\cdot, \mathbf{v}).$*[2]

Combining this with Theorem 3.2, we find that when all counterfactual transports are fixed, $\sim_{\mathrm{EI}}$-identifiability can be achieved. Let the following assumptions be defined: (i) $\mathcal{A}_{\mathrm{BSCM}}$: $\mathcal{M}$ is a BSCM; (ii) $\mathcal{A}_{\leq}$: the total order $\leq$ is a causal order for $\mathcal{M}$; (iii) $\mathcal{A}_{P_{\mathbf{V}}}$: the observational distribution of $\mathcal{M}$ is $P_{\mathbf{V}}$ with strictly positive density; (iv) $\mathcal{A}_{\mathbf{K}}$: there exists a function $\mathbf{K} : \Omega_{\mathbf{V}} \times \Omega_{\mathbf{V}} \times \Omega_{\mathbf{V}} \to \Omega_{\mathbf{V}} = (K_i)_{i \in \mathcal{I}}$ such that counterfactual transport $\mathbf{K}_{\mathcal{M}} = \mathbf{K}$ almost surely.

Let $\mathcal{A}_{\{\mathrm{BSCM}, \leq, P_{\mathbf{V}}, \mathbf{K}\}} = \{\mathcal{A}_{\mathrm{BSCM}}, \mathcal{A}_{\leq}, \mathcal{A}_{P_{\mathbf{V}}}, \mathcal{A}_{\mathbf{K}}\}$. Then:

**Theorem 4.6** (EI-ID from Counterfactual Transport). *An SCM is $\sim_{\mathrm{EI}}$-identifiable from $\mathcal{A}_{\{\mathrm{BSCM}, \leq, P_{\mathbf{V}}, \mathbf{K}\}}$.*

Verifying the assumption set $\mathcal{A}_{\mathbf{K}}$ in practice is challenging, making Theorem 4.6 difficult to apply. By combining Proposition 4.5 and considering Markovian BSCMs, a natural direction is to identify specific types of transport that are always well-defined between conditional distributions. One such transport, the KR transport, is constructed using one-dimensional conditional cumulative density functions in $1 : d$ order, ensuring that the mass at each point of the distribution follows a lexicographical order, and:

**Lemma 4.7** (Santambrogio 2015, Proposition 2.18). *Given any two distributions $P$ and $P'$ on $\mathbb{R}^d$ with strictly positive densities, the KR transport $\mathbf{T} : \mathbb{R}^d \to \mathbb{R}^d$ such that $P = \mathbf{T}_\sharp P'$ always exists and is almost surely unique.*

The KR transport induces a special case of $\mathcal{A}_{\{\mathrm{BSCM}, \leq, P_{\mathbf{V}}, \mathbf{K}\}}$ in Theorem 4.6. Let the following assumptions be defined: (i) $\mathcal{A}_{\mathrm{M}}$: $\mathcal{M}$ is Markovian; (ii) $\mathcal{A}_{\mathrm{KR}}$: for each $i \in \mathcal{I}$, the counterfactual transport component $K_{\mathcal{M},i}$ is almost surely equivalent to the KR transport.

Let $\mathcal{A}_{\{\mathrm{BSCM}, \leq, \mathrm{M}, P_{\mathbf{V}}, \mathrm{KR}\}} = \{\mathcal{A}_{\mathrm{BSCM}}, \mathcal{A}_{\leq}, \mathcal{A}_{\mathrm{M}}, \mathcal{A}_{P_{\mathbf{V}}}, \mathcal{A}_{\mathrm{KR}}\}$. Then:

**Theorem 4.8** (EI-ID from KR Transport). *An SCM is $\sim_{\mathrm{EI}}$-identifiable from $\mathcal{A}_{\{\mathrm{BSCM}, \leq, \mathrm{M}, P_{\mathbf{V}}, \mathrm{KR}\}}$.*

Nevertheless, in most practical scenarios, conditional cumulative density functions are unknown, and one often only has access to samples from the distributions. Therefore, KR transport cannot be constructed as defined, motivating the need for a tractable approximation of KR transport. This leads to a more practical subclass of BSCMs.

---

[1]The BGM proposed in (Nasr-Esfahany et al., 2023) corresponds to the $f_i$ of a BSCM. Furthermore, component-wise bijection and causal mechanism isomorphism in exogenous isomorphism are described as BGM equivalence. Therefore, Theorem 3.2 extends (Nasr-Esfahany et al., 2023, Proposition 6.2).

[2]A more general notion of counterfactual transport was defined in (Lara et al., 2024) from the perspective of coupling between conditional distributions. In this paper, the counterfactual transport in Markovian BSCMs (Definition 4.4 and Proposition 4.5) is a special instance of their more general framework.

# 5. Triangular Monotonic SCM

We call a function $\mathbf{T}(\mathbf{x}) = (T_j(\mathbf{x}_{1:j-1}, x_j))_{j \in 1:d}$, where $\mathbf{T} : \mathbb{R}^d \to \mathbb{R}^d$, a triangular mapping if the $j$-th component $T_j$ depends only on $\mathbf{x}_{1:j}$. The name originates from the fact that the Jacobian matrix of such a mapping is triangular. For a triangular mapping $\mathbf{T}$, we define its monotonicity signature $\xi(\mathbf{T}) = (\xi(T_j))_{j \in 1:d}$ as:

$$\xi(T_j) = \begin{cases} 1 & \forall \mathbf{x}_{1:j-1} \in \mathbb{R}^{j-1}, \; T_j(\mathbf{x}_{1:j-1}, \cdot) \text{ is s.m.i.} \\ -1 & \forall \mathbf{x}_{1:j-1} \in \mathbb{R}^{j-1}, \; T_j(\mathbf{x}_{1:j-1}, \cdot) \text{ is s.m.d.} \\ 0 & \text{otherwise} \end{cases}$$

where s.m.i. and s.m.d. abbreviate strictly monotonically increasing and decreasing, respectively. If $d = \sum_{j=1}^{d} \xi(T_j)$, i.e., every component is consistently s.m.i., we call $\mathbf{T}$ a triangular monotonic increasing (TMI) mapping. TMI mappings are closely related to KR transport:

**Lemma 5.1** ([Jaini et al. 2019](), Theorem 1). *Given any two distributions $P$ and $P'$ on $\mathbb{R}^d$ with strictly positive densities, if a TMI mapping $\mathbf{T} : \mathbb{R}^d \to \mathbb{R}^d$ satisfies $P = \mathbf{T}_\sharp P'$, then $\mathbf{T}$ is almost surely equivalent to the KR transport.*

TMI mappings can be further generalized. For a triangular mapping $\mathbf{T}$, if $d = \sum_{j=1}^{d} |\xi(T_j)|$, i.e., every component is either consistently s.m.i. or s.m.d., we call $\mathbf{T}$ a triangular monotonic (TM) mapping. TM mappings, in addition to being bijective due to monotonicity, exhibit special properties: they remain TM mappings under inversion, composition, and selection of contiguous components. Moreover, their monotonicity signatures adhere to specific rules, which are elaborated in Appendix A.4. Crucially, for two TM mappings $\mathbf{T}^{(1)}$ and $\mathbf{T}^{(2)}$ with identical monotonicity signatures, $\mathbf{T}^{(2)} \circ (\mathbf{T}^{(1)})^{-1}$ is always a TMI mapping.

Next, we aim to relate the solution mapping $\mathbf{\Gamma}$ of an SCM to TM mappings. However, TM mappings are defined over the index set $1:d$, whereas SCMs are defined over the nested index set $\mathcal{I} = \{(i, j) \mid i \in \mathcal{I}, j \in 1:d_i\}$. To bridge this gap, we introduce the concept of flattening, a generalization of permutation. For a finite index set $\mathcal{I}$, we call a bijection $\iota : \mathcal{I} \to 1:|\mathcal{I}|$ a flattening mapping of $\mathcal{I}$, and flattening refers to re-indexing according to $\iota$. Define the re-indexing $\mathbf{P}_\iota : \prod_{i \in \mathcal{I}} \Omega_{X_i} \to \prod_{i=1}^{|\mathcal{I}|} \Omega_{X_i}$ such that for every $i \in \mathcal{I}$,

$$\mathbf{P}_\iota((x_i)_{i \in \mathcal{I}}) = (x_{\iota^{-1}(j)})_{j \in \iota[\mathcal{I}]},$$

where $\iota[\mathcal{I}]$ is the image set. Since $\iota$ is bijective, $\mathbf{P}_\iota$ is also bijective, and for any $J \subseteq 1:|\mathcal{I}|$, we have

$$(\mathbf{P}_\iota^{-1})((x_j)_{j \in J}) = (x_{\iota(i)})_{i \in \iota^{-1}[J]},$$

where $\iota^{-1}[\mathcal{I}]$ is the pre-image set.

For the nested index set $\mathcal{I}$ and a total order $\leq$ on $\mathcal{I}$, if a flattening mapping $\iota$ satisfies $\iota(i, j) - \iota(i, j') = j - j'$

for any $(i, j), (i, j') \in \mathcal{I}$ and $\iota(i, j) < \iota(i', j')$ for any $(i, j), (i', j') \in \mathcal{I}$ with $i < i'$, we call it a vectorization under $\leq$. In other words, vectorization merges all random vectors into a unified order. Based on whether the solution mapping $\mathbf{\Gamma}$ is a TM map under vectorization, we define a more specialized class of SCMs:

**Definition 5.2** (Triangular Monotonic SCM (TM-SCM)). A recursive SCM $\mathcal{M}$ is called a triangular monotonic SCM (TM-SCM) if there exists a vectorization $\iota$ under a causal order such that the re-indexed $\mathbf{P}_\iota \circ \mathbf{\Gamma}$ is a TM mapping.

**Proposition 5.3.** *A recursive SCM $\mathcal{M}$ is a TM-SCM if and only if $f_i(\mathbf{v}, \cdot)$ is a TM mapping and there exists $\xi_i$ such that $\xi(f_i(\mathbf{v}, \cdot)) = \xi_i$ for every $i \in \mathcal{I}$ and all $\mathbf{v} \in \Omega_{\mathbf{V}}$.*

## 5.1. $\sim_{\mathrm{EI}}$-Identification for TM-SCM

As a subclass of BSCMs, TM-SCMs allow the assumption set in Theorem 4.8 to be further simplified. The new identifiability strategy is specifically tied to TM-SCMs. Let $\mathcal{A}_{\{\text{TM-SCM}, \leq, \mathrm{M}, P_{\mathbf{V}}\}} = \{\mathcal{A}_{\text{TM-SCM}}, \mathcal{A}_{\leq}, \mathcal{A}_{\mathrm{M}}, \mathcal{A}_{P_{\mathbf{V}}}\}$, where $\mathcal{A}_{\text{TM-SCM}}$ states that $\mathcal{M}$ is a TM-SCM. Then:

**Corollary 5.4** (EI-ID from Triangular Monotonicity). *An SCM is $\sim_{\mathrm{EI}}$-identifiable from $\mathcal{A}_{\{TM\text{-}SCM, \leq, M, P_{\mathbf{V}}\}}$.*

**Sketch of Proof** By combining the definition of TM-SCMs with the properties of TM mappings, any counterfactual transport in a TM-SCM is a TMI mapping. Furthermore, under Markovianity, given the causal order $\leq$ and observational distribution $P_{\mathbf{V}}$, the KR transport is almost uniquely determined by Lemmas 4.7 and 5.1, which then implies Theorem 4.8. A detailed proof is provided in Appendix A.4.

Corollary 5.4 (together with Theorem 3.2) can be viewed as a strengthened version for Markovian CBNs, which exhibits $\sim_{\mathcal{L}_2}$-identifiability, i.e., SCMs are $\sim_{\mathcal{L}_2}$-identifiable from $\mathcal{A}_{\{\leq, \mathrm{M}, P_{\mathbf{V}}\}}$. Corollary 5.4 augments $\mathcal{A}_{\{\leq, \mathrm{M}, P_{\mathbf{V}}\}}$ with the additional assumption $\mathcal{A}_{\text{TM-SCM}}$, strengthening $\sim_{\mathcal{L}_2}$-identifiability to $\sim_{\mathcal{L}_3}$-identifiability.

Moreover, this extends and unifies the theoretical results in ([Lu et al.](), 2020, Theorem 1), ([Nasr-Esfahany et al.](), 2023, Theorem 5.1), and ([Scetbon et al.](), 2024, Theorem 2.14). Specifically, these theorems focus on counterfactual identifiability from the perspectives of bijectivity, monotonicity, and Markovianity, demonstrating identifiability based on *abduction-action-prediction*. Compared to these results, our theoretical contributions offer the following advancements: (i) Generalizing the measurable space of each endogenous variable from a scalar $\mathbb{R}$ to a vector $\mathbb{R}^{d_i}$, enabling support for a broader class of SCMs; (ii) Demonstrating $\sim_{\mathrm{EI}}$-identifiability based on exogenous isomorphism theory (i.e. Theorem 3.2), which implies $\sim_{\mathcal{L}_3}$-identifiability rather than merely the identifiability of counterfactual outcomes.

## 6. Neural TM-SCM

**Problem formulation**  Consider the true underlying SCM $\mathcal{M}^* = \langle \mathcal{I}, \Omega_{\mathbf{V}}, \Omega_{\mathbf{U}}^*, \boldsymbol{f}^*, P_{\mathbf{U}}^* \rangle$, where $\Omega_{\mathbf{U}}^*$, $\boldsymbol{f}^*$, and $P_{\mathbf{U}}^*$ are unknown. Assuming $\mathcal{M}^*$ is a Markovian TM-SCM with causal order $\leq$, how can one construct a parameterized proxy SCM $\mathcal{M}_\theta = \langle \mathcal{I}, \Omega_{\mathbf{V}}, \Omega_{\mathbf{U}_\theta}, \boldsymbol{f}_\theta, P_{\mathbf{U}_\theta} \rangle$ based on the observational dataset $\mathcal{D}_{\mathbf{V}} = \{\mathbf{v}_i \mid \mathbf{v}^{(i)} \sim P_{\mathbf{V}}\}_{i=1}^N$ such that $\mathcal{M}_\theta$ is as $\mathcal{L}_3$-consistent with $\mathcal{M}^*$ as possible?

According to Corollary 5.4 and Theorem 3.2, this problem is equivalent to constructing $\mathcal{M}_\theta$ such that it satisfies $\mathcal{A}_{\{\text{TM-SCM}, \leq, \text{M}, P_{\mathbf{V}}\}}$ as closely as possible. This entails four construction objectives: (i) **G1**: $\mathcal{M}_\theta \models \mathcal{A}_{\text{TM-SCM}}$, meaning that $\mathcal{M}_\theta$ belongs to the TM-SCM class; (ii) **G2**: $\mathcal{M}_\theta \models \mathcal{A}_{\leq}$, meaning that $\leq$ is one of the causal orders of $\mathcal{M}_\theta$. (iii) **G3**: $\mathcal{M}_\theta \models \mathcal{A}_{\text{M}}$, meaning that $\mathcal{M}_\theta$ satisfies Markovianity. (iv) **G4**: $\mathcal{M}_\theta \models \mathcal{A}_{P_{\mathbf{V}}}$, meaning that $\mathcal{M}_\theta$ induces the observational distribution $P_{\mathbf{V}}$.

To satisfy **G1**, $\mathcal{M}_\theta$ must be a TM-SCM or one of its special subclasses. Furthermore, in this paper, we primarily focus on neural networks that provide the parameters $\theta$ for $\mathcal{M}_\theta$, thereby referring to it as a neural TM-SCM. Previous works have constructed neural SCMs that satisfy this requirement in the scalar case; we categorize these works into four categories and propose corresponding prototype models as extensions to the vector case:

- **DNME**: Constraint the causal mechanism $f_{i,\theta}$ to be a TM mapping, such that the causal mechanism

$$f_{i,\theta}(\mathbf{v}_{\text{pa}(i)}, \mathbf{u}_i) = \mathbf{b}_{i,\theta}(\mathbf{v}_{\text{pa}(i)}) + \mathbf{a}_{i,\theta}(\mathbf{v}_{\text{pa}(i)}) \odot \mathbf{u}_i, \quad (2)$$

where $\mathbf{a}_{i,\theta}(\mathbf{v}_{\text{pa}(i)}), \mathbf{b}_{i,\theta}(\mathbf{v}_{\text{pa}(i)}) \in \mathbb{R}^{d_i}$ with vector $\mathbf{a}_{i,\theta}$ always strictly positive, and $\odot$ denotes component-wise multiplication. Since the Jacobian matrix of $f_{i,\theta}$ w.r.t. $\mathbf{u}_i$ is diagonal, it is named **D**iagonal **N**oise **ME**chanism, such as LSNM (Immer et al., 2023).
- **TNME**: Constraint the causal mechanism $f_{i,\theta}$ to be a TM mapping, such that the causal mechanism

$$f_{i,\theta}(\mathbf{v}_{\text{pa}(i)}, \mathbf{u}_i) = \mathbf{b}_{i,\theta}(\mathbf{v}_{\text{pa}(i)}) + \left(\mathbf{A}_{i,\theta}(\mathbf{v}_{\text{pa}(i)})\right)\mathbf{u}_i^{\mathsf{T}}, \quad (3)$$

where $\mathbf{A}_{i,\theta}(\mathbf{v}_{\text{pa}(i)}) \in \mathbb{R}^{d_i \times d_i}, \mathbf{b}_{i,\theta}(\mathbf{v}_{\text{pa}(i)}) \in \mathbb{R}^{d_i}$ with matrix $\mathbf{A}_{i,\theta}$ always a strictly positive lower triangular matrix. Since the Jacobian matrix of $f_{i,\theta}$ w.r.t. $\mathbf{u}_i$ is lower triangular, it is named **T**riangular **N**oise **ME**chanism, such as FiP (Scetbon et al., 2024).
- **CMSM**: Constraint the re-indexed solution mapping $\mathbf{P}_\iota \circ \boldsymbol{\Gamma}_\theta$ to be a TM mapping by composing multiple TM mappings

$$\mathbf{P}_\iota \circ \boldsymbol{\Gamma}_\theta = \mathbf{T}_{1,\theta} \circ \cdots \circ \mathbf{T}_{n,\theta}, \quad (4)$$

where each $\mathbf{T}_{i,\theta}$ can be an autoregressive affine transformation in normalizing flows (Dinh et al., 2017). Since

the solution map $\boldsymbol{\Gamma}_\theta$ is constructed by composing multiple mappings, it is named **C**omposed **M**apped **S**olution **M**apping, such as CausalNF (Javaloy et al., 2023).
- **TVSM**: Constraints the re-indexed solution mapping $\mathbf{P}_\iota \circ \boldsymbol{\Gamma}_\theta$ to be a TM mapping by defining it as the flow constructed from the solution to the ODE

$$\begin{cases} \mathrm{d}\boldsymbol{x}(t) = \boldsymbol{v}_\theta(\boldsymbol{x}(t), t)\,\mathrm{d}t, \\ \boldsymbol{x}(0) = \mathbf{v}_0, \end{cases} \quad (5)$$

where the vector field $\boldsymbol{v}_\theta$ is a Lipschitz continuous triangular mapping. According to the Picard–Lindelöf theorem, such flows are TMI, see Lemma A.21 in Appendix A.4. Since the solution map $\boldsymbol{\Gamma}_\theta$ is implied by a triangular velocity field, it is named **T**riangular **V**elocity **S**olution **M**apping, such as CFM (Khoa Le et al., 2025).

The relationship between these related works and TM-SCM is detailed in Appendix B.4, and the implementation details of the four prototype models can be found in Appendix C.

To satisfy **G2**, the causal mechanisms $\boldsymbol{f}_\theta$ in the constructed SCM need to maintain a specific causal structure to derive the causal order $\leq$. When parameterizing the causal mechanisms $f_{i,\theta}$ as in (Xia et al., 2023), we address the issue of inappropriate additional independence assumptions—arising from only knowing the causal order—by setting $\text{pa}(i) = \{j \mid j \leq i, j \in \mathcal{I}\}$. Similarly, when indirectly modeling the solution map $\boldsymbol{\Gamma}_\theta$ using an autoregressive generative model as in (Javaloy et al., 2023), we align the autoregressive order with the causal order.

To satisfy **G3**, the modeled exogenous distribution is required to satisfy $P_{\mathbf{U}_\theta} = \prod_{i \in \mathcal{I}} P_{U_{i,\theta}}$. To enhance the expressiveness of the exogenous distribution and to indirectly indicate that identifiability is independent of the specific implementation of the exogenous distribution, we employ unconstrained normalizing flows. Specifically, the log-likelihood of each $P_{U_{i,\theta}}$ is given by the change of variables formula:

$$\log p_{U_{i,\theta}}(u_i) = \log p_{Z_{i,\theta}}(\mathcal{T}_{i,\theta}^{-1}(u_i)) + \log\left|\det \mathbf{J}_{\mathcal{T}_{i,\theta}^{-1}}(u_i)\right|, \quad (6)$$

where $\mathcal{T}_{i,\theta} : \mathcal{Z}_i \to \mathcal{U}_i$ is MAF (Papamakarios et al., 2017), and $\left|\det \mathbf{J}_{\mathcal{T}_{i,\theta}^{-1}}(u_i)\right|$ is the Jacobian determinant at $u_i$.

To satisfy **G4**, in generative model learning, the observational dataset $\mathcal{D}_{\mathbf{V}}$ is commonly used to optimize a learning objective, ensuring that the proxy SCM $\mathcal{M}_\theta$ induces an observational distribution $P_{\mathbf{V},\theta}$ that closely approximates the true observational distribution $P_{\mathbf{V}}$. To achieve this, we adopt the traditional maximum likelihood (MLE) approach, corresponding to the negative log-likelihood (NLL) loss:

$$\arg\min_\theta -\sum_{i=1}^N \log p_{\mathbf{V}_\theta}(\mathbf{v}^{(i)}), \quad (7)$$

where $\log p_{\mathbf{V}_\theta}$ is the log-likelihood of the modeled observational distribution. Since TM-SCM ensures a bijective $\mathbf{\Gamma}_\theta$, the log-likelihood $\log p_{\mathbf{V}_\theta}(\mathbf{v}^{(i)})$ at $\mathbf{v}^{(i)}$ can be computed using the change of variables formula.

## 7. Related Works

**Isomorphism and counterfactual identifiability**  Several prior works have introduced notions similar to exogenous isomorphism and shown that SCMs satisfying these equivalence relations enjoy counterfactual consistency, mirroring the result of Theorem 3.2. This body of work includes counterfactual equivalence as defined in (Peters et al., 2017), which requires identical exogenous distributions; BGM equivalence, introduced for BGM within BSCM in (Nasr-Esfahany et al., 2023); LCM isomorphism, proposed by (Brehmer et al., 2022) for causal representation learning and proven, under additional conditions, to coincide with LCM counterfactual consistency; and domain counterfactual equivalence between ILDs, together with necessary and sufficient conditions, as defined in (Zhou et al., 2024). The latter three lines of work additionally assume that the solution mapping $\mathbf{\Gamma}$ is bijective, whereas Theorem 3.2 applies to arbitrary recursive SCMs.

**TM-SCMs and their identifiability**  Previously, we classified four construction prototypes of TM-SCMs from a construction perspective; a complementary perspective considers the tasks these models address and the corresponding identifiability guarantees. For cause-effect identification, special TM-SCMs—including LiNGAM (Shimizu et al., 2006), ANM (Hoyer et al., 2008), PNL (Zhang & Hyvärinen, 2009), LSNM (Immer et al., 2023), and CAREFL (Khemakhem et al., 2021)—have been proven identifiable with respect to causal direction. For causal effect estimation, CausalNF (Javaloy et al., 2023) establishes representation identifiability of TMI, and similar TM-SCM constructions are adopted by StrAF (Chen et al., 2023), CCNF (Zhou et al., 2025), and CFM (Khoa Le et al., 2025), which therefore inherit the same theoretical guarantees. In counterfactual identification, (Lu et al., 2020, Theorem 1), (Nasr-Esfahany et al., 2023, Theorem 5.1), and (Scetbon et al., 2024, Theorem 2.14) each demonstrate that TM-SCM entails counterfactual identifiability under different settings, and these theorems are all special cases of Corollary 5.4.

**Counterfactual inference with neural SCMs**  Proxy SCMs built with neural network components are widely employed for counterfactual reasoning by directly learning from observational or interventional data. Several methods focus on the tractability of inference rather than identifiability by adopting different neural modules, including DSCM (Pawlowski et al., 2020), Diff-SCM (Sanchez & Tsaftaris, 2022), and VACA (Sánchez-Martin et al., 2022). Other approaches obtain identifiability for a subset of counterfactual queries; for example, CVAE-SCM (Karimi et al., 2020) is identifiable for specific counterfactual queries from causal sufficiency and observational distribution, and NCM (Xia et al., 2023) shows a duality between the identifiability of structure-constrained proxy SCMs and non-parametric identification results. Once parametric assumptions are introduced, complete counterfactual identifiability becomes attainable, exemplified by any subclass of neural TM-SCMs. Additional examples are provided in Appendix B.4.

## 8. Experiments

To demonstrate that neural TM-SCM can effectively address the counterfactual consistency problem in practice, we conducted experiments on synthetic adatasets. These experiments were designed to showcase the model's ability to generate counterfactual results that are consistent with the test set, using only the endogenous samples drawn from the observational distribution as the training set. [3]

**Datasets**  The experiments involve the following synthetic datasets, with details described in Appendix D.1.

- **TM-SCM-SYM**: A collection of four small datasets (BARBELL, STAIR, FORK, BACKDOOR) with up to 4 causal variables, using exogenous distributions that are standard or Markovian multivariate normals and manually defined TM causal mechanisms.
- **ER-DIAG-50** and **ER-TRIL-50**: Each contains 50 datasets with 3–8 causal variables, Markovian multivariate normal exogenous distributions, and Erdős-Rényi causal graphs (edge probability 0.5). ER-DIAG-50 ensures diagonal Jacobians, while ER-TRIL-50 ensures lower triangular Jacobians for TM mappings.

**Metrics**  To evaluate trained $\mathcal{M}_\theta$, we compute $\mathrm{OBS_{WD}}$ (Wasserstein distance) for the fit to the observational distribution and $\mathrm{CTF_{RMSE}}$ (root mean square error) for $\mathcal{L}_3$-consistency with ground truth counterfactual outcomes.

**Ablation on TM-SCM-SYM**  To provide empirical evidence for Corollary 5.4 and Theorem 3.2, and show that neural TM-SCM addresses counterfactual consistency, we performed a small-scale ablation study on TM-SCM-SYM, including: (i) **w/o O**: Reverses causal order $\leq$, ablating $\mathcal{A}_\leq$; (ii) **w/o M**: Uses non-Markovian exogenous distributions, ablating $\mathcal{A}_\mathrm{M}$; (iii) **w/o T**: Constructs non-triangular SCM, ablating $\mathcal{A}_\mathrm{TM\text{-}SCM}$ (supported only by CMSM and TVSM).

Figure 1 shows scatter plots of $\mathrm{OBS_{WD}}$ and $\mathrm{CTF_{RMSE}}$ on the validation set. Configurations w/o O and w/o T fail to converge, highlighting the necessity of $\mathcal{A}_\leq$ and $\mathcal{A}_\mathrm{TM\text{-}SCM}$.

---

[3] Code is available at: https://github.com/cyisk/tmscm

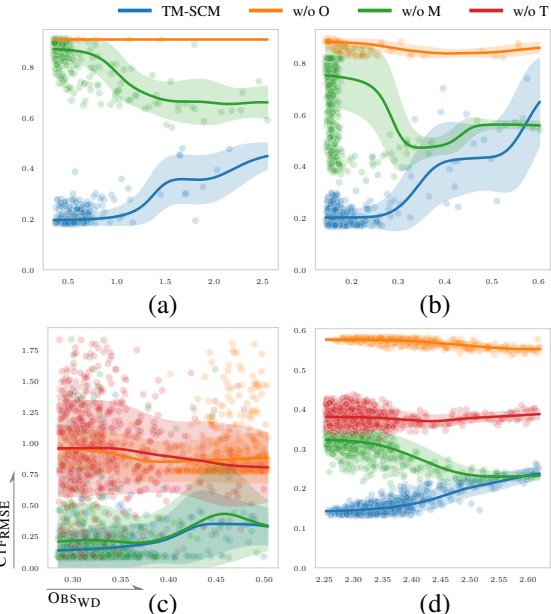

*Table 1.* Ablation results of neural TM-SCM on ER-DIAG-50 and ER-TRIL-50. The values shown are the means of 50 experiments, with the subscript representing the 95% CI. The best-performing results are highlighted in bold.

| METHOD | | ER-DIAG-50 | ER-TRIL-50 |
|---|---|---|---|
| DNME | - | $\mathbf{0.53_{\pm 0.05}}$ | $\mathbf{0.51_{\pm 0.12}}$ |
| | W/O O | $0.78_{\pm 0.05}$ | $0.89_{\pm 0.10}$ |
| | W/O M | $0.62_{\pm 0.04}$ | $0.58_{\pm 0.10}$ |
| TNME | - | $\mathbf{0.47_{\pm 0.05}}$ | $\mathbf{0.55_{\pm 0.12}}$ |
| | W/O O | $11.24_{\pm 20.98}$ | $6.41_{\pm 9.84}$ |
| | W/O M | $0.62_{\pm 0.04}$ | $0.73_{\pm 0.21}$ |
| CMSM | - | $\mathbf{0.37_{\pm 0.05}}$ | $\mathbf{0.42_{\pm 0.12}}$ |
| | W/O O | $2.64_{\pm 3.72}$ | $2.12_{\pm 2.49}$ |
| | W/O M | $1.69_{\pm 2.60}$ | $0.75_{\pm 0.49}$ |
| | W/O T | $0.64_{\pm 0.05}$ | $1.25_{\pm 1.29}$ |
| TVSM | - | $\mathbf{0.46_{\pm 0.05}}$ | $\mathbf{0.50_{\pm 0.12}}$ |
| | W/O O | $0.79_{\pm 0.04}$ | $0.88_{\pm 0.10}$ |
| | W/O M | $0.53_{\pm 0.05}$ | $0.53_{\pm 0.11}$ |
| | W/O T | $0.67_{\pm 0.05}$ | $0.78_{\pm 0.12}$ |

*Figure 1.* Ablation results of neural TM-SCMs on TM-SCM-SYM. Colored curves depict sliding-window regressions, with shaded areas showing 95% CI. (a) DNME for BARBELL; (b) TNME for STAIR; (c) CMSM for FORK; (d) TVSM for BACKDOOR.

Most w/o M settings result in higher $\mathrm{CTF_{RMSE}}$, emphasizing the role of $\mathcal{A}_M$. CMSM shows slightly lower stability than other methods. Additional results are in Appendix D.4.

**Ablation on ER-DIAG-50 and ER-TRIL-50**   We further conducted a more comprehensive evaluation on ER-DIAG-50 and ER-TRIL-50, reinforcing the generality of the theory through experiments on a wide variety of synthetic SCMs. The ablations in these experiments also targeted $\mathcal{A}_\le$, $\mathcal{A}_M$, and $\mathcal{A}_{\text{TM-SCM}}$.

Table 1 presents the final $\mathrm{CTF_{RMSE}}$ on the ER-DIAG-50 and ER-TRIL-50 test sets for each method and its ablations. The results demonstrate that violating $\mathcal{A}_\le$, $\mathcal{A}_M$, or $\mathcal{A}_{\text{TM-SCM}}$ significantly degrades $\mathcal{L}_3$-consistency, emphasizing the importance of these assumptions and the validity of the theory to ensure consistent counterfactual inference. Additional results on these datasets can be found in Appendix D.4.

## 9. Conclusion

In this work, we explored $\sim_{\mathcal{L}_3}$-identifiability, the strongest form of causal model indistinguishability within the PCH. By simplifying the problem through the introduction of exogenous isomorphism and $\sim_{\text{EI}}$-identifiability, we developed concrete methods to achieve identifiability in BSCMs and TM-SCMs. These findings unify and extend existing theories, providing reliability guarantees for counterfactual reasoning. Our empirical evaluations further validate the practicality of the proposed approach, paving the way for reliable applications in counterfactual modeling.

Acknowledgement (Camera ready)

## Acknowledgements

This work was supported in part by the National Key R&D Program of China (No. 2022ZD0120302).

## Impact Statement

This study aims to advance the field of causal inference by providing both theoretical and practical support for the consistency and trustworthiness of the results of counterfactual reasoning. In future applications, the proposed methods can enhance the reliability of reasoning in relevant practical scenarios. However, we emphasize the necessity of maintaining close collaboration with domain experts and stakeholders when applying these models in practice. It is particularly important to assess whether the theoretical assumptions proposed in this paper are satisfied to mitigate the risks of potential negative consequences.

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

# A. Proofs

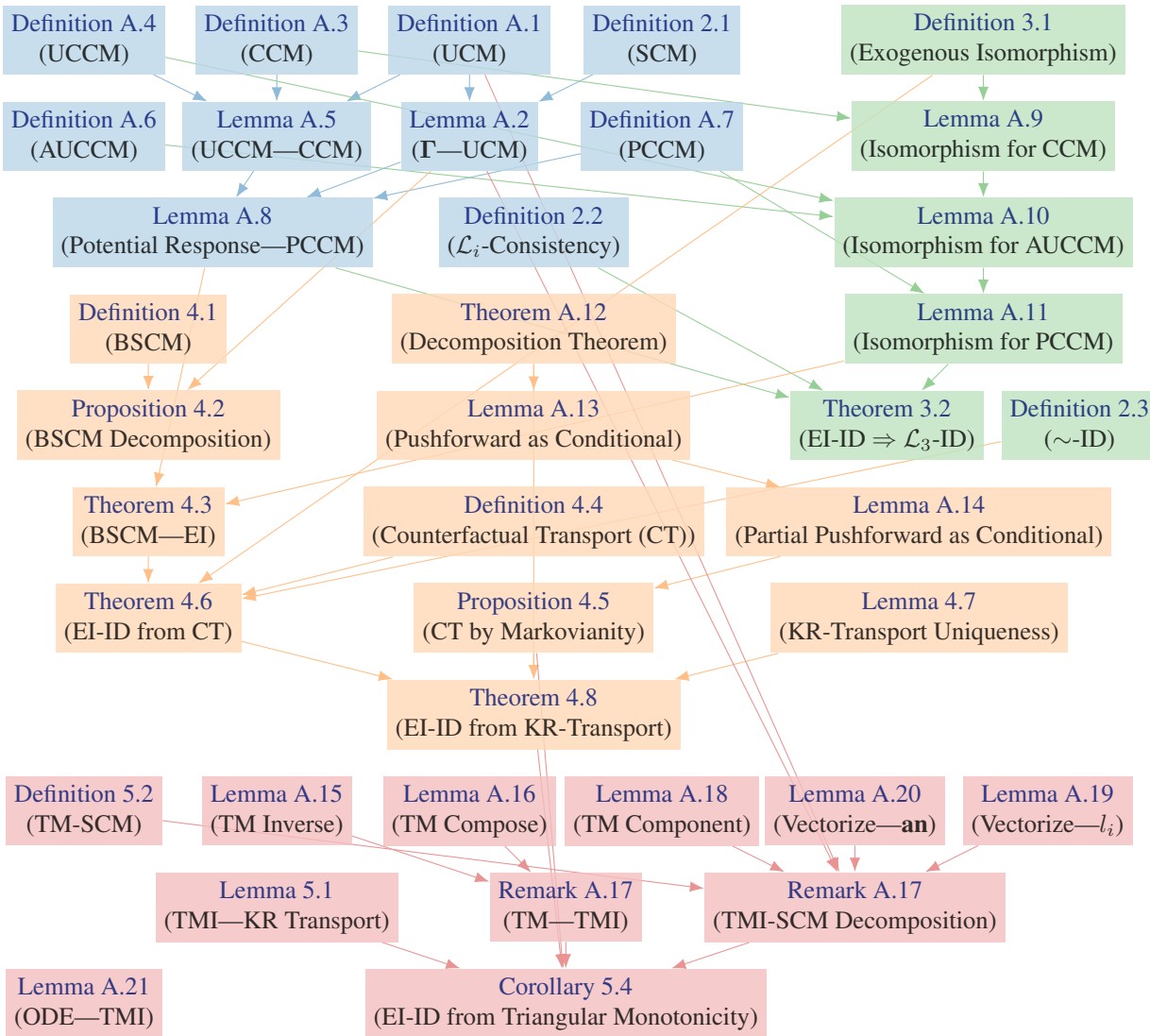

*Figure 2.* Overview of all theorems discussed in the main text and appendix, along with their dependency graph. Nodes represent theorems, and edges indicate dependencie, directed from top to bottom. Different colors denote different topics: theorems related to recursive SCMs are marked in blue; theorems related to exogenous isomorphism are marked in green; theorems related to BSCMs are marked in yellow; and theorems related to TM-SCMs are marked in red.

## A.1. Background and Preliminaries

Definition 2.1 draws characteristics from both (Bongers et al., 2016) and (Bareinboim et al., 2022), but differs in the following aspects: (i) we use a shared index set $\mathcal{I}$ for exogenous and endogenous variables, rather than using an additional index set $\mathcal{J}$ for endogenous variables, because this paper does not focus on the relationships between exogenous variables; (ii) to partially compensate for the limitation introduced in (i), we allow $P_{\mathbf{U}} = \prod_{i \in \mathcal{I}} P_{U_i}$ to not necessarily be a product measure. This relaxation permits endogenous variables to be dependent unless the SCM is Markovian.

Following Definition 2.1, an SCM describes the functional relationships between endogenous and exogenous variables through a set of deterministic equations of the form $v_i = f_i(\mathbf{v}_{\mathrm{pa}(i)}, u_i)$, known as structural equations. (Bongers et al., 2016) discuss the solvability of this system of equations. Specifically, if there exists a pair $(\mathbf{V}, \mathbf{U})$ such that $\mathbf{V} = \boldsymbol{f}(\mathbf{V}, \mathbf{U})$ holds almost surely and the distribution of $\mathbf{U}$ is exactly $P_{\mathbf{U}}$, then the SCM is said to have a solution. If there exists a function

$\Gamma$ such that for almost all $\mathbf{u} \in \Omega_{\mathbf{U}}$ and all $\mathbf{v} \in \Omega_{\mathbf{V}}$, $\mathbf{v} = \Gamma(\mathbf{u})$ implies $\mathbf{v} = \boldsymbol{f}(\mathbf{v}, \mathbf{u})$, then the SCM is said to be solvable. According to (Bongers et al., 2016, Theorem 3.2), the SCM has a solution if and only if it is solvable.

A recursive SCM is a special type of SCM with certain desirable properties. In this subsection, we will prove a fundamental lemma on recursive SCMs to facilitate subsequent proofs and introduce some notations. For an index $i \in \mathcal{I}$ in a recursive SCM, the lower set $\mathrm{an}(i) = \{j \mid j \prec i\}$ of the partial order $\preceq$ is called the causal ancestors of $i$, while the lower set $\mathrm{pr}(i) = \{j \mid j < i\}$ of its linear extension $\leq$ is called the causal prefix of $i$. Additionally, $\mathrm{an}^*(i) = \mathrm{an}(i) \cup \{i\}$ and $\mathrm{pr}^*(i) = \mathrm{pr}(i) \cup \{i\}$. Consider a recursive SCM $\mathcal{M} = \langle \mathcal{I}, \Omega_{\mathbf{V}}, \Omega_{\mathbf{U}}, \boldsymbol{f}, P_{\mathbf{U}} \rangle$, we can recursively expand the endogenous part of $f_i$ to obtain a new function:

**Definition A.1** (Unrolled Causal Mechanism (UCM)). For each causal mechanism $f_i : \Omega_{\mathbf{V}_{\mathrm{pa}(i)}} \times \Omega_{U_i} \to \Omega_{V_i}$ in $\mathcal{M}$, let $\tilde{f}_i : \Omega_{\mathbf{U}_{\mathrm{an}^*(i)}} \to \Omega_{V_i}$ be the result of recursively unrolling the endogenous part $\Omega_{\mathbf{V}_{\mathrm{pa}(i)}}$ of the causal mechanism $f_i$, such that for any $\mathbf{u} \in \Omega_{\mathbf{U}}$,

$$\tilde{f}_i(\mathbf{u}_{\mathrm{an}^*(i)}) = f_i\left(\left(\tilde{f}_k(\mathbf{u}_{\mathrm{an}^*(k)})\right)_{k \in \mathrm{pa}(i)}, u_i\right).$$

This is referred to as the Unrolled Causal Mechanism (UCM).

The following lemma demonstrates that the tuple of UCMs constitutes the solution mapping $\Gamma$ of a recursive SCM, thereby indirectly proving the existence and uniqueness of solutions for recursive SCMs.

**Lemma A.2.** *For all* $\mathbf{u} \in \Omega_{\mathbf{U}}$, $\mathbf{v} = \boldsymbol{f}(\mathbf{v}, \mathbf{u})$ *if and only if* $\mathbf{v} = \Gamma(\mathbf{u})$, *where* $\Gamma(\mathbf{u}) = (\tilde{f}_i(\mathbf{u}_{an^*(i)}))_{i \in \mathcal{I}}$.

*Proof.* Let $\mathbf{u} \in \Omega_{\mathbf{U}}$ be arbitrary.

$\implies$ : Suppose the structural equations $\mathbf{v} = \boldsymbol{f}(\mathbf{v}, \mathbf{u})$, meaning each component satisfies $v_i = f_i(\mathbf{v}_{\mathrm{pa}^*(i)}, u_i)$. We proceed by induction based on the partial order $\preceq$. When $\mathrm{pa}(i) = \emptyset$, we have $v_i = f_i(u_i)$, and by the definition of UCM, $\tilde{f}_i(u_i) = f_i(u_i) = v_i$. When $\mathrm{pa}(i) \neq \emptyset$, assume that for each $k \in \mathrm{pa}(i)$, $v_k = \tilde{f}_k(\mathbf{u}_{\mathrm{an}^*(k)})$ holds. Then,

$$\begin{aligned}
\tilde{f}_i(\mathbf{u}_{\mathrm{an}^*(i)}) &= f_i((\tilde{f}_k(\mathbf{u}_{\mathrm{an}^*(k)}))_{k \in \mathrm{pa}(i)}, u_i) && \text{Definition A.1} \\
&= f_i(\mathbf{v}_{\mathrm{pa}(i)}, u_i). && \text{Induction} \\
&= v_i.
\end{aligned}$$

Thus, for each $i \in \mathcal{I}$, we have $v_i = \tilde{f}_i(\mathbf{u}_{\mathrm{an}^*(i)})$, implying that $(v_i)_{i \in \mathcal{I}} = (\tilde{f}_i(\mathbf{u}_{\mathrm{an}^*(i)}))_{i \in \mathcal{I}}$, i.e., $\mathbf{v} = \Gamma(\mathbf{u})$.

$\impliedby$ : Suppose $\mathbf{v} = \Gamma(\mathbf{u})$, meaning each component satisfies $v_i = \tilde{f}_i(\mathbf{u}_{\mathrm{an}^*(i)})$. Then, for each $i \in \mathcal{I}$,

$$\begin{aligned}
v_i &= \tilde{f}_i(\mathbf{u}_{\mathrm{an}^*(i)}) \\
&= f_i((\tilde{f}_k(\mathbf{u}_{\mathrm{an}^*(k)}))_{k \in \mathrm{pa}(i)}, u_i) && \text{Definition A.1} \\
&= f_i(\mathbf{v}_{\mathrm{pa}(k)}, u_i). && v_k = \tilde{f}_k(\mathbf{u}_{\mathrm{an}^*(k)})
\end{aligned}$$

where the last equality follows from $v_k = \tilde{f}_k(\mathbf{u}_{\mathrm{an}^*(k)})$ for each $k \in \mathrm{pa}(i)$. Therefore, the structural equation $v_i = f_i(\mathbf{v}_{\mathrm{pa}(i)}, u_i)$ holds for every $i \in \mathcal{I}$, and thus the system of structural equations $\mathbf{v} = \boldsymbol{f}(\mathbf{v}, \mathbf{u})$ holds. $\square$

We decompose the problem and, based on the causal mechanisms, define the following compound structure as an extension of the causal mechanisms at the counterfactual level, establishing a connection with the potential responses in the PCH. Suppose there are $n$ submodels of recursive SCMs, $\mathcal{M}_{[\mathbf{x}_1]}, \mathcal{M}_{[\mathbf{x}_2]}, \dots, \mathcal{M}_{[\mathbf{x}_n]}$, and let the UCM corresponding to $f_{i[\mathbf{x}_j]}$ in each submodel $\mathcal{M}_{[\mathbf{x}_j]}$ be denoted by $\tilde{f}_{i[\mathbf{x}_j]}$.

**Definition A.3** (Compound Causal Mechanism (CCM)). Let $f_{i[\mathbf{x}_*]} : \prod_{j=1}^{n} \Omega_{\mathbf{V}_{\mathrm{pa}(i)}} \times \Omega_{U_i} \to \prod_{j=1}^{n} \Omega_{V_i}$ be the tuple of causal mechanisms indexed by $i \in \mathcal{I}$ across the submodels, denoted as $(f_{i[\mathbf{x}_j]})_{j \in 1:n}$, where $\mathbf{x}_* = (\mathbf{x}_j)_{j \in 1:n}$. This satisfies, for each $j \in 1 : n$, all $\mathbf{v}_j \in \Omega_{\mathbf{V}}$, and all $u_i \in \Omega_{U_i}$,

$$f_{i[\mathbf{x}_*]}((\mathbf{v}_{\mathrm{pa}(i),j})_{j \in 1:n}, u_i) = (f_{i[\mathbf{x}_j]}(\mathbf{v}_{\mathrm{pa}(i),j}, u_i))_{j \in 1:n}.$$

This is referred to as the Compound Causal Mechanism (CCM).

**Definition A.4** (Unrolled Compound Causal Mechanism (UCCM)). Let $\tilde{f}_{i[\mathbf{x}_*]} : \Omega_{\mathbf{U}_{\mathrm{an}^*(i)}} \to \prod_{j=1}^n \Omega_{V_i}$ be the result of recursively unrolling the $\prod_{j=1}^n \Omega_{\mathbf{V}_{\mathrm{pa}(i)}}$ part of the CCM $f_{i[\mathbf{x}_*]}$, such that for all $\mathbf{u} \in \Omega_{\mathbf{U}}$,

$$\tilde{f}_{i[\mathbf{x}_*]}(\mathbf{u}_{\mathrm{an}^*(i)}) = f_{i[\mathbf{x}_*]}\left( \left( \tilde{f}_{k[\mathbf{x}_*]}(\mathbf{u}_{\mathrm{an}^*(k)}) \right)_{k \in \mathrm{pa}(i)}, u_i \right).$$

This is referred to as the Unrolled Compound Causal Mechanism (UCCM).

**Lemma A.5.** *For each $i \in \mathcal{I}$ and all $\mathbf{u} \in \Omega_{\mathbf{U}}$, the UCCM $\tilde{f}_{i[\mathbf{x}_*]}(\mathbf{u}_{an^*(i)})$ is equal to the tuple of UCMs $(\tilde{f}_{i[\mathbf{x}_j]}(\mathbf{u}_{an^*(i)}))_{j \in 1:n}$.*

*Proof.* Given any $\mathbf{u} \in \Omega_{\mathbf{U}}$, we proceed by induction based on the partial order $\preceq$.

When $\mathrm{pa}(i) = \emptyset$, according to the definition of UCCM, we have $\tilde{f}_{i[\mathbf{x}_*]}(\mathbf{u}_{\mathrm{an}^*(i)}) = \tilde{f}_{i[\mathbf{x}_*]}(u_i) = f_{i[\mathbf{x}_*]}(u_i)$. Additionally, according to the definition of UCM, we also have $\tilde{f}_{i[\mathbf{x}_j]}(\mathbf{u}_{\mathrm{an}^*(i)}) = \tilde{f}_{i[\mathbf{x}_j]}(u_i)$. Furthermore, based on the definition of CCM, $f_{i[\mathbf{x}_*]}(u_i) = (f_{i[\mathbf{x}_j]}(u_i))_{j \in 1:n}$. Therefore, $\tilde{f}_{i[\mathbf{x}_*]}(\mathbf{u}_{\mathrm{an}^*(i)}) = (\tilde{f}_{i[\mathbf{x}_j]}(\mathbf{u}_{\mathrm{an}^*(i)}))_{j \in 1:n}$.

When $\mathrm{pa}(i) \neq \emptyset$, assume that for each $k \in \mathrm{pa}(i)$, the following holds: $\tilde{f}_{k[\mathbf{x}_*]}(\mathbf{u}_{\mathrm{an}^*(k)}) = (\tilde{f}_{k[\mathbf{x}_j]}(\mathbf{u}_{\mathrm{an}^*(k)}))_{j \in 1:n}$. Then,

$$
\begin{aligned}
\tilde{f}_{i[\mathbf{x}_*]}(\mathbf{u}_{\mathrm{an}^*(i)}) &= f_{i[\mathbf{x}_*]}((\tilde{f}_{k[\mathbf{x}_*]}(\mathbf{u}_{\mathrm{an}^*(k)}))_{k \in \mathrm{pa}(i)}, u_i) && \text{Definition A.4} \\
&= f_{i[\mathbf{x}_*]}(((\tilde{f}_{k[\mathbf{x}_j]}(\mathbf{u}_{\mathrm{an}^*(k)}))_{j \in 1:n})_{k \in \mathrm{pa}(i)}, u_i) && \text{Induction} \\
&= \left( f_{i[\mathbf{x}_j]}((\tilde{f}_{k[\mathbf{x}_j]}(\mathbf{u}_{\mathrm{an}^*(k)}))_{k \in \mathrm{pa}(i)}, u_i) \right)_{j \in 1:n} && \text{Definition A.3} \\
&= (\tilde{f}_{i[\mathbf{x}_j]}(\mathbf{u}_{\mathrm{an}^*(k)}))_{j \in 1:n}. && \text{Definition A.1}
\end{aligned}
$$

Therefore, for each $i \in \mathcal{I}$ and all $\mathbf{u} \in \Omega_{\mathbf{U}}$, we have $\tilde{f}_{k[\mathbf{x}_*]}(\mathbf{u}_{\mathrm{an}^*(k)}) = (\tilde{f}_{k[\mathbf{x}_j]}(\mathbf{u}_{\mathrm{an}^*(k)}))_{j \in 1:n}$. $\qquad \square$

**Definition A.6** (Augmented Unrolled Compound Causal Mechanism (AUCCM)). Augmenting the domain of $\tilde{f}_{i[\mathbf{x}_*]}$ from $\Omega_{\mathbf{U}_{\mathrm{an}^*(i)}}$ to $\Omega_{\mathbf{U}_{\mathrm{pr}^*(i)}}$, resulting in $\tilde{g}_{i[\mathbf{x}_*]} : \Omega_{\mathbf{U}_{\mathrm{pr}^*(i)}} \to \prod_{j=1}^n \Omega_{V_i}$. This ensures that $\tilde{f}_{i[\mathbf{x}_*]}(\mathbf{u}_{\mathrm{an}^*(i)}) = \tilde{g}_{i[\mathbf{x}_*]}(\mathbf{u}_{\mathrm{pr}^*(i)})$ for all $\mathbf{u} \in \Omega_{\mathbf{U}}$, which is referred to as the Augmented Unrolled Compound Causal Mechanism (AUCCM).

**Definition A.7** (Prefix Compound Causal Mechanism (PCCM)). Define $\tilde{\mathbf{g}}_{\mathrm{pr}^*(i)[\mathbf{x}_*]} : \Omega_{\mathbf{U}_{\mathrm{pr}^*(i)}} \to \prod_{j=1}^n \Omega_{\mathbf{V}_{\mathrm{pr}^*(i)}}$ as a tuple of multiple AUCCMs, $(\tilde{g}_{k[\mathbf{x}_*]})_{k \in \mathrm{pr}^*(i)}$. This is referred to as the Prefix Compound Causal Mechanism (PCCM).

**Lemma A.8.** *For all $\mathbf{u} \in \Omega_{\mathbf{U}}$, the potential response $\mathbf{V}_{\mathcal{M}_*}(\mathbf{u})$ equals the PCCM $\tilde{\mathbf{g}}_{\mathcal{I}[\mathbf{x}_*]}(\mathbf{u})$.*

*Proof.* Given any $\mathbf{u} \in \Omega_{\mathbf{U}}$, consider the component indexed by $i$ in the $j$-th submodel, namely $(\mathbf{V}_{\mathcal{M}_*}(\mathbf{u}))_{i,j}$ and $(\tilde{\mathbf{g}}_{\mathcal{I}[\mathbf{x}_*]}(\mathbf{u}))_{i,j}$. According to the definition of potential response, $(\mathbf{V}_{\mathcal{M}_*}(\mathbf{u}))_{i,j} = (\mathbf{V}_{\mathcal{M}_{[\mathbf{x}_j]}}(\mathbf{u}))_i = (\mathbf{\Gamma}_{[\mathbf{x}_j]}(\mathbf{u}))_i$, where the $i$-th component of the solution mapping $\mathbf{\Gamma}_{[\mathbf{x}_j]}(\mathbf{u})$ is $\tilde{f}_{i[\mathbf{x}_j]}(\mathbf{u}_{\mathrm{an}^*(i)})$ according to Lemma A.2. On the other hand, by the definition of PCCM, $(\tilde{\mathbf{g}}_{\mathcal{I}[\mathbf{x}_*]}(\mathbf{u}))_{i,j} = (\tilde{g}_{i[\mathbf{x}_*]}(\mathbf{u}))_j$, and by the definition of AUCCM, $(\tilde{g}_{i[\mathbf{x}_*]}(\mathbf{u}))_j = (\tilde{f}_{i[\mathbf{x}_*]}(\mathbf{u}_{\mathrm{an}^*(i)}))_j$, where, according to Lemma A.5, $(\tilde{f}_{i[\mathbf{x}_*]}(\mathbf{u}_{\mathrm{an}^*(i)}))_j$ is the UCM $\tilde{f}_{i[\mathbf{x}_j]}(\mathbf{u}_{\mathrm{an}^*(i)})$ in the submodel $\mathcal{M}_{[\mathbf{x}_j]}$. Therefore,

$$(\mathbf{V}_{\mathcal{M}_*}(\mathbf{u}))_{i,j} = (\mathbf{\Gamma}_{[\mathbf{x}_j]}(\mathbf{u}))_i \overset{\text{Lemma A.2}}{=\!=\!=\!=\!=} \tilde{f}_{i[\mathbf{x}_j]}(\mathbf{u}_{\mathrm{an}^*(i)}) \overset{\text{Lemma A.5}}{=\!=\!=\!=\!=} (\tilde{f}_{i[\mathbf{x}_*]}(\mathbf{u}_{\mathrm{an}^*(i)}))_j = (\tilde{\mathbf{g}}_{\mathcal{I}[\mathbf{x}_*]}(\mathbf{u}))_{i,j}.$$

Since this equality holds for every component $i, j$, the potential response $\mathbf{V}_{\mathcal{M}_*}(\mathbf{u})$ equals the PCCM $\tilde{\mathbf{g}}_{\mathcal{I}[\mathbf{x}_*]}(\mathbf{u})$. $\qquad \square$

### A.2. Exogenous Isomorphism

The Definitions A.3, A.4, A.6 and A.7 enable us to progressively derive the isomorphism of causal mechanisms in exogenous isomorphism to the potential responses. Specifically, consider two recursive SCMs $\mathcal{M}^{(1)}$ and $\mathcal{M}^{(2)}$ such that $\mathcal{M}^{(1)} \sim_{\mathrm{EI}} \mathcal{M}^{(2)}$. The following lemmas then hold in succession. To prevent ambiguity during the proof process, we use the full notation $f_i^{(k)}(\mathbf{v}_{\mathrm{pa}^{(k)}(i)}, \cdot)$ instead of the abbreviated $f_i^{(k)}(\mathbf{v}, \cdot)$.

**Lemma A.9.** *For each $i \in \mathcal{I}$, for almost all $\mathbf{u}^{(1)} \in \Omega_{\mathbf{U}}^{(1)}$ and for all $\mathbf{v} \in \Omega_{\mathbf{V}}$, the CCM satisfies*

$$f_{i[\mathbf{x}_*]}^{(2)}(\mathbf{v}_{pa^{(2)}(i)}, h_i(u_i^{(1)})) = f_{i[\mathbf{x}_*]}^{(1)}(\mathbf{v}_{pa^{(1)}(i)}, u_i^{(1)}).$$

*Proof.* For each $i \in \mathcal{I}$ and any $\mathbf{v} \in \Omega_{\mathbf{V}}$, consider $u_i^{(1)} \in \Omega_{U_i}^{(1)}$ such that $f_i^{(2)}(\mathbf{v}_{\mathrm{pa}^{(2)}(i)}, h_i(u_i^{(1)})) = f_i^{(1)}(\mathbf{v}_{\mathrm{pa}^{(1)}(i)}, u_i^{(1)})$. Then,

$$
\begin{aligned}
f_{i[\mathbf{x}_*]}^{(2)}(\mathbf{v}_{\mathrm{pa}^{(2)}(i)}, h_i(u_i^{(1)})) &= (f_{i[\mathbf{x}_j]}(\mathbf{v}_{\mathrm{pa}^{(2)}(i),j}, h_i(u_i^{(1)})))_{j \in 1:n} && \text{Definition A.3} \\
&= \left( \begin{cases} (\mathbf{x}_j)_i & i \in I_{\mathbf{X}_j} \\ f_i^{(2)}(\mathbf{v}_{\mathrm{pa}^{(2)}(i),j}, h_i(u_i^{(1)})) & i \in \mathcal{I} \setminus I_{\mathbf{X}_j} \end{cases} \right)_{j \in 1:n} && \text{Definition of submodel} \\
&= \left( \begin{cases} (\mathbf{x}_j)_i & i \in I_{\mathbf{X}_j} \\ f_i^{(1)}(\mathbf{v}_{\mathrm{pa}^{(1)}(i),j}, u_i^{(1)}) & i \in \mathcal{I} \setminus I_{\mathbf{X}_j} \end{cases} \right)_{j \in 1:n} && \mathcal{M}^{(1)} \sim_{\mathrm{EI}} \mathcal{M}^{(2)} \\
&= (f_{i[\mathbf{x}_j]}^{(1)}(\mathbf{v}_{\mathrm{pa}^{(1)}(i),j}, u_i^{(1)}))_{j \in 1:n} && \text{Definition of submodel} \\
&= f_{i[\mathbf{x}_*]}^{(1)}(\mathbf{v}_{\mathrm{pa}^{(1)}(i)}, u_i^{(1)}). && \text{Definition A.3}
\end{aligned}
$$

Since $f_i^{(2)}(\mathbf{v}_{\mathrm{pa}^{(2)}(i)}, h_i(u_i^{(1)})) = f_i^{(1)}(\mathbf{v}_{\mathrm{pa}^{(1)}(i)}, u_i^{(1)})$ holds for almost all $u_i^{(1)} \in \Omega_{U_i}^{(1)}$, the proposition is thus established. $\qquad \square$

**Lemma A.10.** *For each $i \in \mathcal{I}$ and for almost all $\mathbf{u}^{(1)} \in \Omega_{\mathbf{U}}^{(1)}$, the AUCCM satisfies*

$$
(\tilde{g}_{i[\mathbf{x}_*]}^{(2)} \circ \mathbf{h}_{pr^*(i)})(\mathbf{u}_{pr^*(i)}^{(1)}) = \tilde{g}_{i[\mathbf{x}_*]}^{(1)}(\mathbf{u}_{pr^*(i)}^{(1)}).
$$

*Proof.* Consider $\mathbf{u}^{(1)} \in \Omega_{\mathbf{U}}^{(1)}$ for which Lemma A.9 holds. We proceed by induction based on the common causal order $\leq$ (hence the necessity of the causal order).

When $|\operatorname{pr}(i)| = 0$, we have $\mathrm{pr}^*(i) = \{i\}$. By Lemma A.9:

$$
\begin{aligned}
(\tilde{g}_{i[\mathbf{x}_*]}^{(2)} \circ \mathbf{h}_{\mathrm{pr}^*(i)})(\mathbf{u}_{\mathrm{pr}^*(i)}^{(1)}) &= \tilde{g}_{i[\mathbf{x}_*]}^{(2)}(h_i(u_i^{(1)})) && \mathrm{pr}^*(i) = \{i\} \\
&= \tilde{f}_{i[\mathbf{x}_*]}^{(2)}(h_i(u_i^{(1)})) && \text{Definition A.6} \\
&= f_{i[\mathbf{x}_*]}^{(2)}(h_i(u_i^{(1)})) && \text{Definition A.4} \\
&= f_{i[\mathbf{x}_*]}^{(1)}(u_i^{(1)}) && \text{Lemma A.9} \\
&= \tilde{f}_{i[\mathbf{x}_*]}^{(1)}(u_i^{(1)}) && \text{Definition A.4} \\
&= \tilde{g}_{i[\mathbf{x}_*]}^{(1)}(u_i^{(1)}) && \text{Definition A.6} \\
&= \tilde{g}_{i[\mathbf{x}_*]}^{(1)}(\mathbf{u}_{\mathrm{pr}^*(i)}^{(1)}). && \mathrm{pr}^*(i) = \{i\}
\end{aligned}
$$

When $|\operatorname{pr}(i)| > 0$, assume that for each $k \in \operatorname{pr}(i)$, $(\tilde{g}_{k[\mathbf{x}_*]}^{(2)} \circ \mathbf{h}_{\mathrm{pr}^*(k)})(\mathbf{u}_{\mathrm{pr}^*(k)}^{(1)}) = \tilde{g}_{k[\mathbf{x}_*]}^{(1)}(\mathbf{u}_{\mathrm{pr}^*(k)}^{(1)})$ holds. Then,

$$
\begin{aligned}
\tilde{g}_{i[\mathbf{x}_*]}^{(2)}(\mathbf{h}_{\mathrm{pr}^*(i)}(\mathbf{u}_{\mathrm{pr}^*(i)}^{(1)})) &= \tilde{f}_{i[\mathbf{x}_*]}^{(2)}(\mathbf{h}_{\mathrm{an}^{*(2)}(i)}(\mathbf{u}_{\mathrm{an}^{*(2)}(i)}^{(1)})) && \text{Definition A.6} \\
&= f_{i[\mathbf{x}_*]}^{(2)}((\tilde{f}_{k[\mathbf{x}_*]}^{(2)}(\mathbf{h}_{\mathrm{an}^{*(2)}(k)}(\mathbf{u}_{\mathrm{an}^{*(2)}(k)}^{(1)})))_{k \in \mathrm{pa}^{(2)}(i)}, h_i(u_i^{(1)})) && \text{Definition A.4} \\
&= f_{i[\mathbf{x}_*]}^{(2)}((\tilde{g}_{k[\mathbf{x}_*]}^{(2)}(\mathbf{h}_{\mathrm{pr}^*(k)}(\mathbf{u}_{\mathrm{pr}^*(k)}^{(1)})))_{k \in \mathrm{pa}^{(2)}(i)}, h_i(u_i^{(1)})) && \text{Definition A.6} \\
&= f_{i[\mathbf{x}_*]}^{(2)}((\tilde{g}_{k[\mathbf{x}_*]}^{(1)}(\mathbf{u}_{\mathrm{pr}^*(k)}^{(1)}))_{k \in \mathrm{pa}^{(2)}(i)}, h_i(u_i^{(1)})) && \text{Induction} \\
&= f_{i[\mathbf{x}_*]}^{(1)}((\tilde{g}_{k[\mathbf{x}_*]}^{(1)}(\mathbf{u}_{\mathrm{pr}^*(k)}^{(1)}))_{k \in \mathrm{pa}^{(1)}(i)}, u_i^{(1)}) && \text{Lemma A.9} \\
&= f_{i[\mathbf{x}_*]}^{(1)}((\tilde{f}_{k[\mathbf{x}_*]}^{(1)}(\mathbf{u}_{\mathrm{an}^{*(1)}(k)}^{(1)}))_{k \in \mathrm{pa}^{(1)}(i)}, u_i^{(1)}) && \text{Definition A.6} \\
&= \tilde{f}_{i[\mathbf{x}_*]}^{(1)}(\mathbf{u}_{\mathrm{an}^{*(1)}(i)}^{(1)}) && \text{Definition A.4} \\
&= \tilde{g}_{i[\mathbf{x}_*]}^{(1)}(\mathbf{u}_{\mathrm{pr}^*(i)}^{(1)}). && \text{Definition A.6}
\end{aligned}
$$

Since Lemma A.9 holds for almost all $\mathbf{u}^{(1)} \in \Omega_{\mathbf{U}}^{(1)}$, the proposition is thus established. $\qquad \square$

**Lemma A.11.** *For each $i \in \mathcal{I}$ and for almost all $\mathbf{u}^{(1)} \in \Omega_{\mathbf{U}}^{(1)}$, the PCCM satisfies*

$$(\tilde{\mathbf{g}}_{pr^*(i)[\mathbf{x}_*]}^{(2)} \circ \mathbf{h}_{pr^*(i)})(\mathbf{u}_{pr^*(i)}^{(1)}) = \tilde{\mathbf{g}}_{pr^*(i)[\mathbf{x}_*]}^{(1)}(\mathbf{u}_{pr^*(i)}^{(1)}).$$

*Proof.* Consider $\mathbf{u}^{(1)} \in \Omega_{\mathbf{U}}^{(1)}$ for which Lemma A.10 holds. Then,

$$\tilde{\mathbf{g}}_{pr^*(i)[\mathbf{x}_*]}^{(2)}(\mathbf{h}_{pr^*(i)}(\mathbf{u}_{pr^*(i)}^{(1)})) = (\tilde{g}_{k[\mathbf{x}_*]}^{(2)}(\mathbf{h}_{pr^*(i)}(\mathbf{u}_{pr^*(i)}^{(1)})))_{k \in pr^*(i)} \qquad \text{Definition A.7}$$

$$= (\tilde{g}_{k[\mathbf{x}_*]}^{(1)}(\mathbf{u}_{pr^*(i)}^{(1)}))_{k \in pr^*(i)} \qquad \text{Lemma A.10}$$

$$= \tilde{\mathbf{g}}_{pr^*(i)[\mathbf{x}_*]}^{(1)}(\mathbf{u}_{pr^*(i)}^{(1)}). \qquad \text{Definition A.7}$$

Since Lemma A.10 holds for almost all $\mathbf{u}^{(1)} \in \Omega_{\mathbf{U}}^{(1)}$, the proposition is thus established. $\qquad \square$

**Theorem 3.2** ($\sim_{\mathrm{EI}}$ Implies $\sim_{\mathcal{L}_3}$). *For recursive SCMs $\mathcal{M}^{(1)}$ and $\mathcal{M}^{(2)}$, if $\mathcal{M}^{(1)} \sim_{\mathrm{EI}} \mathcal{M}^{(2)}$, then $\mathcal{M}^{(1)} \sim_{\mathcal{L}_3} \mathcal{M}^{(2)}$.*

*Proof.* Consider $\mathbf{u}^{(1)} \in \Omega_{\mathbf{U}}^{(1)}$ for which Lemma A.8 and Lemma A.11 hold simultaneously. Then, when $pr^*(i) = \mathcal{I}$, we have

$$\mathbf{V}_{\mathcal{M}_*}^{(1)}(\mathbf{u}^{(1)}) \xup020 \overset{\text{Lemma A.8}}{=\!=\!=\!=\!=} \tilde{\mathbf{g}}_{\mathcal{I}[\mathbf{x}_*]}^{(1)}(\mathbf{u}^{(1)}) \overset{\text{Lemma A.11}}{=\!=\!=\!=\!=} (\tilde{\mathbf{g}}_{\mathcal{I}[\mathbf{x}_*]}^{(2)} \circ \mathbf{h})(\mathbf{u}^{(1)}) \overset{\text{Lemma A.8}}{=\!=\!=\!=\!=} (\mathbf{V}_{\mathcal{M}_*}^{(2)} \circ \mathbf{h})(\mathbf{u}^{(1)}).$$

Since Lemmas Lemma A.8 and Lemma A.11 hold for almost all $\mathbf{u}^{(1)} \in \Omega_{\mathbf{U}}^{(1)}$, it follows that $\mathbf{V}_{\mathcal{M}_*}^{(1)}$ almost surely equals $\mathbf{V}_{\mathcal{M}_*}^{(2)} \circ \mathbf{h}$. Moreover, since $P_{\mathbf{U}}^{(2)} = \mathbf{h}_{\sharp} P_{\mathbf{U}}^{(1)}$, it follows that

$$P_{\mathbf{V}_*}^{\mathcal{M}^{(1)}} = (\mathbf{V}_{\mathcal{M}_*}^{(1)})_{\sharp} P_{\mathbf{U}}^{(1)} = (\mathbf{V}_{\mathcal{M}_*}^{(2)} \circ \mathbf{h})_{\sharp} P_{\mathbf{U}}^{(1)} = (\mathbf{V}_{\mathcal{M}_*}^{(2)})_{\sharp}(\mathbf{h}_{\sharp} P_{\mathbf{U}}^{(1)}) = (\mathbf{V}_{\mathcal{M}_*}^{(2)})_{\sharp} P_{\mathbf{U}}^{(2)} = P_{\mathbf{V}_*}^{\mathcal{M}^{(2)}}.$$

Therefore, for any counterfactual random variables $\mathbf{Y}_* \subseteq \mathbf{V}_*$ and any event $\mathcal{Y}_*$, we have $P_{\mathbf{Y}_*}^{\mathcal{M}^{(1)}}(\mathcal{Y}_*) = P_{\mathbf{Y}_*}^{\mathcal{M}^{(2)}}(\mathcal{Y}_*)$, as they are obtained by marginalizing $P_{\mathbf{V}_*}^{\mathcal{M}^{(1)}} = P_{\mathbf{V}_*}^{\mathcal{M}^{(2)}}$. According to the $\mathcal{L}_3$-valuation, any term of the form $\mathbb{P}(\mathbf{Y}_* \in \mathcal{Y}_*)$ has equal assignments in $\mathcal{M}^{(1)}$ and $\mathcal{M}^{(2)}$. Since for any $\varphi \in \mathcal{L}_3$, $\varphi$ consists of terms of the form $\mathbb{P}(\mathbf{Y}_* \in \mathcal{Y}_*)$, it follows that $\varphi(\mathcal{M}^{(1)}) = \varphi(\mathcal{M}^{(2)})$. Therefore, according to the definition of $\mathcal{L}_3$-theory, we have

$$\mathcal{L}_3(\mathcal{M}^{(1)}) = \{\varphi(\mathcal{M}^{(1)}) \mid \varphi \in \mathcal{L}_3\} = \{\varphi(\mathcal{M}^{(2)}) \mid \varphi \in \mathcal{L}_3\} = \mathcal{L}_3(\mathcal{M}^{(2)}),$$

i.e., $\mathcal{M}^{(1)} \sim_{\mathcal{L}_3} \mathcal{M}^{(2)}$, according to Definition 2.2. $\qquad \square$

## A.3. Bijective SCM

From the perspective of measure theory, a (regular) conditional distribution is defined as a probability kernel $P_{\mathbf{Y}|\mathbf{X}} : \mathcal{F}_{\mathbf{Y}} \times \Omega_{\mathbf{X}} \to [0, 1]$, where it is required that the index sets satisfy $I_{\mathbf{X}} \cap I_{\mathbf{Y}} = \emptyset$. If for any $\mathcal{X} \in \mathcal{F}_{\mathbf{X}}$ and $\mathcal{Y} \in \mathcal{F}_{\mathbf{Y}}$ there exists a joint distribution $P_{\mathbf{X},\mathbf{Y}}(\mathcal{X} \times \mathcal{Y}) = \int_{\mathcal{X}} P_{\mathbf{Y}|\mathbf{X}}(\mathcal{Y}, \mathbf{x}) P_{\mathbf{X}}(\mathrm{d}\mathbf{x})$, where $P_{\mathbf{X}}$ is the marginal distribution, then such a probability kernel $P_{\mathbf{Y}|\mathbf{X}}$ is referred to as a regular conditional distribution given $\mathbf{X}$. The Decomposition Theorem establishes its existence and uniqueness:

**Theorem A.12** (Decomposition Theorem). *If the space $\Omega_{\mathbf{X}}$ is a Polish space, then the regular conditional distribution $P_{\mathbf{Y}|\mathbf{X}} : \mathcal{F}_{\mathbf{Y}} \times \Omega_{\mathbf{X}} \to [0, 1]$ exists and is almost surely unique.*

**Lemma A.13.** *If $\mathbf{Y} = f(\mathbf{X})$, then for all $\mathcal{Y} \in \mathcal{F}_{\mathbf{Y}}$ and for almost every $\mathbf{x} \in \Omega_{\mathbf{X}}$, the conditional distribution $P_{\mathbf{Y}|\mathbf{X}}(\mathcal{Y}, \mathbf{x}) = \delta_{f(\mathbf{x})}(\mathcal{Y}) = \mathbb{1}_{\mathcal{Y}}(f(\mathbf{x}))$, where $\delta_{f(\mathbf{x})}(\mathcal{Y})$ is the Dirac measure and $\mathbb{1}_{\mathcal{Y}}(f(\mathbf{x}))$ is the indicator function.*

*Proof.* For any $\mathcal{X} \in \mathcal{F}_{\mathbf{X}}$ and $\mathcal{Y} \in \mathcal{F}_{\mathbf{Y}}$, according to the definition of the joint distribution,

$$\begin{aligned} P_{\mathbf{X},\mathbf{Y}}(\mathcal{X} \times \mathcal{Y}) &= ((\mathbf{X}, \mathbf{Y})_{\sharp} P)(\mathcal{X} \times \mathcal{Y}) \\ &= P(\{\omega \mid \omega \in \mathbf{Y}^{-1}[\mathcal{Y}] \cap \mathbf{X}^{-1}[\mathcal{X}]\}) \\ &= P(\{\omega \mid \omega \in (f(\mathbf{X}))^{-1}[\mathcal{Y}] \cap \mathbf{X}^{-1}[\mathcal{X}]\}) \\ &= P(\{\omega \mid \omega \in \mathbf{X}^{-1}[f^{-1}[\mathcal{Y}]] \cap \mathbf{X}^{-1}[\mathcal{X}]\}) \\ &= P(\{\omega \mid \omega \in \mathbf{X}^{-1}[f^{-1}[\mathcal{Y}] \cap \mathcal{X}]\}) \qquad\qquad f^{-1}[S] \cap f^{-1}[T] = f^{-1}[S \cap T] \\ &= P_{\mathbf{X}}(f^{-1}[\mathcal{Y}] \cap \mathcal{X}). \end{aligned}$$

Then, for the Lebesgue integral of the indicator function (or Dirac measure),

$$\int_{\mathcal{X}} \mathbb{1}_{\mathcal{Y}}(f(\mathbf{x})) P_{\mathbf{X}}(\mathrm{d}\mathbf{x}) = \int_{\mathcal{X}} \mathbb{1}_{f^{-1}[\mathcal{Y}]}(\mathbf{x}) P_{\mathbf{X}}(\mathrm{d}\mathbf{x})$$
$$= \int_{f^{-1}[\mathcal{Y}] \cap \mathcal{X}} P_{\mathbf{X}}(\mathrm{d}\mathbf{x})$$
$$= P_{\mathbf{X}}(f^{-1}[\mathcal{Y}] \cap \mathcal{X})$$
$$= P_{\mathbf{Y},\mathbf{X}}(\mathcal{Y} \times \mathcal{X}).$$

According to Theorem A.12, since $P_{\mathbf{Y}|\mathbf{X}}$ that satisfies $P_{\mathbf{Y},\mathbf{X}}(\mathcal{Y} \times \mathcal{X}) = \int_{\mathcal{X}} P_{\mathbf{Y}|\mathbf{X}}(\mathcal{Y}, \mathbf{x}) P_{\mathbf{X}}(\mathrm{d}\mathbf{x})$ is almost surely unique, it follows that for all $\mathcal{Y} \in \mathcal{F}_{\mathbf{Y}}$ and for almost every $\mathbf{x} \in \Omega_{\mathbf{X}}$, $P_{\mathbf{Y}|\mathbf{X}} = \delta_{f(\mathbf{x})}(\mathcal{Y}) = \mathbb{1}_{\mathcal{Y}}(f(\mathbf{x}))$. $\qquad\square$

**Lemma A.14.** *If* $\mathbf{Y} = f(\mathbf{X}, \mathbf{Z})$, *then for all* $\mathcal{Y} \in \mathcal{F}_{\mathbf{Y}}$ *and for almost every* $\mathbf{z} \in \Omega_{\mathbf{Z}}$, *the conditional distribution* $P_{\mathbf{Y}|\mathbf{Z}}(\mathcal{Y}, \mathbf{z}) = \left( (f(\cdot, \mathbf{z}))_\sharp P_{\mathbf{X}|\mathbf{Z}}(\cdot, \mathbf{z}) \right)(\mathcal{Y})$.

*Proof.* For any $\mathcal{Y} \in \Sigma_{\mathcal{Y}}$, consider the condition that Lemma A.13 holds for $\mathbf{z} \in \Omega_{\mathbf{Z}}$. Then,

$$P_{\mathbf{Y}|\mathbf{Z}}(\mathcal{Y}, \mathbf{z}) = P_{\mathbf{Y},\mathbf{X}|\mathbf{Z}}(\mathcal{Y} \times \Omega_{\mathbf{X}}, \mathbf{z})$$
$$= \int_{\Omega_{\mathbf{X}}} P_{\mathbf{Y}|\mathbf{X},\mathbf{Z}}(\mathcal{Y}, \mathbf{x}, \mathbf{z}) P_{\mathbf{X}|\mathbf{Z}}(\mathrm{d}\mathbf{x}, \mathbf{z}) \qquad \text{Factorization}$$
$$= \int_{\Omega_{\mathbf{X}}} \mathbb{1}_{\mathcal{Y}}(f(\mathbf{x}, \mathbf{z})) P_{\mathbf{X}|\mathbf{Z}}(\mathrm{d}\mathbf{x}, \mathbf{z}) \qquad \text{Lemma A.13}$$
$$= \int_{\Omega_{\mathbf{X}}} \mathbb{1}_{(f(\cdot,\mathbf{z}))^{-1}[\mathcal{Y}]}(\mathbf{x}) P_{\mathbf{X}|\mathbf{Z}}(\mathrm{d}\mathbf{x}, \mathbf{z})$$
$$= \int_{(f(\cdot,\mathbf{z}))^{-1}[\mathcal{Y}]} P_{\mathbf{X}|\mathbf{Z}}(\mathrm{d}\mathbf{x}, \mathbf{z})$$
$$= P_{\mathbf{X}|\mathbf{Z}}\left( (f(\cdot, \mathbf{z}))^{-1}[\mathcal{Y}], \mathbf{z} \right)$$
$$= \left( (f(\cdot, \mathbf{z}))_\sharp P_{\mathbf{X}|\mathbf{Z}}(\cdot, \mathbf{z}) \right)(\mathcal{Y}).$$

Since Lemma A.13 holds for almost every $\mathbf{z} \in \Omega_{\mathbf{Z}}$, the original proposition follows. $\qquad\square$

**Proposition 4.2.** *A recursive SCM* $\mathcal{M}$ *is a BSCM if and only if* $f_i(\mathbf{v}, \cdot)$ *is a bijection for every* $i \in \mathcal{I}$ *and all* $\mathbf{v} \in \Omega_{\mathbf{V}}$.

*Proof.* $\implies$ : Suppose that the SCM is a BSCM. Consider each $i \in \mathcal{I}$. According to Definition 4.1, this is equivalent to proving that $f_i(\mathbf{v}_{\mathrm{pa}(i)}, \cdot)$ is both injective and surjective for all $\mathbf{v} \in \Omega_{\mathbf{V}}$, under the assumption that $\mathbf{\Gamma}$ is a bijection.

First, we show that $f_i(\mathbf{v}_{\mathrm{pa}(i)}, \cdot)$ is injective for all $\mathbf{v} \in \Omega_{\mathbf{V}}$. Given any $\mathbf{v} \in \Omega_{\mathbf{V}}$ and any $u_i, u_i' \in \Omega_{\mathbf{U}}$ such that $f_i(\mathbf{v}_{\mathrm{pa}(i)}, u_i) = f_i(\mathbf{v}_{\mathrm{pa}(i)}, u_i')$, let $v_i^* = f_i(\mathbf{v}_{\mathrm{pa}(i)}, u_i)$ (note that $v_i^*$ may not equal $v_i$). Construct $\mathbf{v}^* = (\mathbf{v}_{\mathrm{pa}(i)}, v_i^*, \mathbf{v}_{\mathcal{I} \setminus \{i\} \setminus \mathrm{pa}(i)}^*)$, where $\mathbf{v}_{\mathcal{I} \setminus \{i\} \setminus \mathrm{pa}(i)}^*$ is arbitrary. By the definition of a BSCM and since $\mathbf{\Gamma}$ is a bijection, there exists a unique $\mathbf{u}^*$ such that $\mathbf{v}^* = \mathbf{\Gamma}(\mathbf{u}^*)$. Replace the $i$-th component of $\mathbf{u}^*$ with $u_i$ and $u_i'$ to obtain $\mathbf{u}$ and $\mathbf{u}'$, respectively. Since $f_i(\mathbf{v}_{\mathrm{pa}(i)}, u_i) = f_i(\mathbf{v}_{\mathrm{pa}(i)}, u_i') = v_i^*$, it follows that $\mathbf{v}^* = \mathbf{\Gamma}(\mathbf{u}) = \mathbf{\Gamma}(\mathbf{u}')$. Given the uniqueness of $\mathbf{u}^*$ such that $\mathbf{v}^* = \mathbf{\Gamma}(\mathbf{u}^*)$, we have $\mathbf{u} = \mathbf{u}' = \mathbf{u}^*$, implying that $u_i = u_i'$. Therefore, for any $u_i, u_i' \in \Omega_{\mathbf{U}}$, if $f_i(\mathbf{v}_{\mathrm{pa}(i)}, u_i) = f_i(\mathbf{v}_{\mathrm{pa}(i)}, u_i')$, then $u_i = u_i'$, proving that $f_i(\mathbf{v}_{\mathrm{pa}(i)}, \cdot)$ is injective.

Next, we show that $f_i(\mathbf{v}_{\mathrm{pa}(i)}, \cdot)$ is surjective for all $\mathbf{v} \in \Omega_{\mathbf{V}}$. Assume, for contradiction, that there exists $\mathbf{v} \in \Omega_{\mathbf{V}}$ such that $f_i(\mathbf{v}_{\mathrm{pa}(i)}, \cdot)$ is not surjective. Then, there exists $v_i'$ such that for all $u_i \in \Omega_{U_i}$, $f_i(\mathbf{v}_{\mathrm{pa}(i)}, u_i) \neq v_i'$. Construct $\mathbf{v}^* = (\mathbf{v}_{\mathrm{pa}(i)}, v_i', \mathbf{v}_{\mathcal{I} \setminus \{i\} \setminus \mathrm{pa}(i)}^*)$, where $\mathbf{v}_{\mathcal{I} \setminus \{i\} \setminus \mathrm{pa}(i)}^*$ is arbitrary. By the definition of a BSCM and since $\mathbf{\Gamma}$ is a bijection, there exists $\mathbf{u}^*$ such that $\mathbf{v}^* = \mathbf{\Gamma}(\mathbf{u}^*)$, where the $i$-th component satisfies $f_i(\mathbf{v}_{\mathrm{pa}(i)}, u_i^*) = v_i'$. This contradicts the assumption that for all $u_i \in \Omega_{U_i}$, $f_i(\mathbf{v}_{\mathrm{pa}(i)}, u_i) \neq v_i'$. Therefore, our assumption is false, and $f_i(\mathbf{v}_{\mathrm{pa}(i)}, \cdot)$ must be surjective for all $\mathbf{v} \in \Omega_{\mathbf{V}}$.

In summary, for all $\mathbf{v} \in \Omega_{\mathbf{V}}$, $f_i(\mathbf{v}_{\mathrm{pa}(i)}, \cdot)$ is a bijection.

$\impliedby$ : Suppose that $f_i(\mathbf{v}_{\mathrm{pa}(i)}, \cdot)$ is a bijection for each $i \in \mathcal{I}$ and for all $\mathbf{v} \in \Omega_{\mathbf{V}}$. This is equivalent to proving that $\mathbf{\Gamma}$ is both injective and surjective.

First, we show that $\mathbf{\Gamma}$ is injective. Let $\mathbf{u}, \mathbf{u}' \in \Omega_{\mathbf{U}}$ such that $\mathbf{v} = \mathbf{\Gamma}(\mathbf{u}) = \mathbf{\Gamma}(\mathbf{u}')$. According to Lemma Lemma A.2, for each $i \in \mathcal{I}$, we have $v_i = f_i(\mathbf{v}_{\mathrm{pa}(i)}, u_i) = f_i(\mathbf{v}_{\mathrm{pa}(i)}, u_i')$. Since $f_i(\mathbf{v}_{\mathrm{pa}(i)}, \cdot)$ is a bijection, it follows that $u_i = u_i'$. Therefore, $\mathbf{u} = \mathbf{u}'$, proving that $\mathbf{\Gamma}$ is injective.

Next, we show that $\mathbf{\Gamma}$ is surjective. For each $\mathbf{v} \in \Omega_{\mathbf{V}}$ and each $i \in \mathcal{I}$, consider the $i$-th component $v_i$. Since $f_i(\mathbf{v}_{\mathrm{pa}(i)}, \cdot)$ is a bijection for any $\mathbf{v} \in \Omega_{\mathbf{V}}$, there exists $u_i = (f_i(\mathbf{v}_{\mathrm{pa}(i)}, \cdot))^{-1}(v_i)$ such that $v_i = f_i(\mathbf{v}_{\mathrm{pa}(i)}, u_i)$. Let $\mathbf{u} = ((f_i(\mathbf{v}_{\mathrm{pa}(i)}, \cdot))^{-1}(v_i))_{i \in \mathcal{I}}$. According to the definition of the solution function $\mathbf{\Gamma}$, substituting $\mathbf{u}$ into $\mathbf{\Gamma}$ yields $\mathbf{\Gamma}(\mathbf{u}) = \mathbf{v}$. Thus, for any $\mathbf{v} \in \Omega_{\mathbf{V}}$, there exists $\mathbf{u}$ such that $\mathbf{\Gamma}(\mathbf{u}) = \mathbf{v}$, proving that $\mathbf{\Gamma}$ is surjective.

Therefore, $\mathbf{\Gamma}$ is a bijection. By Definition 4.1, $\mathcal{M}$ is a BSCM. $\qquad\square$

**Theorem 4.3** (BSCM-EI). *If two BSCMs $\mathcal{M}^{(1)}$ and $\mathcal{M}^{(2)}$ share a common causal order $\leq$ and the same observational distribution $P_{\mathbf{V}}$, then $\mathcal{M}^{(1)} \sim_{\mathrm{EI}} \mathcal{M}^{(2)}$ if and only if for every $i \in \mathcal{I}$, there exists a bijection $h_i : \Omega_{U_i}^{(1)} \to \Omega_{U_i}^{(2)}$ such that for all $\mathbf{v} \in \Omega_{\mathbf{V}}$,*

$$(f_i^{(2)}(\mathbf{v}, \cdot))^{-1} \circ (f_i^{(1)}(\mathbf{v}, \cdot)) = h_i$$

*almost surely.*

*Proof.* $\implies$ : Suppose that $\mathcal{M}^{(1)} \sim_{\mathrm{EI}} \mathcal{M}^{(2)}$. According to the definition of exogenous isomorphism, there exists a bijection $\mathbf{h} = (h_i)_{i \in \mathcal{I}}$ such that each $h_i$ is a bijection and the causal mechanisms are preserved. For each $i \in \mathcal{I}$ and given any $\mathbf{v} \in \Omega_{\mathbf{V}}$, consider the causal mechanism $f_i^{(2)}(\mathbf{v}, h_i(u_i^{(1)})) = f_i^{(1)}(\mathbf{v}, u_i^{(1)})$ for some $u_i^{(1)} \in \Omega_{U_i}^{(1)}$. Since $f_i^{(2)}(\mathbf{v}, \cdot)$ is a bijection, composing both sides with $(f_i^{(2)}(\mathbf{v}, \cdot))^{-1}$ yields

$$h_i(u_i^{(1)}) = (f_i^{(2)}(\mathbf{v}, u_i^{(1)}))^{-1} \circ (f_i^{(1)}(\mathbf{v}, \cdot)).$$

Since $f_i^{(2)}(\mathbf{v}, h_i(u_i^{(1)})) = f_i^{(1)}(\mathbf{v}, u_i^{(1)})$ holds for almost every $u_i^{(1)} \in \Omega_{U_i}^{(1)}$, it follows that

$$(f_i^{(2)}(\mathbf{v}, \cdot))^{-1} \circ f_i^{(1)}(\mathbf{v}, \cdot) = h_i$$

almost surely.

$\impliedby$ : For each $i \in \mathcal{I}$, suppose there exists a bijection $h_i : \Omega_{U_i}^{(1)} \to \Omega_{U_i}^{(2)}$ such that

$$(f_i^{(2)}(\mathbf{v}, \cdot))^{-1} \circ f_i^{(1)}(\mathbf{v}, \cdot) = h_i$$

almost surely. By the associativity of function composition, for any $\mathbf{v} \in \Omega_{\mathbf{V}}$, we have

$$f_i^{(2)}(\mathbf{v}, \cdot) \circ h_i = f_i^{(2)}(\mathbf{v}, \cdot) \circ \left( (f_i^{(2)}(\mathbf{v}, \cdot))^{-1} \circ f_i^{(1)}(\mathbf{v}, \cdot) \right) = f_i^{(1)}(\mathbf{v}, \cdot)$$

almost surely. This equality holds for each $i \in \mathcal{I}$ and all $\mathbf{v} \in \Omega_{\mathbf{V}}$, thereby preserving the causal mechanisms.

Moreover, since the observational distribution satisfies

$$P_{\mathbf{V}} = (\mathbf{\Gamma}^{(1)})_{\sharp} P_{\mathbf{U}}^{(1)} = (\mathbf{\Gamma}^{(2)})_{\sharp} P_{\mathbf{U}}^{(2)},$$

and both $\mathbf{\Gamma}^{(1)}$ and $\mathbf{\Gamma}^{(2)}$ are bijections, it follows by pullback that

$$((\mathbf{\Gamma}^{(2)})^{-1} \circ \mathbf{\Gamma}^{(1)})_{\sharp} P_{\mathbf{U}}^{(1)} = P_{\mathbf{U}}^{(2)}.$$

By Lemma A.8 and Lemma A.11, we have $\mathbf{\Gamma}^{(1)} = \mathbf{\Gamma}^{(2)} \circ \mathbf{h}$, that is, $(\mathbf{\Gamma}^{(2)})^{-1} \circ \mathbf{\Gamma}^{(1)} = \mathbf{h}$. Therefore, $\mathbf{h}_{\sharp} P_{\mathbf{U}}^{(1)} = P_{\mathbf{U}}^{(2)}$, which preserves the exogenous distributions. Lastly, the component-wise bijections are satisfied by the assumption of $h_i : \Omega_{U_i}^{(1)} \to \Omega_{U_i}^{(2)}$. Hence, $\mathbf{h} = (h_i)_{i \in \mathcal{I}}$ constitutes an exogenous isomorphism. Given the existence of such an exogenous isomorphism and the common causal order, it follows that $\mathcal{M}^{(1)} \sim_{\mathrm{EI}} \mathcal{M}^{(2)}$. $\qquad\square$

Before proceeding with the proof below, we visually depict the objects of study in Theorem 4.3 and Definition 4.4 in Figure 3, allowing readers to intuitively grasp the strong connection between these two concepts. Intuitively, these objects are related through a "flipping" operation; formally, this relationship is characterized algebraically by the computation of associativity and inverses. Subsequently, we focus on Definition 4.4, which enables the problem to be confined within the same BSCM.

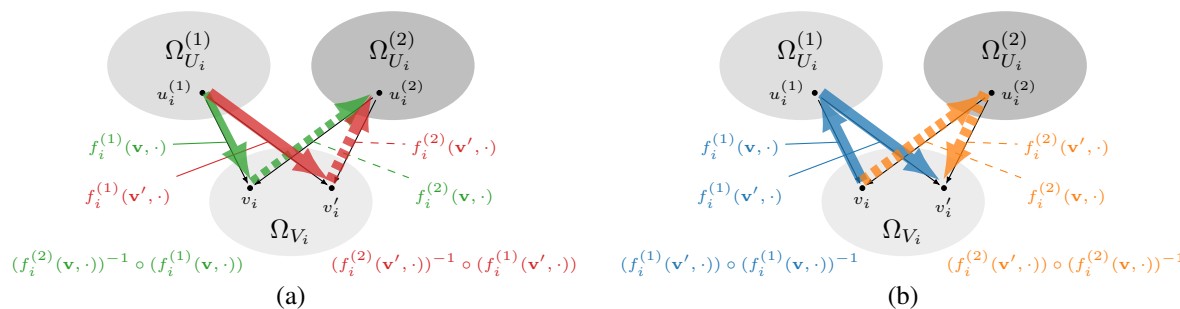

*Figure 3.* (a) The objects of study in Theorem 4.3, $(f_i^{(2)}(\mathbf{v},\cdot))^{-1}\circ(f_i^{(1)}(\mathbf{v},\cdot))$ (green) and $(f_i^{(2)}(\mathbf{v}',\cdot))^{-1}\circ(f_i^{(1)}(\mathbf{v}',\cdot))$ (red), constructed across different BSCMs; (b) The objects of study in Definition 4.4, $(f_i^{(1)}(\mathbf{v}',\cdot))\circ(f_i^{(1)}(\mathbf{v},\cdot))^{-1}$ (blue) and $(f_i^{(2)}(\mathbf{v}',\cdot))\circ(f_i^{(2)}(\mathbf{v},\cdot))^{-1}$ (yellow), constructed within the same BSCM.

**Proposition 4.5.** *If the BSCM $\mathcal{M}$ is Markovian, then for almost all $\mathbf{v},\mathbf{v}'\in\Omega_{\mathbf{V}}$, the conditional distributions satisfy $P_{V_i|\mathbf{V}_{pa(i)}}(\cdot,\mathbf{v}') = (K_{\mathcal{M},i}(\cdot,\mathbf{v},\mathbf{v}'))_\sharp P_{V_i|\mathbf{V}_{pa(i)}}(\cdot,\mathbf{v}).$*

*Proof.* According to the recursive SCM, there exists a unique solution such that $V_i = f_i(\mathbf{V}_{pa(i)}, U_i)$. Based on Proposition 4.2, the causal mechanism $f_i(\mathbf{v}_{pa(i)},\cdot)$ in the BSCM is a bijection for each $i\in\mathcal{I}$ and for all $\mathbf{v}\in\Omega_{\mathbf{V}}$. Therefore, we have $U_i = (f_i(\mathbf{V}_{pa(i)},\cdot))^{-1}(V_i)$.

According to Lemma A.14, for any $\mathbf{v}\in\Omega_{\mathbf{V}}$, it holds that

$$P_{V_i|\mathbf{V}_{pa(i)}}(\cdot,\mathbf{v}_{pa(i)}) = (f_i(\mathbf{v}_{pa(i)},\cdot))_\sharp P_{U_i|\mathbf{V}_{pa(i)}}(\cdot,\mathbf{v}_{pa(i)}),$$
$$P_{U_i|\mathbf{V}_{pa(i)}}(\cdot,\mathbf{v}_{pa(i)}) = \left((f_i(\mathbf{v}_{pa(i)},\cdot))^{-1}\right)_\sharp P_{V_i|\mathbf{V}_{pa(i)}}(\cdot,\mathbf{v}_{pa(i)}).$$

By the Markovianity assumption, $P_{U_i|\mathbf{V}_{pa(i)}} = P_{U_i}$. Therefore, for any $\mathbf{v},\mathbf{v}'\in\Omega_{\mathbf{V}}$, the counterfactual transport satisfies

$$
\begin{aligned}
(K_{\mathcal{M},i}(\cdot,\mathbf{v},\mathbf{v}'))_\sharp P_{V_i|\mathbf{V}_{pa(i)}}(\cdot,\mathbf{v}_{pa(i)}) &= \left((f_i(\mathbf{v}'_{pa(i)},\cdot))\circ(f_i(\mathbf{v}_{pa(i)},\cdot))^{-1}\right)_\sharp P_{V_i|\mathbf{V}_{pa(i)}}(\cdot,\mathbf{v}_{pa(i)})\\
&= \left(f_i(\mathbf{v}'_{pa(i)},\cdot)\right)_\sharp \left(\left((f_i(\mathbf{v}_{pa(i)},\cdot))^{-1}\right)_\sharp P_{V_i|\mathbf{V}_{pa(i)}}(\cdot,\mathbf{v}_{pa(i)})\right)\\
&= \left(f_i(\mathbf{v}'_{pa(i)},\cdot)\right)_\sharp P_{U_i|\mathbf{V}_{pa(i)}}(\cdot,\mathbf{v}_{pa(i)}) && \text{Lemma A.14}\\
&= \left(f_i(\mathbf{v}'_{pa(i)},\cdot)\right)_\sharp P_{U_i} && \text{Markovianity}\\
&= \left(f_i(\mathbf{v}'_{pa(i)},\cdot)\right)_\sharp P_{U_i|\mathbf{V}_{pa(i)}}(\cdot,\mathbf{v}'_{pa(i)}) && \text{Markovianity}\\
&= P_{V_i|\mathbf{V}_{pa(i)}}(\cdot,\mathbf{v}'_{pa(i)}). && \text{Lemma A.14}
\end{aligned}
$$

This shows that the conditional distribution satisfies $P_{V_i|\mathbf{V}_{pa(i)}}(\cdot,\mathbf{v}') = (K_{\mathcal{M},i}(\cdot,\mathbf{v},\mathbf{v}'))_\sharp P_{V_i|\mathbf{V}_{pa(i)}}(\cdot,\mathbf{v}).$ $\square$

**Theorem 4.6** (EI-ID from Counterfactual Transport). *An SCM is $\sim_{\text{EI}}$-identifiable from $\mathcal{A}_{\{BSCM,\leq,P_{\mathbf{V}},\mathbf{K}\}}$.*

*Proof.* Consider any pair of models $\mathcal{M}^{(1)},\mathcal{M}^{(2)}\models\mathcal{A}_{\{BSCM,\leq,P_{\mathbf{V}},\mathbf{K}\}}$ and each $i\in\mathcal{I}$. For any pair $\mathbf{v},\mathbf{v}'\in\Omega_{\mathbf{V}}$, let the components of the counterfactual transport be $K_{\mathcal{M}^{(1)},i}(\cdot,\mathbf{v},\mathbf{v}')$ and $K_{\mathcal{M}^{(2)},i}(\cdot,\mathbf{v},\mathbf{v}')$, respectively. According to the validity of $\mathcal{A}_{\mathbf{K}}(\mathcal{M})$, it holds that

$$K_{\mathcal{M}^{(1)},i}(\cdot,\mathbf{v},\mathbf{v}') = K_{\mathcal{M}^{(2)},i}(\cdot,\mathbf{v},\mathbf{v}') = K_i(\cdot,\mathbf{v},\mathbf{v}')$$

almost surely. By the definition of counterfactual transport, this implies

$$(f_i^{(1)}(\mathbf{v}'_{pa(i)},\cdot))\circ(f_i^{(1)}(\mathbf{v}_{pa(i)},\cdot))^{-1} = (f_i^{(2)}(\mathbf{v}'_{pa(i)},\cdot))\circ(f_i^{(2)}(\mathbf{v}_{pa(i)},\cdot))^{-1}$$

almost surely. Since the causal mechanisms $f_i^{(1)}(\mathbf{v}_{\mathrm{pa}(i)}, \cdot)$ and $f_i^{(2)}(\mathbf{v}'_{\mathrm{pa}(i)}, \cdot)$ are bijections, we can compose them on the right with $f_i^{(1)}(\mathbf{v}_{\mathrm{pa}(i)}, \cdot)$ and on the left with $(f_i^{(2)}(\mathbf{v}'_{\mathrm{pa}(i)}, \cdot))^{-1}$, respectively. By the associativity of function composition, we obtain

$$(f_i^{(2)}(\mathbf{v}'_{\mathrm{pa}(i)}, \cdot))^{-1} \circ f_i^{(1)}(\mathbf{v}'_{\mathrm{pa}(i)}, \cdot) = (f_i^{(2)}(\mathbf{v}_{\mathrm{pa}(i)}, \cdot))^{-1} \circ f_i^{(1)}(\mathbf{v}_{\mathrm{pa}(i)}, \cdot)$$

almost surely, and both sides of the equation remain bijections. Since this equality holds for any pair $\mathbf{v}, \mathbf{v}' \in \Omega_{\mathbf{V}}$, they are all equivalent to some bijection $h_i : \Omega_{U_i}^{(1)} \to \Omega_{U_i}^{(2)}$. That is, for each $i \in \mathcal{I}$, there exists a bijection $h_i : \Omega_{U_i}^{(1)} \to \Omega_{U_i}^{(2)}$ such that for all $\mathbf{v} \in \Omega_{\mathbf{V}}$,

$$(f_i^{(2)}(\mathbf{v}, \cdot))^{-1} \circ f_i^{(1)}(\mathbf{v}, \cdot) = h_i$$

almost surely. Furthermore, a common causal order exists by assumption. Therefore, by Theorem 4.3, it follows that $\mathcal{M}^{(1)} \sim_{\mathrm{IE}} \mathcal{M}^{(2)}$. $\qquad\square$

**Theorem 4.8** (EI-ID from KR Transport). *An SCM is $\sim_{\mathrm{EI}}$-identifiable from $\mathcal{A}_{\{BSCM,\leq,M,P_{\mathbf{V}},KR\}}$.*

*Proof.* We need to prove that if $\mathcal{M} \models \mathcal{A}_{\{BSCM,\leq,M,P_{\mathbf{V}},KR\}}$, then $\mathcal{M} \models \mathcal{A}_{\{BSCM,\leq,P_{\mathbf{V}},\mathbf{K}\}}$, and subsequently apply Theorem 4.6. Specifically, we only need to demonstrate that if $\mathcal{M} \models \mathcal{A}_{\{BSCM,\leq,M,P_{\mathbf{V}},KR\}}$, then $\mathcal{A}_{\mathbf{K}}(\mathcal{M})$ holds.

Assume that $\mathcal{M} \models \mathcal{A}_{\{BSCM,\leq,M,P_{\mathbf{V}},KR\}}$. Then, $\mathcal{M}$ is a Markov BSCM with causal order $\leq$ and observational distribution $P_{\mathbf{V}}$. For each $i \in \mathcal{I}$, consider any pair $\mathbf{v}, \mathbf{v}' \in \Omega_{\mathbf{V}}$. Given the observational distribution $P_{\mathbf{V}}$, the conditional distribution $P_{V_i \mid \mathbf{V}_{\mathrm{pa}(i)}}(\cdot, \mathbf{v}_{\mathrm{pa}(i)})$ is almost surely uniquely determined by Theorem A.12. Then, according to Lemma 4.7, the KR transport between $P_{V_i \mid \mathbf{V}_{\mathrm{pa}(i)}}(\cdot, \mathbf{v}_{\mathrm{pa}(i)})$ and $P_{V_i \mid \mathbf{V}_{\mathrm{pa}(i)}}(\cdot, \mathbf{v}'_{\mathrm{pa}(i)})$ is almost surely unique. Let the partial function $K_i(\cdot, \mathbf{v}, \mathbf{v}')$ be any version of these KR transports. Therefore, $K_i(\cdot, \mathbf{v}, \mathbf{v}')$ is almost uniquely determined, and hence $\mathbf{K} = (K_i)_{i \in \mathcal{I}}$ is also almost uniquely determined.

Since $\mathcal{A}_{\mathrm{KR}}(\mathcal{M})$ holds, the components of the counterfactual transport $K_{\mathcal{M},i}$ are almost surely equivalent to the KR transports. Moreover, because $K_{\mathcal{M},i}$ is a transport between conditional distributions by Proposition 4.5, and by Lemma 4.7, the KR transport between two distributions is almost surely unique given the distributions. Therefore, the component $K_{\mathcal{M},i}(\cdot, \mathbf{v}, \mathbf{v}')$ of the counterfactual transport is almost surely equivalent to $K_i(\cdot, \mathbf{v}, \mathbf{v}')$. Since the counterfactual transport $\mathbf{K}_{\mathcal{M}} = (K_i)_{i \in \mathcal{I}}$, it follows that the counterfactual transport $\mathbf{K}_{\mathcal{M}} = \mathbf{K}$ holds almost surely. Thus, $\mathcal{A}_{\mathbf{K}}(\mathcal{M})$ is satisfied.

Consequently, we have proven that if $\mathcal{M} \models \mathcal{A}_{\{BSCM,\leq,M,P_{\mathbf{V}},KR\}}$, then $\mathcal{M} \models \mathcal{A}_{\{BSCM,\leq,P_{\mathbf{V}},\mathbf{K}\}}$. By Theorem 4.6, for any pair $\mathcal{M}^{(1)}, \mathcal{M}^{(2)} \models \mathcal{A}_{\{BSCM,\leq,P_{\mathbf{V}},\mathbf{K}\}}$, it holds that $\mathcal{M}^{(1)} \sim_{\mathrm{EI}} \mathcal{M}^{(2)}$. Therefore, for any pair $\mathcal{M}^{(1)}, \mathcal{M}^{(2)} \models \mathcal{A}_{\{BSCM,\leq,M,P_{\mathbf{V}},KR\}}$, we also have $\mathcal{M}^{(1)} \sim_{\mathrm{EI}} \mathcal{M}^{(2)}$. This implies that the SCM is $\sim_{\mathrm{EI}}$-identifiable from $\mathcal{A}_{\{BSCM,\leq,M,P_{\mathbf{V}},KR\}}$. $\qquad\square$

### A.4. Triangular Monotonic SCM

There are some properties of TM mappings:

**Lemma A.15.** *For a TM mapping $\mathbf{T} : \mathbb{R}^d \to \mathbb{R}^d$, its inverse $\mathbf{T}^{-1}$ is also a TM mapping, and the monotonicity signature satisfies $\xi(\mathbf{T}) = \xi(\mathbf{T}^{-1})$.*

*Proof.* To clarify the proof, we denote the domain or codomain by the symbols $\Omega_{\mathbf{X}}$ and $\Omega_{\mathbf{Z}}$, both of which actually refer to $\mathbb{R}^d$. For a TM mapping $\mathbf{T} : \Omega_{\mathbf{X}} \to \Omega_{\mathbf{Z}}$, assume it is of the form $\mathbf{T}(\mathbf{x}) = (T_j(\mathbf{x}_{1:j-1}, x_j))_{j \in 1:d}$, where $T_j : \Omega_{\mathbf{X}_{1:j-1}} \times \Omega_{X_j} \to \Omega_{Z_j}$. According to the definition of a TM mapping, for any $\mathbf{x}_{1:j-1} \in \Omega_{\mathbf{X}_{1:j-1}}$, the function $T_j(\mathbf{x}_{1:j-1}, \cdot) : \Omega_{X_j} \to \Omega_{Z_j}$ is consistently strictly monotonic. That is, for each $j \in 1:d$, there exists $\xi_j \in \{-1, 1\}$ such that $\xi(T_j) = \xi_j$.

We construct $\widetilde{\mathbf{T}}_{1:j} : \Omega_{\mathbf{Z}_{1:j}} \to \Omega_{\mathbf{X}_{1:j}}$ such that it is of the form $\widetilde{\mathbf{T}}_{1:j} = (\widetilde{T}_k)_{k \in 1:j}$ and

$$\widetilde{T}_j(\mathbf{z}_{1:j}, z_j) = \left(T_j(\widetilde{\mathbf{T}}_{1:j}(\mathbf{z}_{1:j}), \cdot)\right)^{-1}(z_j).$$

for any $j \in 1:d$ and any $\mathbf{z}_{1:j} \in \Omega_{\mathbf{Z}_{1:j}}$. Since $\widetilde{T}_j$ depends only on $\mathbf{z}_{1:j}$, $\widetilde{\mathbf{T}}$ is a triangular mapping.

Let $\widetilde{\mathbf{T}} = \widetilde{\mathbf{T}}_{1:j}$. Next, we prove that $\widetilde{\mathbf{T}} = \mathbf{T}^{-1}$, meaning that $\widetilde{\mathbf{T}}$ is the explicit form of the inverse function of $\mathbf{T}$. According to the definition of an invertible function, we need to show that $\mathbf{z} = \mathbf{T}(\widetilde{\mathbf{T}}(\mathbf{z}))$ for any $\mathbf{z} \in \Omega_{\mathbf{Z}}$. Consider each component,

i.e., prove that $z_j = (\mathbf{T}(\widetilde{\mathbf{T}}(\mathbf{z})))_j$ for any $j \in 1\!:\!d$. First, by the form of $\mathbf{T}$, we can expand its $j$-th component as

$$(\mathbf{T}(\widetilde{\mathbf{T}}(\mathbf{z})))_j = T_j((\widetilde{\mathbf{T}}(\mathbf{z}))_{1:j-1}, (\widetilde{\mathbf{T}}(\mathbf{z}))_j).$$

According to the previous construction, $(\widetilde{\mathbf{T}}(\mathbf{z}))_{1:j-1} = \widetilde{\mathbf{T}}_{1:j-1}(\mathbf{z}_{1:j-1})$ and $(\widetilde{\mathbf{T}}(\mathbf{z}))_j = \widetilde{T}_j(\mathbf{z}_{1:j}, z_j)$. Substituting these into the above equation, we obtain

$$(\mathbf{T}(\widetilde{\mathbf{T}}(\mathbf{z})))_j = T_j \left( \widetilde{\mathbf{T}}_{1:j-1}(\mathbf{z}_{1:j-1}), \left( T_j(\widetilde{\mathbf{T}}_{1:j}(\mathbf{z}_{1:j}), \cdot) \right)^{-1} (z_j) \right).$$

Now, we need to show that the above expression equals $z_j$ for each $j \in 1\!:\!d$. Let $\mathbf{T}_{1:j-1}(\mathbf{z}_{1:j-1}) = \mathbf{x}'_{1:j-1}$. Since $\mathbf{T}$ is a TM mapping, the function $T_j(\mathbf{x}'_{1:j-1}, \cdot) : \Omega_{X_j} \to \Omega_{Z_j}$ is always strictly monotonic and hence invertible. Therefore, the expression simplifies to

$$T_j(\mathbf{x}'_{1:j-1}, \cdot) \circ T_j(\mathbf{x}'_{1:j-1}, \cdot)^{-1}(z_j) = z_j.$$

Thus, $(\mathbf{T}(\widetilde{\mathbf{T}}(\mathbf{z})))_j = z_j$ holds for each $j \in 1\!:\!d$, implying that $\widetilde{\mathbf{T}} = \mathbf{T}^{-1}$. Since $\widetilde{\mathbf{T}}$ is a triangular mapping, $\mathbf{T}^{-1}$ is also a triangular mapping.

Next, we consider the monotonicity signature. For each $j \in 1\!:\!d$, recalling that

$$\widetilde{T}_j(\mathbf{z}_{1:j}, z_j) = \left( T_j(\widetilde{\mathbf{T}}_{1:j}(\mathbf{z}_{1:j}), \cdot) \right)^{-1} (z_j).$$

If for any $\mathbf{x}_{1:j-1} \in \Omega_{\mathbf{X}_{1:j-1}}$, the function $T_j(\mathbf{x}_{1:j-1}, \cdot) : \Omega_{X_j} \to \Omega_{Z_j}$ is s.m.i., then its inverse $T_j^{-1}(\mathbf{x}_{1:j-1}, \cdot) : \Omega_{Z_j} \to \Omega_{X_j}$ is also s.m.i. For any $\mathbf{z}_{1:j-1} \in \Omega_{\mathbf{Z}_{1:j-1}}$, let $\mathbf{T}_{1:j-1}(\mathbf{z}_{1:j-1}) = \mathbf{x}'_{1:j-1}$. Since $\mathbf{x}'_{1:j-1} \in \Omega_{\mathbf{X}_{1:j-1}}$, the function $(T_j(\mathbf{x}'_{1:j-1}, \cdot))^{-1}$ is always s.m.i. Therefore, $\widetilde{T}_j$ is s.m.i. with respect to $z_j$ given any $\mathbf{z}_{1:j-1} \in \Omega_{\mathbf{Z}_{1:j-1}}$. Similarly, when $T_j(\mathbf{x}_{1:j-1}, \cdot)$ is s.m.d., a similar conclusion holds. Therefore, according to the definition of the monotonicity signature, for each $j \in 1\!:\!d$, there exists $\xi_j \in \{-1, 1\}$ such that $\xi(T_j) = \xi(\widetilde{T}_j) = \xi_j$. Consequently,

$$\xi(\mathbf{T}^{-1}) = \xi(\widetilde{\mathbf{T}}) = (\xi(\widetilde{T}_j))_{j \in 1:d} = (\xi_j)_{j \in 1:d} = (\xi(T_j))_{j \in 1:d} = \xi(\mathbf{T}),$$

i.e., the monotonicity signature satisfies $\xi(\mathbf{T}^{-1}) = \xi(\mathbf{T})$.

Moreover, since $\sum_{j=1}^{d} |\xi_j| = d$ and $\mathbf{T}^{-1}$ is a triangular mapping, according to the definition of a TM mapping, $\mathbf{T}^{-1}$ is indeed a TM mapping. $\qquad \square$

**Lemma A.16.** *For two TM mappings $\mathbf{T}^{(1)} : \mathbb{R}^d \to \mathbb{R}^d$ and $\mathbf{T}^{(2)} : \mathbb{R}^d \to \mathbb{R}^d$, their composition $\mathbf{T}^{(1)} \circ \mathbf{T}^{(2)}$ is also a TM mapping, and the monotonicity signature satisfies $\xi(\mathbf{T}^{(1)} \circ \mathbf{T}^{(2)}) = \xi(\mathbf{T}^{(1)}) \odot \xi(\mathbf{T}^{(2)})$, where $\odot$ denotes the component-wise multiplication.*

*Proof.* To clarify the proof, we denote the domains or codomains by the symbols $\Omega_{\mathbf{X}}$, $\Omega_{\mathbf{Y}}$, and $\Omega_{\mathbf{Z}}$, all of which actually refer to $\mathbb{R}^d$. For the TM mapping $\mathbf{T}^{(1)} : \Omega_{\mathbf{Y}} \to \Omega_{\mathbf{Z}}$, assume it is of the form $\mathbf{T}^{(1)}(\mathbf{y}) = (T_j^{(1)}(\mathbf{y}_{1:j-1}, y_j))_{j \in 1:d}$, where $T_j^{(1)} : \Omega_{\mathbf{Y}_{1:j-1}} \times \Omega_{Y_j} \to \Omega_{Z_j}$. According to the definition of a TM mapping, for any $\mathbf{y}_{1:j-1} \in \Omega_{\mathbf{Y}_{1:j-1}}$, the function $T_j^{(1)}(\mathbf{y}_{1:j-1}, \cdot) : \Omega_{Y_j} \to \Omega_{Z_j}$ is consistently strictly monotonic. Therefore, for each $j \in 1\!:\!d$, there exists $\xi_j^{(1)} \in \{-1, 1\}$ such that $\xi(T_j^{(1)}) = \xi_j^{(1)}$. The same holds for the TM mapping $\mathbf{T}^{(2)}$.

We construct $\widetilde{\mathbf{T}}_{1:j} : \Omega_{\mathbf{X}_{1:j}} \to \Omega_{\mathbf{Z}_{1:j}}$ such that it is of the form $\widetilde{\mathbf{T}}_{1:j} = (\widetilde{T}_k)_{k \in 1:j}$ and

$$\widetilde{T}_j(\mathbf{x}_{1:j}, x_j) = T_j^{(1)} \left( \mathbf{T}_{1:j}^{(2)}(\mathbf{x}_{1:j}), T_j^{(2)}(\mathbf{x}_{1:j}, x_j) \right),$$

where $\mathbf{T}_{1:j}^{(2)} = (T_k^{(2)})_{k \in 1:j}$. Since $\widetilde{T}_j$ depends only on $\mathbf{x}_{1:j}$, $\widetilde{\mathbf{T}}$ is a triangular mapping.

According to the form of $\mathbf{T}^{(1)}$, for any $j \in 1\!:\!d$ and any $\mathbf{x} \in \Omega_{\mathbf{X}}$, let $\mathbf{y} = \mathbf{T}^{(2)}(\mathbf{x})$. Then,

$$((\mathbf{T}^{(1)} \circ \mathbf{T}^{(2)})(\mathbf{x}))_j = (\mathbf{T}^{(1)}(\mathbf{y}))_j = T_j^{(1)}(\mathbf{y}_{1:j}, y_j).$$

According to the form of $\mathbf{T}^{(2)}$, we have

$$\mathbf{y}_{1:j} = (\mathbf{T}^{(2)}(\mathbf{x}))_{1:j} = (T_k^{(2)}(\mathbf{x}_{1:k-1}, x_k))_{k \in 1:j} = \mathbf{T}_{1:j}^{(2)}(\mathbf{x}_{1:j}),$$

and $y_j = T^{(2)}(\mathbf{x}_{1:j}, x_j)$. Therefore,

$$((\mathbf{T}^{(1)} \circ \mathbf{T}^{(2)})(\mathbf{x}))_j = T_j^{(1)}(\mathbf{y}_{1:j}, y_j)$$
$$= T_j^{(1)}\left(\mathbf{T}_{1:j}^{(2)}(\mathbf{x}_{1:j}), T^{(2)}(\mathbf{x}_{1:j}, x_j)\right) = \widetilde{T}_j(\mathbf{x}_{1:j}, x_j)$$

for any $j \in 1:d$. Let $\widetilde{\mathbf{T}} = \widetilde{\mathbf{T}}_{1:j}$. That is, $\widetilde{\mathbf{T}} = \mathbf{T}^{(1)} \circ \mathbf{T}^{(2)}$. Since $\widetilde{\mathbf{T}}$ is a triangular mapping, $\mathbf{T}^{(1)} \circ \mathbf{T}^{(2)}$ is also a triangular mapping.

Next, we consider the monotonicity signature. For each $j \in 1:d$, recalling that

$$\widetilde{T}_j(\mathbf{x}_{1:j}, x_j) = T_j^{(1)}\left(\mathbf{T}_{1:j}^{(2)}(\mathbf{x}_{1:j}), T^{(2)}(\mathbf{x}_{1:j}, x_j)\right).$$

If for any $\mathbf{x}_{1:j-1} \in \Omega_{\mathbf{X}_{1:j-1}}$, the function $T_j^{(2)}(\mathbf{x}_{1:j-1}, \cdot) : \Omega_{X_j} \to \Omega_{Y_j}$ is s.m.i., and for any $\mathbf{y}_{1:j-1} \in \Omega_{\mathbf{Y}_{1:j-1}}$, the function $T_j^{(1)}(\mathbf{y}_{1:j-1}, \cdot) : \Omega_{Y_j} \to \Omega_{Z_j}$ is also s.m.i., then the composition $T_j^{(1)}(\mathbf{y}_{1:j-1}, \cdot) \circ T_j^{(2)}(\mathbf{x}_{1:j-1}, \cdot)$ is s.m.i. According to the definition of monotonic functions, if both functions have the same monotonicity, the composition is s.m.i.; if they have opposite monotonicities, the composition is s.m.d. Therefore, according to the definition of the monotonicity signature, for each $j \in 1:d$, we have $\xi(\widetilde{T}_j) = \xi_j^{(1)} * \xi_j^{(2)}$. Consequently,

$$\xi(\mathbf{T}^{(1)} \circ \mathbf{T}^{(2)}) = \xi(\widetilde{\mathbf{T}}) = (\xi(\widetilde{T}_j))_{j \in 1:d} = (\xi_j^{(1)} * \xi_j^{(2)})_{j \in 1:d} = (\xi(T_j^{(1)}) * \xi(T_j^{(2)}))_{j \in 1:d} = \xi(\mathbf{T}^{(1)}) \odot \xi(\mathbf{T}^{(2)}),$$

i.e., the monotonicity signature satisfies $\xi(\mathbf{T}^{(1)} \circ \mathbf{T}^{(2)}) = \xi(\mathbf{T}^{(1)}) \odot \xi(\mathbf{T}^{(2)})$.

Moreover, since $\xi_j^{(1)}, \xi_j^{(2)} \in \{-1, 1\}$, it follows that $|\xi_j^{(1)} * \xi_j^{(2)}| = 1$ for any $j \in J$. Therefore, $\sum_{j=1}^d |\xi_j^{(1)} * \xi_j^{(2)}| = d$. Additionally, since $\mathbf{T}^{(1)} \circ \mathbf{T}^{(2)}$ is a triangular mapping, according to the definition of a TM mapping, $\mathbf{T}^{(1)} \circ \mathbf{T}^{(2)}$ is indeed a TM mapping. $\qquad\square$

*Remark* A.17. For two TM mappings $\mathbf{T}^{(1)} : \mathbb{R}^d \to \mathbb{R}^d$ and $\mathbf{T}^{(2)} : \mathbb{R}^d \to \mathbb{R}^d$, if $\xi(\mathbf{T}^{(1)}) = \xi(\mathbf{T}^{(2)})$, then $\xi(\mathbf{T}^{(1)} \circ \mathbf{T}^{(2)}) = (1)_{1:d}$, meaning that $\mathbf{T}^{(1)} \circ \mathbf{T}^{(2)}$ is a TMI mapping.

*Proof.* This result trivially follows from the component-wise multiplication and Lemma A.16. $\qquad\square$

**Lemma A.18.** *For a TM mapping* $\mathbf{T} : \mathbb{R}^d \to \mathbb{R}^d$*, its subcomponent* $\mathbf{T}_{i:j}$ *restricted to any* $\mathbf{x}_{1:i-1} \in \mathbb{R}^{i-1}$*, i.e.,* $\mathbf{T}_{i:j}(\mathbf{x}_{1:i-1}, \cdot)$*, is also a TM mapping for any* $i, j \in 1:d$ *with* $i < j$*, and its monotonicity signature satisfies* $\xi(\mathbf{T}_{i:j}(\mathbf{x}_{1:i-1}, \cdot))_{i:j} = (\xi(\mathbf{T}))_{i:j}$.

*Proof.* To clarify the proof, we denote the domain or codomain by the symbols $\Omega_{\mathbf{X}}$ and $\Omega_{\mathbf{Z}}$, both of which actually refer to $\mathbb{R}^d$. For a TM mapping $\mathbf{T} : \Omega_{\mathbf{X}} \to \Omega_{\mathbf{Z}}$, assume it is of the form $\mathbf{T}(\mathbf{x}) = (T_j(\mathbf{x}_{1:j-1}, x_j))_{j \in J}$, where $T_j : \Omega_{\mathbf{X}_{1:j-1}} \times \Omega_{X_j} \to \Omega_{Z_j}$. According to the definition of a TM mapping, for any $\mathbf{x}_{1:j-1} \in \Omega_{\mathbf{X}_{1:j-1}}$, the function $T_j(\mathbf{x}_{1:j-1}, \cdot) : \Omega_{X_j} \to \Omega_{Z_j}$ is consistently strictly monotonic. Therefore, for each $j \in 1:d$, there exists $\xi_j \in \{-1, 1\}$ such that $\xi(T_j) = \xi_j$.

The subcomponent $\mathbf{T}_{i:j}$ is of the form $(T_k(\mathbf{x}_{1:i-1}, \mathbf{x}_{i:j-1}, x_j))_{k \in i:j}$. For any $\mathbf{x}_{1:i-1} \in \mathbb{R}^{i-1}$, the restriction $\mathbf{T}_{i:j}(\mathbf{x}_{1:i-1}, \cdot)$ is of the form $(T_k(\mathbf{x}_{1:i-1}, \mathbf{x}_{i:j-1}, x_j))_{k \in i:j}$. Since $T_k(\mathbf{x}_{1:i-1}, \cdot)$ depends only on $\mathbf{x}_{i:j}$, $\mathbf{T}_{i:j}(\mathbf{x}_{1:i-1}, \cdot)$ is a triangular mapping.

Now consider the monotonicity signature. Assume that for each $k \in i : j$, according to the definition of $\xi(T_k)$, $T_k$ is s.m.i. with respect to $x_k$ given any $\mathbf{x}_{1:k-1} \in \Omega_{\mathbf{X}_{1:k-1}}$. Since $(\mathbf{x}'_{1:i-1}, \mathbf{x}_{i:k-1}) \in \Omega_{\mathbf{X}_{1:k-1}}$, $T_k(\mathbf{x}'_{1:i-1}, \cdot)$ is also s.m.i. with respect to $x_k$ given any $\mathbf{x}_{i:k-1} \in \Omega_{\mathbf{X}_{i:k-1}}$. The same conclusion holds when $T_k$ is s.m.d.

Thus, according to the definition of the monotonicity signature, for any $\mathbf{x}_{1:i-1} \in \mathbb{R}^{i-1}$ and each $k \in i : j$, we have $\xi(T_k(\mathbf{x}_{1:i-1}, \cdot)) = \xi_k$. Consequently,

$$\xi(\mathbf{T}_{i:j}(\mathbf{x}_{1:i-1}, \cdot)) = (\xi(T_k(\mathbf{x}_{1:i-1}, \cdot)))_{k \in i:j} = (\xi_k)_{k \in i:j} = (\xi(\mathbf{T}))_{i:j}.$$

Moreover, since $\sum_{k=i}^j |\xi_k| = j - i + 1$, and because $\mathbf{T}_{i:j}(\mathbf{x}_{1:i-1}, \cdot)$ is a triangular mapping, it follows from the definition of TM mappings that $\mathbf{T}_{i:j}(\mathbf{x}_{1:i-1}, \cdot)$ is a TM mapping. $\qquad\square$

**Lemma A.19.** *For the vectorization $\iota$ under $\leq$, for each $i \in \mathcal{I}$, there exists $l_i = 1 + \sum_{i \in pr(i)} d_i$, such that the image set $\iota[\{(i,j)\}_{j \in 1:d_i}] = l_i : l_i + d_i - 1$, where $pr(i)$ denotes the prefix of index $i$ under $\leq$.*

*Proof.* Recall that the vectorization $\iota$ is a flattening mapping such that for any $(i,j),(i,j') \in \mathcal{I}$, $\iota(i,j) - \iota(i,j') = j - j'$, and for any $(i,j),(i',j') \in \mathcal{I}$, $i < i'$ implies $\iota(i,j) < \iota(i',j')$. We prove the result using induction over the partial order $\leq$.

When $|\operatorname{pr}(i)| = 0$, we have $l_i = 1$. Since there does not exist any $i' \in \mathcal{I}$ such that $i' < i$, there are no pairs $(i',j')$ such that $(i',j') < (i,1)$. Thus, by the definition of vectorization, $\iota(i,1) = 1$. For any $\{(i,j)\}_{j \in 1:d_i}$, we have $\iota(i,j) - \iota(i,1) = j - 1$, so $\iota(i,j) = j - 1 + 1 = j$. Therefore, the image set is $\iota[\{(i,j)\}_{j \in 1:d_i}] = 1 : d_i = l_i : l_i + d_i - 1$.

Suppose for any $k \in \mathcal{I}$ with $|\operatorname{pr}(k)| < |\operatorname{pr}(i)|$, there exists $l_k = 1 + \sum_{i \in \operatorname{pr}(k)} d_i$ such that the image set $\iota[\{(k,j)\}_{j \in 1:d_k}] = l_k : l_k + d_k - 1$. Let $l_i = 1 + \sum_{i \in \operatorname{pr}(i)} d_i$.

Since $\iota$ is a bijection, the preimage $\iota^{-1}[1 : l_i - 1] = \iota^{-1}[\bigcup_{\operatorname{pr}(k) < \operatorname{pr}(i)}(l_k : l_k + d_k - 1)] = \bigcup_{\operatorname{pr}(k) < \operatorname{pr}(i)}\{(k,j)\}_{j \in 1:d_k}$ by the inductive hypothesis. Therefore, $\iota(i,1) = 1 + \sum_{i \in \operatorname{pr}(i)} d_i$.

For any $\{(i,j)\}_{j \in 1:d_i}$, we have $\iota(i,j) - \iota(i,1) = j - 1$, so $\iota(i,j) = j - 1 + (1 + \sum_{i \in \operatorname{pr}(i)} d_i) = j + \sum_{i \in \operatorname{pr}(i)} d_i$. Therefore, the image set is $\iota[\{(i,j)\}_{j \in 1:d_i}] = 1 + \sum_{i \in \operatorname{pr}(i)} d_i : d_i + \sum_{i \in \operatorname{pr}(i)} = l_i : l_i + d_i - 1$. $\qquad\square$

**Lemma A.20.** *For the vectorization $\iota$ under $\leq$, if $\leq$ is a causal order, let $\boldsymbol{an}(i) = \{(j,k) \mid j \in an(i), k \in 1:d_j\}$. Then, for each $i \in \mathcal{I}$, the image set satisfies $\iota[\boldsymbol{an}(i)] \subseteq 1:l_i - 1$.*

*Proof.* We prove this by contradiction. Assume there exists $j \in an(i)$ such that $l_j + d_j - 1 \geq l_i$. Since $j \neq i$ (as $i \notin an(i)$ due to acyclicity), and using the properties of vectorization, it follows that $l_j \geq l_i + d_i$.

Thus, $l_j > l_i$ holds if and only if $j > i$ (by the properties of vectorization). However, since $j \in an(i)$, we know $j < i$ (by the definition of causal order). This contradicts the assumption that $j > i$. Therefore, the assumption $l_j + d_j - 1 \geq l_i$ does not hold. Hence, for any $i \in \mathcal{I}$, we have $\iota[\boldsymbol{an}(i)] \subseteq 1:l_i - 1$. $\qquad\square$

**Proposition 5.3.** *A recursive SCM $\mathcal{M}$ is a TM-SCM if and only if $f_i(\mathbf{v},\cdot)$ is a TM mapping and there exists $\xi_i$ such that $\xi(f_i(\mathbf{v},\cdot)) = \xi_i$ for every $i \in \mathcal{I}$ and all $\mathbf{v} \in \Omega_{\mathbf{V}}$.*

*Proof.* Consider a causal order $\leq$ with its vectorization $\iota$. Define the reindexing mapping $P_{\iota,i} = (P_{\iota,i,j})_{j \in d_i}$. By Lemma A.19, the vectorization satisfies $\mathbf{P}_{\iota,i}((x_{i,j})_{j \in 1:d_i}) = (x_j)_{j \in l_i:l_i+d_i-1}$ and its inverse $\mathbf{P}_{\iota,i}^{-1}((x_j)_{j \in l_i:l_i+d_i-1}) = (x_{i,j})_{j \in 1:d_i}$. For simplicity, when we refer to $v_i$ or $u_i$ indexed by $\mathcal{I}$, we actually mean the entire vector $(x_{i,j})_{j \in 1:d_i}$.

$\implies$: Assume that the SCM $\mathcal{M}$ is a TM-SCM. By Definition 5.2, there exists a causal order $\leq$ with its vectorization $\iota$ such that the reindexed solution mapping $\mathbf{P}_\iota \circ \Gamma$ is a TM mapping. Let $\mathbf{P}_\iota \circ \Gamma = \mathbf{T} = (T_j)_{j \in 1:D}$, where each $T_j$ is of the form $T_j(\mathbf{u}_{1:j-1}, u_j)$, and $D = \sum_{i \in \mathcal{I}} d_i$. Then, for each $i \in \mathcal{I}$, we have $(\Gamma)_i = \mathbf{P}_{\iota,i}^{-1}((T_j)_{j \in l_i:l_i+d_i-1}) = \mathbf{P}_{\iota,i}^{-1} \circ \mathbf{T}_{l_i:l_i+d_i-1}$.

By Lemma A.2, $\Gamma(\mathbf{u}) = (\tilde{f}_i(\mathbf{u}_{an^*(i)}))_{i \in \mathcal{I}}$, so $\tilde{f}_i = \mathbf{P}_{\iota,i}^{-1} \circ \mathbf{T}_{l_i:l_i+d_i-1}$. The function $\mathbf{T}_{l_i:l_i+d_i-1}(\mathbf{u}_{1:l_i-1}, \cdot)$ is of the form $\mathbf{T}_{l_i:l_i+d_i-1}(\mathbf{u}_{1:l_i-1}, \mathbf{u}_{l_i:l_i+d_i-1})$. Thus, $(\mathbf{P}_{\iota,i}^{-1} \circ \mathbf{T}_{l_i:l_i+d_i-1})(\mathbf{u}_{\iota^{-1}[1:l_i-1]}, \cdot)$ is of the form

$$(\mathbf{P}_{\iota,i}^{-1} \circ \mathbf{T}_{l_i:l_i+d_i-1})(\mathbf{u}_{\iota^{-1}[1:l_i-1]}, (u_{i,j})_{j \in 1:d_i}).$$

By Lemma A.20 and the fact that $\mathbf{u}_{an(i)} = (\mathbf{u}_{i,j})_{\boldsymbol{an}(i)}$, $\tilde{f}_i$ is independent of $\mathbf{u}_{1:l_i-1 \setminus \iota[\boldsymbol{an}(i)]}$. Hence,

$$\tilde{f}_i(\mathbf{u}_{an(i)}, u_i) = (\mathbf{P}_{\iota,i}^{-1} \circ \mathbf{T}_{l_i:l_i+d_i-1})(\mathbf{u}_{\boldsymbol{an}(i)}, (u_{i,j})_{j \in 1:d_i}),$$

implying that $\tilde{f}_i(\mathbf{u}_{an(i)}, \cdot)$ is a triangular mapping.

By Lemma A.18, the monotonicity signature satisfies $\xi(\mathbf{T}_{l_i:l_i+d_i-1}(\mathbf{x}_{1:l_i-1}, \cdot))_{l_i:l_i+d_i-1} = (\xi(\mathbf{T}))_{l_i:l_i+d_i-1}$. Thus, $\xi(\tilde{f}_i(\mathbf{u}_{an(i)}, \cdot)) = (\xi(\mathbf{T}))_{l_i:l_i+d_i-1}$. Since $\sum_{j \in l_i:l_i+d_i-1} |(\xi(\mathbf{T}))_j| = d_i$, it follows that $\tilde{f}_i(\mathbf{u}_{an(i)}, \cdot)$ is a TM mapping.

Finally, since TM mappings are bijections, for any given $\mathbf{v} \in \Omega_{\mathbf{V}}$, there exists $\mathbf{u}' \in \Omega_{\mathbf{U}}$ such that $\mathbf{u}' = \Gamma^{-1}(\mathbf{v})$. By Definition A.1, $f_i((\tilde{f}_k(\mathbf{u}'_{an^*(k)}))_{k \in pa(i)}, \cdot) = \tilde{f}_i(\mathbf{u}'_{an(i)}, \cdot)$. Therefore, for all $\mathbf{v} \in \Omega_{\mathbf{V}}$, $f_i(\mathbf{v}, \cdot)$ is a TM mapping, and there exists $\xi_i = (\xi(\mathbf{T}))_{l_i:l_i+d_i-1}$ such that $\xi(f_i(\mathbf{v}, \cdot)) = \xi_i$.

$\Longleftarrow$ : Let $\leq$ be any causal order and $\iota$ its vectorization. Assume $f_i(\mathbf{v}, \cdot)$ is a TM mapping and there exists $\xi_i$ such that $\xi(f_i(\mathbf{v}, \cdot)) = \xi_i$ for each $i \in \mathcal{I}$ and all $\mathbf{v} \in \Omega_{\mathbf{V}}$. For any given $\mathbf{v} \in \Omega_{\mathbf{V}}$, there exists $\mathbf{u}' \in \Omega_{\mathbf{U}}$ such that $\mathbf{u}' = \boldsymbol{\Gamma}^{-1}(\mathbf{v})$. By Definition A.1,

$$f_i((\tilde{f}_k(\mathbf{u}'_{\mathrm{an}^*(k)}))_{k \in \mathrm{pa}(i)}, \cdot) = \tilde{f}_i(\mathbf{u}'_{\mathrm{an}(i)}, \cdot).$$

By assumption, $\tilde{f}_i(\mathbf{u}_{\mathrm{an}(i)}, \cdot)$ is a TM mapping with $\xi(\tilde{f}_i(\mathbf{v}, \cdot)) = \xi_i$. Using Lemma A.20, we construct $\mathbf{T}_{l_i:l_i+d_i-1} = \mathbf{P}_{\iota,i} \circ \tilde{f}_i$, where $\xi(T_j) = (\xi_i)_{j-l_i+1}$ for each $j \in l_i : l_i + d_i - 1$.

By Lemma A.2, $\boldsymbol{\Gamma} = (\tilde{f}_i)_{i \in \mathcal{I}}$. Thus,

$$\mathbf{P}_\iota \circ \boldsymbol{\Gamma} = (\mathbf{P}_{\iota,i} \circ \tilde{f}_i)_{i \in \mathcal{I}} = (\mathbf{T}_{l_i:l_i+d_i-1})_{i \in \mathcal{I}} = (T_j)_{1:D}.$$

Each $T_j$ depends only on $\mathbf{u}_{1:l_i-1}$, and since $\sum_{j \in 1:D} |\xi(T_j)| = \sum_{i \in \mathcal{I}} d_i = D$, $\mathbf{P}_\iota \circ \boldsymbol{\Gamma}$ is a TM mapping. By Definition 5.2, $\mathcal{M}$ is a TM-SCM. $\qquad\square$

**Corollary 5.4** (EI-ID from Triangular Monotonicity). *An SCM is $\sim_{\mathrm{EI}}$-identifiable from $\mathcal{A}_{\{TM\text{-}SCM, \leq, M, P_{\mathbf{V}}\}}$.*

*Proof.* We aim to prove that if $\mathcal{M} \models \mathcal{A}_{\{TM\text{-}SCM, \leq, M, P_{\mathbf{V}}\}}$, then $\mathcal{M} \models \mathcal{A}_{\{BSCM, \leq, M, P_{\mathbf{V}}, KR\}}$, allowing us to utilize Theorem 4.8. Specifically, it suffices to show that if $\mathcal{M} \models \mathcal{A}_{\{TM\text{-}SCM, \leq, M, P_{\mathbf{V}}\}}$, then $\mathcal{A}_{KR}(\mathcal{M})$ holds.

Assume $\mathcal{M} \models \mathcal{A}_{\{TM\text{-}SCM, \leq, M, P_{\mathbf{V}}\}}$. Then $\mathcal{M}$ is a Markovian BSCM, inducing the causal order $\leq$, with the observed distribution $P_{\mathbf{V}}$. Since $\mathcal{M}$ is a TM-SCM, by Proposition 5.3, for each $i \in \mathcal{I}$ and all $\mathbf{v} \in \Omega_{\mathbf{V}}$, the causal mechanism $f_i(\mathbf{v}, \cdot)$ is a TM mapping, and there exists $\xi_i$ such that $\xi(f_i(\mathbf{v}, \cdot)) = \xi_i$. Furthermore, by the definition of counterfactual transport and Remark A.17, $K_{\mathcal{M},i} = (f_i(\mathbf{v}', \cdot)) \circ (f_i(\mathbf{v}, \cdot))^{-1}$ is a TMI mapping for any pair $\mathbf{v}, \mathbf{v}' \in \Omega_{\mathbf{V}}$.

Now, for each $i \in \mathcal{I}$, consider an arbitrary pair $\mathbf{v}, \mathbf{v}' \in \Omega_{\mathbf{V}}$. Given the observed distribution $P_{\mathbf{V}}$, the conditional distribution $P_{V_i | \mathbf{V}_{\mathrm{pa}(i)}}(\cdot, \mathbf{v}_{\mathrm{pa}(i)})$ is almost surely uniquely determined by Theorem A.12. Consequently, the conditional distributions $P_{V_i | \mathbf{V}_{\mathrm{pr}(i)}}(\cdot, \mathbf{v}_{\mathrm{pr}(i)})$ and $P_{V_i | \mathbf{V}_{\mathrm{pr}(i)}}(\cdot, \mathbf{v}'_{\mathrm{pr}(i)})$ are also almost surely determined.

Given these distributions, by Lemma 5.1 and Proposition 4.5, the TMI mapping $K_{\mathcal{M},i} = (f_i(\mathbf{v}', \cdot)) \circ (f_i(\mathbf{v}, \cdot))^{-1}$ is almost surely equivalent to KR transport. Thus, $\mathcal{A}_{KR}(\mathcal{M})$ holds. Hence, we have proven that if $\mathcal{M} \models \mathcal{A}_{\{TM\text{-}SCM, \leq, M, P_{\mathbf{V}}\}}$, then $\mathcal{M} \models \mathcal{A}_{\{BSCM, \leq, M, P_{\mathbf{V}}, KR\}}$.

By Theorem 4.8, for any pair $\mathcal{M}^{(1)}, \mathcal{M}^{(2)} \models \mathcal{A}_{\{BSCM, \leq, M, P_{\mathbf{V}}, KR\}}$, we have $\mathcal{M}^{(1)} \sim_{\mathrm{EI}} \mathcal{M}^{(2)}$. Therefore, for any pair $\mathcal{M}^{(1)}, \mathcal{M}^{(2)} \models \mathcal{A}_{\{TM\text{-}SCM, \leq, M, P_{\mathbf{V}}\}}$, we also have $\mathcal{M}^{(1)} \sim_{\mathrm{EI}} \mathcal{M}^{(2)}$. Thus, SCMs from $\mathcal{A}_{\{TM\text{-}SCM, \leq, M, P_{\mathbf{V}}\}}$ are $\sim_{\mathrm{EI}}$-identifiable. $\qquad\square$

**Lemma A.21** (restated from Khoa Le et al. 2025, Theorem 1). *If the velocity field $\boldsymbol{v} : \mathbb{R}^d \times [0, 1] \to \mathbb{R}^d$ is a Lipschitz continuous triangular mapping for any $t \in [0, 1]$, then for the ODE*

$$\begin{cases} \mathrm{d}\boldsymbol{x}(t) = \boldsymbol{v}(\boldsymbol{x}(t), t)\,\mathrm{d}t, \\ \boldsymbol{x}(0) = \mathbf{x}_0, \end{cases}$$

*the flow $\mathcal{T}(\cdot, t)$ is a TMI mapping for any $t \in [0, 1]$.*

*Proof.* We first show that $\mathcal{T}(\cdot, t)$ is a triangular mapping. Writing $\mathcal{T}(\cdot, t)$ as $(\mathcal{T}_j(\cdot, t))_{j=1:d}$ and $\boldsymbol{x}(t)$ as $(\boldsymbol{x}_j(t))_{j=1:d}$, consider dimension $j = 1$. Since $\boldsymbol{v}_1(\cdot, t)$ is a triangular mapping depending only on $\boldsymbol{x}_1(t)$, the corresponding ODE can be written as

$$\begin{cases} \mathrm{d}\boldsymbol{x}_1(t) = \boldsymbol{v}_1(\boldsymbol{x}_1(t), t)\,\mathrm{d}t, \\ \boldsymbol{x}_1(0) = \mathbf{x}_{1,0}. \end{cases}$$

This is a one-dimensional ODE. By Lipschitz continuity and the Picard–Lindelöf theorem, the solution exists and is unique, implying that $\boldsymbol{x}_1(t)$ depends only on $\mathbf{x}_{1,0}$. Therefore, by the definition of the flow, $\mathcal{T}_1(\mathbf{x}_0, t) = \boldsymbol{x}_1(t)$ depends only on $\mathbf{x}_{1,0}$.

Now, assume by induction that $\mathcal{T}_{j-1}(\mathbf{x}_0, t) = \boldsymbol{x}_{j-1}(t)$ depends only on $\mathbf{x}_{1:j-1,0}$. For dimension $j$, since $\boldsymbol{v}_j(\cdot, t)$ is a triangular mapping depending only on $\boldsymbol{x}_{1:j}(t)$, the corresponding ODE can be written as

$$\begin{cases} \mathrm{d}\boldsymbol{x}_j(t) = \boldsymbol{v}_j(\boldsymbol{x}_{1:j}(t), t)\,\mathrm{d}t, \\ \boldsymbol{x}_{1:j}(0) = \mathbf{x}_{1:j,0}, \end{cases}$$

or equivalently,

$$\begin{cases} \mathrm{d}\boldsymbol{x}_j(t) = \boldsymbol{v}_j(\boldsymbol{x}_{1:j-1}(t), \boldsymbol{x}_j(t), t)\,\mathrm{d}t, \\ \boldsymbol{x}_j(0) = \mathbf{x}_{j,0}, \\ \boldsymbol{x}_{1:j-1}(0) = \mathbf{x}_{1:j-1,0}. \end{cases}$$

Here, $\boldsymbol{x}_{1:j-1}(t)$ depends only on $\mathbf{x}_{1:j-1,0}$ and can be treated as a constant. Thus, the ODE for $\boldsymbol{x}_j(t)$ becomes a one-dimensional ODE. By Lipschitz continuity and the Picard–Lindelöf theorem, the solution exists and is unique, implying that $\boldsymbol{x}_j(t)$ depends only on $\mathbf{x}_{j,0}$. Consequently, $\boldsymbol{x}_j(t)$ depends only on $\mathbf{x}_{1:j,0}$ when $\mathbf{x}_{1:j-1,0}$ is unspecified. By the definition of the flow, $\mathcal{T}_j(\mathbf{x}_0, t) = \boldsymbol{x}_j(t)$ depends only on $\mathbf{x}_{1:j,0}$.

By induction, $\mathcal{T}(\cdot, t)$ is a triangular mapping.

Next, we show that $\mathcal{T}(\cdot, t)$ is a TMI mapping. For each dimension $j$, consider two different initial values $x_{j,0}, x'_{j,0} \in \mathbb{R}$ such that $x_{j,0} < x'_{j,0}$. From the previous proof, given $\mathbf{x}_{1:j-1,0}$, the ODE is one-dimensional. Using a proof by contradiction, suppose that at time $t = t'$, we have $\boldsymbol{x}_j(t) \geq \boldsymbol{x}'_j(t)$, where $\boldsymbol{x}_j(t)$ and $\boldsymbol{x}'_j(t)$ are the solutions starting at $x_{j,0}$ and $x'_{j,0}$, respectively. By continuity, there exists $t^* \leq t'$ such that $\boldsymbol{x}_j(t^*) = \boldsymbol{x}'_j(t^*)$. However, by Lipschitz continuity and the Picard–Lindelöf theorem, the solution to the ODE is unique for any $t$. Thus, if $\boldsymbol{x}_j(t^*) = \boldsymbol{x}'_j(t^*)$ at some $t$, then $\boldsymbol{x}_j(t) = \boldsymbol{x}'_j(t)$ for all $t$, including $t = 0$. This contradicts the assumption $x_{j,0} < x'_{j,0}$, so the hypothesis is false.

Therefore, given $\mathbf{x}_{1:j-1,0}$, if $x_{j,0} < x'_{j,0}$, then $\boldsymbol{x}_j(t) < \boldsymbol{x}'_j(t)$. In other words, $\xi(\mathcal{T}_j(\cdot, t)) = \xi(\boldsymbol{x}_j) = 1$.

In summary, $\mathcal{T}(\cdot, t)$ is a triangular mapping and $\sum_{j=1}^d \xi(\mathcal{T}_j(\cdot, t)) = d$. By the definition of a TMI mapping, $\mathcal{T}(\cdot, t)$ is a TMI mapping. $\qquad\square$

# B. Related Works

## B.1. Identifiability

**Counterfactual Identification**    Previous work on counterfactual identification has typically focused on identifying specific classes of counterfactual statements $\boldsymbol{\varphi} \subseteq \mathcal{L}_3$, which can be categorized as follows: (i) Constraining causal structures to identify specific counterfactual effects. This involves representing counterfactual statements using symbols, graphs, and available observational and interventional distributions. Examples include the identification of counterfactual probabilities (Shpitser & Pearl, 2008), nested counterfactual probabilities (Correa et al., 2021), sufficient or necessary probabilities (Tian & Pearl, 2000), path-specific effects (Avin et al., 2005), discrimination effects (Zhang & Bareinboim, 2018), and identification via neural networks (Xia et al., 2023). (ii) Constraining causal mechanisms to identify counterfactual outcomes. This focuses on deterministic counterfactuals inferred through *abduction-action-prediction* (Pearl, 2009), a special class of counterfactual statements of the form $\mathbf{y}_{[\mathbf{x}]} \,|\, \mathbf{x}', \mathbf{y}'$. Identifiability is achieved primarily by imposing parametric constraints such as monotonicity, as in (Lu et al., 2020), (Nasr-Esfahany et al., 2023), and (Scetbon et al., 2024).

**Identifiability**    The identifiability exhibited by generative models has been a continuously discussed topic in the literature, spanning tasks such as representation learning (Bengio et al., 2013), causal discovery (Glymour et al., 2019), causal representation learning (Schölkopf et al., 2021), and causal quantity reasoning (Pearl, 2009). Identifiability in representation learning refers to uniquely determining latent factors under certain ambiguities, such as permutation and scaling (Hyvärinen et al., 2023), affine transformations (Kivva et al., 2022), or component-wise invertible transformations (Gresele et al., 2021). In causal discovery, identifiability pertains to structural identifiability, which requires determining equivalence classes of causal graphs (Spirtes & Zhang, 2016; Vowels et al., 2022) or identifying the direction of cause and effect (Shimizu et al., 2006; Hoyer et al., 2008). In causal representation learning, identifiability involves determining latent causal variables and causal structures, where causal variables must satisfy component-wise diffeomorphic mappings, and causal structures require graph isomorphism (Brehmer et al., 2022; von Kügelgen et al., 2023). Finally, in causal quantity reasoning, identifiability refers to the consistency of causal statements (Pearl, 2009), which was previously termed causal identification.

## B.2. Triangular Monotonic SCMs

To demonstrate the broad applicability of Corollary 5.4, we summarize several representative works from the past and establish their connection to our theory by proving that they are special cases of the TM-SCM. This highlights the significance of our theoretical contributions. Overall, these works can be categorized based on the criteria outlined in Table 2, which further inspires us to derive four prototypical models of neural TM-SCMs.

*Table 2.* Representative works that are special cases of the TM-SCM.

| Model | Research Domain | TM Object | TM Type | TM Principle |
|-------|-----------------|-----------|---------|--------------|
| LiNGAM | Cause-effect Identification | $f_i$ | ANM | $\mathbf{b}_i^\mathsf{T}\mathbf{v}_{\mathrm{pa}(i)} + u_i$ |
| ANM | Cause-effect Identification | $f_i$ | ANM | $g_i(\mathbf{v}_{\mathrm{pa}(i)}) + u_i$ |
| LSNM | Cause-effect Identification | $f_i$ | ANM | $l_i(\mathbf{v}_{\mathrm{pa}(i)}) + s_i(\mathbf{v}_{\mathrm{pa}(i)}) \cdot u_i$ |
| BGM | Counterfactual Identification | $f_i$ | UMM | Prior |
| PNL | Cause-effect Identification | $f_i$ | UMM | $h(g_i(\mathbf{v}_{\mathrm{pa}(i)}) + u_i)$ |
| FiP | Counterfactual Identification | $f_i$ | UMM | $C^1$ smoothness |
| CAREFL | Cause-effect Identification | $\mathbf{P}_\iota \circ \mathbf{\Gamma}$ | DCF | Affine Transform |
| StrAF | Causal Quantity Estimation | $\mathbf{P}_\iota \circ \mathbf{\Gamma}$ | DCF | UMNN Transform |
| CausalNF | Causal Quantity Estimation | $\mathbf{P}_\iota \circ \mathbf{\Gamma}$ | DCF | Monotonic RQS Transform |
| CCNF | Causal Quantity Estimation | $\mathbf{P}_\iota \circ \mathbf{\Gamma}$ | DCF | Partial Causal Transform |
| CKM | Generalization | $\mathbf{P}_\iota \circ \mathbf{\Gamma}$ | CCF | Picard–Lindelöf |
| CFM | Causal Quantity Estimation | $\mathbf{P}_\iota \circ \mathbf{\Gamma}$ | CCF | Picard–Lindelöf |

**Original Research Domain**  The related works summarized here originate from various research domains, each with distinct objectives. For instance, many studies on specialized SCMs aim to address the task of cause-effect identification in causal discovery. Others, as previously mentioned, focus on identifying counterfactual outcomes by imposing constraints on causal mechanisms. Some works explore how to generalize SCMs to other scenarios. Another line of research, more aligned with deep generative models, seeks to address efficient causal effect estimation. Interestingly, despite their differing objectives, the SCM formulations of these works uniformly fall under the TM-SCM framework.

**Triangular Monotonicity Object**  The TM-SCM framework admits two equivalent definitions. One, as described in Definition 5.2, requires the solution mapping $\mathbf{P}_\iota \circ \mathbf{\Gamma}$, obtained by reindexing under the causal ordering vectorization $\iota$, to be a TM mapping. The other, given in Proposition 5.3, ensures that the causal mechanism $f_i(\mathbf{v}, \cdot)$ for each $i \in \mathcal{I}$ and any $\mathbf{v} \in \Omega_\mathbf{V}$ is a TM mapping with consistent monotonicity signatures. Therefore, these works can be categorized based on whether they focus on constraining the global solution mapping $\mathbf{P}_\iota \circ \mathbf{\Gamma}$ or individual causal mechanisms $f_i$.

**Triangular Monotonicity Type**  Existing works can be further categorized based on the types of triangular monotonicity they achieve:

- **Additive Noise Mechanism (ANM)**: Additive noise mechanisms restrict each causal mechanism $f_i$ in the SCM to the form $\alpha(\mathbf{v}_{\mathrm{pa}(i)}) + \beta(\mathbf{v}_{\mathrm{pa}(i)}) \cdot u_i$, where $\beta$ is required to be a strictly positive function. Consequently, even for vector-valued cases, the causal mechanism $f_i(\mathbf{v}, \cdot)$ for each $i \in \mathcal{I}$ and any $\mathbf{v} \in \Omega_\mathbf{V}$ is always a TMI mapping. This follows from the Jacobian matrix being diagonal, with each diagonal element $(\beta(\mathbf{v}_{\mathrm{pa}(i)}))_j > 0$ due to the constraints on $\beta$. Relevant works include LiNGAM (Shimizu et al., 2006), ANM (Hoyer et al., 2008), and LSNM (Immer et al., 2023). These studies have inspired the construction of DNME-type neural TM-SCMs.
- **Univariate Monotonic Mechanism (UMM)**: In the univariate case, functions are necessarily strictly Monotonicity if they are invertible. Thus, it suffices to further constrain the monotonicity signature of each causal mechanism $f_i(\mathbf{v}, \cdot)$ to be consistent for every $i \in \mathcal{I}$ and any $\mathbf{v} \in \Omega_\mathbf{V}$. This consistency can be ensured by various approaches: (1) enforcing the function to be s.m.i or s.m.d through prior construction, (2) indirectly composing an additive noise mechanism with an invertible function, or (3) deriving monotonicity consistency from $C^1$ smoothness. These approaches correspond to BGM (Nasr-Esfahany et al., 2023), PNL (Zhang & Hyvärinen, 2009), and FiP (Scetbon et al., 2024), respectively. These works have inspired the construction of TNME-type neural TM-SCMs, extending from the univariate to the multivariate case.
- **Discrete Causal Flow (DCF)**: Certain discrete-time transformations in normalizing flows, such as affine transformations (Dinh et al., 2017), unconstrained monotonic neural networks (Wehenkel & Louppe, 2019), and monotonic rational quadratic splines (Durkan et al., 2019), can be shown to be TMI mappings. When these transformation blocks are composed, the entire normalizing flow also remains a TMI mapping. Several works establish a connection between $\mathbf{\Gamma}$ and normalizing flows, such as CAREFL (Khemakhem et al., 2021), StrAF (Chen et al., 2023), CausalNF (Javaloy et al., 2023), and CCNF (Zhou et al., 2025), each employing different transformation blocks. These works can be classified as the CMSM-type neural TM-SCMs.
- **Continuous Causal Flow (CCF)**: In continuous-time settings where ODEs are used to construct SCMs, structural constraints are introduced into the velocity field, providing a novel pathway to induce TM mappings. Specifically, when the velocity field is triangular, the solution to the ODE is always a TMI mapping, as guaranteed by the Picard–Lindelöf

condition (see Lemma A.21). Some works establish equivalence between the ODE solution and the solution mapping $\Gamma$ of an SCM, such as CKM (Peters et al., 2022) and CFM (Khoa Le et al., 2025). These studies lay the foundation for constructing TVSM-type neural TM-SCMs.

Corollary 5.4 establishes that, under the additional assumptions of Markov property and identical causal ordering, and provided they induce the same observational distribution, the constructed models are identifiable up to the $\sim_{\mathrm{EI}}$-equivalence class. Furthermore, by Theorem 3.2, these models are mutually $\mathcal{L}_3$-consistent, implying that they are indistinguishable at the counterfactual level. Therefore, when these models are claimed to be counterfactually consistent or counterfactually identifiable, Corollary 5.4 and Theorem 3.2 theoretically justify such assertions.

## B.3. Theories

We now elaborate on how other related theorems mentioned in the main text can be interpreted as special cases of Theorem 3.2 and Corollary 5.4, starting with a restatement of these theorems using the notation of this paper.

**Special cases of Theorem 3.2**    Several prior works have introduced notions similar to exogenous isomorphism and shown that SCMs satisfying these equivalence relations enjoy counterfactual consistency, mirroring the result of Theorem 3.2.

For instance, (Peters et al., 2017) formalizes the concept of counterfactual equivalence and proves that, when two SCMs share the same exogenous distribution, counterfactual equivalence is guaranteed by the following theorem:

**Theorem B.1** (restated from Peters et al. 2017, Proposition 6.49)**.** *For two recursive SCMs $\mathcal{M}^{(1)}$ and $\mathcal{M}^{(2)}$, if their exogenous distributions satisfy $P_{\mathbf{U}}^{(1)} = P_{\mathbf{U}}^{(2)} = P_{\mathbf{U}}$, and for each $i \in \mathcal{I}$, for almost every $u_i \in \Omega_{U_i}$ and all $\mathbf{v} \in \Omega_{\mathbf{V}}$,*

$$f_i^{(2)}(\mathbf{v}_{pa^{(2)}(i)}, u_i) = f_i^{(1)}(\mathbf{v}_{pa^{(1)}(i)}, u_i),$$

*then $\mathcal{M}^{(1)}$ and $\mathcal{M}^{(2)}$ are counterfactually equivalent.*

The original purpose of this theorem was to establish the minimality property of SCMs: namely, if a mechanism in an SCM does not depend on one of its parent endogenous variables, one can always select an SCM with a sparser structure. However, as elaborated in the main text, the counterfactual equivalence defined in this way mandates a fixed latent representation.

(Nasr-Esfahany et al., 2023) relaxes counterfactual equivalence to non-fixed latent encodings: rather than requiring the exogenous distributions to coincide exactly, it only demands a component-wise bijection.

**Theorem B.2** (restated from Nasr-Esfahany et al. 2023, Proposition 6.2)**.** *Suppose $f(\mathbf{V}_{pa(i)}, U_i)$ is a BGM, meaning that for each realization $\mathbf{v}_{pa(i)}$ of $\mathbf{V}_{pa(i)}$, $f(\mathbf{v}_{pa(i)}, \cdot)$ is invertible. For two BGMs $f_1$ and $f_2$, if for almost every $u_i^{(1)} \in \Omega_{U_i}^{(1)}$ and all $\mathbf{v} \in \Omega_{\mathbf{V}}$,*

$$f^{(2)}(\mathbf{v}_{pa^{(2)}(i)}, g(u_i^{(1)})) = f^{(1)}(\mathbf{v}_{pa^{(1)}(i)}, u_i^{(1)}),$$

*where $g : \Omega_{U_i}^{(1)} \to \Omega_{U_i}^{(2)}$ is a bijection, then we say that $f_1$ and $f_2$ are equivalent. Two BGMs are equivalent if and only if they produce the same counterfactual outcomes.*

Although this result is weaker than that of (Peters et al., 2017), it is confined to BSCMs. Moreover, the original paper considers only a single causal mechanism within a BSCM, referred to as a BGM. In addition, the theorem addresses counterfactual outcomes rather than counterfactual distributions, a gap filled later by Theorem B.6. In contrast, although conceptually similar, Theorem 3.2 is not limited to BSCMs; it applies to any recursive SCM.

Recent advances in causal representation learning yield analogous results. Specifically, (Brehmer et al., 2022) investigates a class of models termed Latent Causal Models (LCMs). Because the causal mechanisms in an LCM are assumed to be point-wise diffeomorphic, an LCM can be viewed as a BSCM augmented with latent endogenous variables. Using an isomorphism-based criterion to characterize equivalence between LCMs, the authors show that a component-wise diffeomorphic latent representation aligns the counterfactual distributions of two LCMs (counterfactual distributions are referred to in their paper as weakly supervised distributions).

**Theorem B.3** (restated from Brehmer et al. 2022, Theorem 1)**.** *For LCMs $\mathcal{M}$ and $\mathcal{M}'$. If they can be reparameterized through the graph isomorphism and elementwise diffeomorphisms, we say there is an LCM isomorphism between them, or LCM equivalence, and we write $\mathcal{M} \sim_{LCM} \mathcal{M}'$. Assume that (i) the two models share the same observation space; (ii) the domains of every endogenous variable and its corresponding exogenous noise are $\mathbb{R}$; (iii) the intervention set contains all*

*atomic, perfect interventions; and (iv) the intervention distributions have full support. Under these conditions, $\mathcal{M} \sim_{LCM} \mathcal{M}'$ if and only if $\mathcal{M}$ and $\mathcal{M}'$ entail equal weakly supervised distributions.*

(Zhou et al., 2024) also investigates domain counterfactual equivalence between Invertible Latent Domain Causal Models (ILDs). An ILD incorporates latent endogenous variables together with a mixing function $g$. Under a fully connected DAG with a known causal ordering, the causal mechanism $f_d$ in domain $d$ is autoregressive. Because ILD assumes invertibility, the latent portion of its SCM can be regarded as a BSCM, and the domain counterfactual can be interpreted as a special instance of the generalized counterfactual in which the domain itself serves as the intervention. Accordingly, a domain-specific mechanism $f_d$ corresponds to the mechanism $f_{[d]}$ in the submodel $\mathcal{M}_{[d]}$.

**Theorem B.4** (restated from Zhou et al. 2024, Theorem 1). *For two ILDs $\mathcal{M}$ and $\mathcal{M}'$ whose mixing functions are $g$ and $g'$ and whose causal mechanisms are $f$ and $f'$, if for any domains $d, d'$ it holds that*

$$g \circ f_{d'} \circ f_d^{-1} \circ g^{-1} = g' \circ f'_{d'} \circ (f'_d)^{-1} \circ (g')^{-1},$$

*then $\mathcal{M}$ and $\mathcal{M}'$ are said to be domain counterfactual equivalent, denoted $\mathcal{M} \sim_{DC} \mathcal{M}'$. The relation $\mathcal{M} \sim_{DC} \mathcal{M}'$ holds if and only if there exist invertible maps $h_1, h_2$ such that $g' = g \circ h_1^{-1}$ is invertible and $f'_d = h_1 \circ f_d \circ h_2$ is autoregressive.*

In summary, Theorem B.3 and Theorem B.4 achieve results comparable to Theorem B.2. Their main contribution is to extend counterfactual equivalence to the setting of invertible latent causal models, where, in addition to the standard counterfactual reasoning, an unknown mixing function maps latent causal variables to the observable space, thereby advancing the development of causal representation learning.

**Spectial cases of Corollary 5.4**   Several existing results on counterfactual identifiability for SCMs under triangular monotonicity can be regarded as special cases of Corollary 5.4.

(Lu et al., 2020) applied counterfactual reasoning in reinforcement learning, where Theorem 1 states that counterfactual outcomes are identifiable when noise is independent of states and actions, and the state transition function is strictly monotonically increasing:

**Theorem B.5** (restated from Lu et al. 2020, Theorem 1). *Assume $S_{t+1} = f(S_t, A_t, U_{t+1})$, where $U_{t+1} \perp\!\!\!\perp (S_t, A_t)$, and $S_t, A_t, U_t$ denote the state, action, and noise at time step $t$ in reinforcement learning, respectively. Suppose $f$ is smooth and strictly monotonically increasing for fixed $S_t$ and $A_t$. Given the observed values $S_t = s_t$, $A_t = a$, and $S_{t+1} = s_{t+1}$, the counterfactual outcome $S_{t+1, A_t = a'} \mid S_t = s_t, A_t = a, S_{t+1} = s_{t+1}$ is identifiable for a counterfactual action $A_t = a'$.*

Here, the independence of noise from states and actions corresponds to the Markovianity assumption. The function $f$, being smooth and strictly monotonically increasing, represents a univariate TM mapping. The causal ordering assumption is implicitly given by time step $t$, and the observational distribution assumption is implicitly encoded in the observed values $S_t = s_t$, $A_t = a$, and $S_{t+1} = s_{t+1}$. Thus, all four assumptions in Corollary 5.4 are satisfied, making the SCM described in Theorem B.5 $\sim_{EI}$-identifiable. By Theorem 3.2, this SCM is also $\sim_{\mathcal{L}_3}$-identifiable, thereby ensuring the identifiability of counterfactual outcomes.

(Nasr-Esfahany et al., 2023) further distilled the properties described in Theorem B.5 to uncover results more aligned with general SCM formalism. Specifically, they examined a special causal mechanism known as BGM:

**Theorem B.6** (restated from Nasr-Esfahany et al. 2023, Theorem 5.1). *Suppose $f(\mathbf{V}_{pa(i)}, U_i)$ is a BGM, meaning that for each realization $\mathbf{v}_{pa(i)}$ of $\mathbf{V}_{pa(i)}$, $f(\mathbf{v}_{pa(i)}, \cdot)$ is invertible. Then $f$ is counterfactually identifiable given $P_{\mathbf{V}_{pa(i)}, V_i}$ if: (i) the Markovianity assumption holds, i.e., $U_i \perp\!\!\!\perp \mathbf{V}_{pa(i)}$; and (ii) for all $\mathbf{v}_{pa(i)}$, $f(\mathbf{v}_{pa(i)}, \cdot)$ is either strictly monotonically increasing or strictly monotonically decreasing.*

Compared to Theorem B.5, Theorem B.6 extends the analysis to strictly monotonically decreasing functions, which still conform to the definition of TM mappings. The assumption of causal ordering is implicitly provided by the known $pa(i)$. Lastly, while the original theorem is restricted to the univariate case, Corollary 5.4 does not impose such a restriction. Thus, Corollary 5.4 accompanied by Theorem 3.2 fully subsumes Theorem B.6.

(Scetbon et al., 2024) adopted an alternative fixed-point-based representation of SCMs, referring to the identifiability of counterfactual outcomes as weakly partial recovery of fixed-point SCMs, and proposed the following theorem:

**Theorem B.7** (restated from Scetbon et al. 2024, Theorem 2.14). *Assume that $\Omega_{V_i}$ and $\Omega_{U_i}$ are subsets of $\mathbb{R}$ for each $i \in \mathcal{I}$, $P_{\mathbf{U}}$ is absolutely continuous with a continuous density, and each $U_i$ is mutually independent. Let $[\mathcal{M}]^{INV}$ denote the set of*

*SCMs satisfying the observational distribution $P_\mathbf{V}$, causal ordering $\leq$, and causal mechanisms $\boldsymbol{f}$ that are $C^1$ and such that $f(\mathbf{v}_{pa(i)}, \cdot)$ is bijective for all $i \in \mathcal{I}$. Then $[\mathcal{M}]^{INV} \neq \emptyset$, and all models in this set are consistent on $(\boldsymbol{f}_{[\mathbf{x}]} \circ \boldsymbol{f}^{-1})_\sharp P_\mathbf{V}$ for any intervention $\mathbf{x} \in \Omega_\mathbf{X}$, where $\mathbf{X} \subseteq \mathbf{V}$.*

Here, $(\boldsymbol{f}_{[\mathbf{x}]} \circ \boldsymbol{f}^{-1})_\sharp P_\mathbf{V}$ represents the pushforward form of counterfactual outcomes defined via the *abduction-action-prediction* framework, indicating that the theorem essentially discusses counterfactual identifiability. The mutual independence of $U_i$ corresponds to the Markovianity assumption. The causal ordering is represented by a permutation matrix $\mathbf{P}$ in the original paper, and the observational distribution is described as a pushforward of the exogenous distribution.

Compared to Theorem B.6, the key difference lies in the lack of an explicit assumption that $f(\mathbf{v}_{\mathrm{pa}(i)}, \cdot)$ is strictly monotonic. Instead, this property is implicitly ensured through $C^1$ smoothness. Specifically, leveraging $C^1$ smoothness and (Scetbon et al., 2024, Lemma G.8), they indirectly proved that the partial derivative of $f(\mathbf{v}_{\mathrm{pa}(i)}, \cdot)$ with respect to $u_i$ is always strictly positive or strictly negative, which implies consistency of the strict monotonicity of $f(\mathbf{v}_{\mathrm{pa}(i)}, \cdot)$.

There are other generalizations of the monotonicity assumption for counterfactual identifiability. (Wu et al., 2025) employs the rank preservation assumption to identify the counterfactual outcome, which relaxes the strict monotonicity assumption. As discussed in the main text, strict monotonicity essentially corresponds to a lexicographical order; rank preservation can therefore be viewed as a redefinition of this lexicographical order, thereby generalizing the monotonicity assumption.

### B.4. Neural SCMs

As for the related work on neural TM-SCM, we focus here on methods designed for counterfactual inference tasks that leverage proxy models constructed by mimicking the structure of SCMs and parameterizing them with neural networks, which we refer to as neural SCMs.

**Neural SCMs without proven identifiability**    A series of works connect SCMs with various deep generative models, utilizing the inverse processes of these models to address the tractability issues in counterfactual inference. DSCM (Pawlowski et al., 2020) employs normalizing flows and variational inference to propose a general standardized framework for constructing SCMs using neural networks, enabling tractable inference of exogenous variables for counterfactual reasoning. Diff-SCM (Sanchez & Tsaftaris, 2022) introduces diffusion models tailored for counterfactual inference on high-dimensional data, where latent variables are inferred through the diffusion process, and gradients are intervened upon during the reverse diffusion process, ultimately applied to counterfactual image generation. VACA (Sánchez-Martin et al., 2022) parameterizes SCMs using a variational graph autoencoder (VGAE), distinguishing itself by directly modeling the global causal mechanism rather than individual conditional mechanisms and encoding variable dependencies within the SCM using graph neural networks. While these models introduce the concept of mimicking SCMs to construct proxy models and enable counterfactual reasoning, they do not explicitly demonstrate the reliability of their inference results, i.e., whether the counterfactual outcomes are identifiable.

**Neural SCMs proven to be identifiable for counterfactual effects**    A series of studies address the identifiability of counterfactual effects in proxy SCMs by leveraging deep generative models. CVAE-SCM (Karimi et al., 2020) employs conditional variational autoencoders (CVAE) to model conditional mechanisms and proves that under the Markov assumption, certain types of counterfactual queries are identifiable solely from the observational distribution. Consequently, the constructed model ensures consistency for these counterfactual queries. MLE-NCM (Xia et al., 2021) systematically explores the identifiability of connecting neural networks to SCMs and demonstrates that, in discrete cases and with sufficiently expressive neural networks, proxy models constructed from causal graphs and observational distributions exhibit duality in $\mathcal{L}_2$ causal queries. That is, a causal query is identifiable on the proxy model if and only if it is identifiable on the causal graph. iVGAE (Zečević et al., 2021) extends MLE-NCM by showing that such duality also holds when using VGAE. GAN-NCM (Xia et al., 2023) further generalizes MLE-NCM to $\mathcal{L}_3$ causal queries, proving that proxy models constructed from causal graphs and interventional distributions exhibit duality in the identifiability of counterfactual queries. The model is trained using generative adversarial networks (GANs).

**Neural SCMs with empirically identifiable counterfactual outcomes**    A series of studies primarily focus on proving model identifiability and empirically demonstrate the identifiability of counterfactual outcomes through experiments. CAREFL (Khemakhem et al., 2021) connects causal inference with autoregressive normalizing flows (ANF), utilizing affine transformations primarily for cause-effect identification tasks in causal discovery and proposing methods for counterfactual reasoning. CausalNF (Javaloy et al., 2023) extends this approach to other types of ANF transformations and proves model

identifiability in latent encodings. CCNF (Zhou et al., 2025) further introduces partial causal transformations to enhance inference performance. CFM (Khoa Le et al., 2025) generalizes this line of work from discrete normalizing flows to continuous normalizing flows, employing flow matching for modeling. While these studies demonstrate the identifiability of counterfactual outcomes on synthetic datasets, the connection between latent encoding identifiability (a problem in representation learning) and counterfactual identifiability (a problem in causal quantities) remains unclear. Proving under which conditions the former implies the latter is one of the key contributions of this work.

**Neural SCMs proven to be identifiable for counterfactual outcomes** A series of studies have established the identifiability of counterfactual outcomes under specified assumptions. BGM (Nasr-Esfahany et al., 2023) investigates the identifiability of causal mechanisms for counterfactual outcomes under the bijection assumption and proposes three identification methods, ultimately using conditional normalizing flows to construct a proxy SCM to empirically support their findings. FiP (Scetbon et al., 2024) formalizes SCMs and counterfactual distributions using fixed points, framing the identifiability problem for counterfactual outcomes as the weak partial recovery problem. They identify a condition for recovering counterfactual distributions, though their experiments return to additive noise models based on attention mechanisms while employing optimal transport to model noise distributions. DCM (Chao et al., 2024) assumes that the exogenous distribution of the true latent SCM is uniform, focusing on encoder-decoder models, and proposes two identification methods as extensions of (Nasr-Esfahany et al., 2023, Proposition 6.2). These methods rely on diffusion models to construct proxy SCMs. As shown in Appendix B.3, the first two studies are special cases of TM-SCM, while the last study leverages (Nasr-Esfahany et al., 2023, Proposition 6.2), which corresponds to a special instance of Theorem 3.2 in this work that considers only one single causal mechanism. Although these studies demonstrate the identifiability of counterfactual outcomes, these results represent only a specific subset of counterfactual statements. In contrast, a key contribution of this work is proving that the theoretical foundations extend beyond the identifiability of counterfactual outcomes to encompass the more general and stronger $\sim_{\mathcal{L}_3}$-identifiability.

## C. Neural TM-SCM

In this section, we present the implementation details of various neural TM-SCMs used in the experiments.

### C.1. Vectorization

In this subsection, we will specifically address the implementation issues related to vectorization and de-vectorization, particularly how to switch between the symbolic form of endogenous or exogenous values (primarily used for reasoning on $\mathcal{M}$) and the vectorized form (primarily used for reasoning on the solution mapping $\Gamma$) when a TM-SCM is given.

**Vectorization** Suppose we aim to vectorize an endogenous or exogenous value $\mathbf{x}$, resulting in its vectorized form $\mathbf{P}_\iota(\mathbf{x})$. Since these values originate from TM-SCM, their indices are fully aligned, and thus we do not distinguish between them here. Vectorization requires a predefined re-indexing map $\iota$. According to the definition provided in the main text, this only additionally requires a causal order $\leq$ in the TM-SCM. The causal order can be assumed to be directly provided, as in the neural TM-SCM problem setting described in the main text. For the ground truth TM-SCM, any causal order can be obtained through topological sorting. Subsequently, we can construct such a map $\iota$ based on Lemma A.19, defined as

$$\iota(i, j) = j + \sum_{k \in \mathrm{pr}(i)} d_k.$$

From an implementation perspective, the collection of values $(\mathbf{x}_i)_{i \in \mathcal{I}}$, representing the symbolic form of $\mathbf{x}$, can be expressed as a dictionary where keys are $\mathcal{I}$ and values are vectors $\mathbf{x}_i$. Vectorization in this context involves concatenating the vectors $\mathbf{x}_i$ corresponding to different causal variables in the causal order $\leq$, resulting in

$$\mathbf{P}_\iota(\mathbf{x}) = \bigoplus_{\substack{|\mathrm{pr}(i)|=k}}^{k=n} \mathbf{x}_i,$$

where $\mathrm{pr}(i)$ denotes the causal prefix of $i \in \mathcal{I}$ under $\leq$, and $\oplus$ represents vector concatenation.

**De-vectorization** Suppose we aim to de-vectorize an endogenous or exogenous value $\mathbf{P}_\iota(\mathbf{x})$ to obtain the symbolic form of $\mathbf{x}$. This operation requires knowledge of the vectorization mapping $\iota$, which is provided during the vectorization process.

According to Lemma A.19 and $(\mathbf{P}_\iota)^{-1}$, each causal variable is given by

$$\mathbf{x}_i = (\mathbf{P}_\iota(\mathbf{x}))_{j \in l_i : l_i + d_i},$$

where $l_i = \sum_{k \in \mathrm{pr}(i)} d_k$. From an implementation perspective, this is equivalent to selecting the slice corresponding to $l_i : l_i + d_i$ from the vector $\mathbf{P}_\iota(\mathbf{x})$.

## C.2. Exogenous Distribution

In the main text, we design the use of normalizing flows to model the exogenous distribution. However, in practical experiments, to demonstrate the insensitivity of our theory to the type of exogenous distribution, we additionally consider other types of exogenous distributions. Recall that, according to the Markovianity assumption, it is required that $P_{\mathbf{U}_\theta} = \prod_{i \in \mathcal{I}} P_{U_{i,\theta}}$, so the following discussion focuses on each $P_{U_{i,\theta}}$.

**Standard normal distribution**   A simple modeling approach is to use a standard normal distribution, such that

$$\log p_{U_{i,\theta}}(u_i) = \log \mathcal{N}(u_i \,|\, \mathbf{0}, \mathbf{I}),$$

where $\mathcal{N}$ denotes the density of a Gaussian distribution. However, this modeling approach may not provide sufficient expressive power. For example, in the causal mechanisms of DNME, the diagonal structure results in non-interacting dimensions, and using a standard normal distribution in such cases would enforce independence between these dimensions. This limitation is reflected in the experiments presented in the Appendix D.

**Normalizing flow**   As introduced in the main text, the log-likelihood for each $P_{U_{i,\theta}}$ is given by

$$\log p_{U_{i,\theta}}(u_i) = \log p_{Z_{i,\theta}}(\mathcal{T}_{i,\theta}^{-1}(u_i)) + \log \left| \det \mathbf{J}_{\mathcal{T}_{i,\theta}^{-1}}(u_i) \right|,$$

where $\mathcal{T}_{i,\theta} : \mathcal{Z}_i \to \mathcal{U}_i$ is a masked autoregressive flow (MAF), and $\left| \det \mathbf{J}_{\mathcal{T}_{i,\theta}^{-1}}(u_i) \right|$ denotes the Jacobian determinant at $\mathbf{u}$.

**Gaussian mixture model**   The log-likelihood for each $P_{U_{i,\theta}}$ is given by

$$\log p_{U_{i,\theta}}(u_i) = \log \left( \sum_{k=1}^{K} w_{k,i,\theta} \cdot \mathcal{N}(u_i \,|\, \boldsymbol{\mu}_{k,i,\theta}, \boldsymbol{\Sigma}_{k,i,\theta}) \right),$$

where $\mathcal{N}$ the density of a Gaussian distribution with mean $\boldsymbol{\mu}_{k,i,\theta}$ and covariance matrix $\boldsymbol{\Sigma}_{k,i,\theta}$, both parameterized by neural networks. Additionally, $\sum_{k=1}^{K} w_{k,i,\theta} = 1$ for any $i \in \mathcal{I}$.

In the implementation, these parameterized distribution models are provided by the normalizing flow library *Zuko* (Rozet et al., 2024).

## C.3. DNME

We begin with the simplest DNME prototype to demonstrate how to construct the model while enabling the derivation of inverses and log absolute Jacobian determinants. Specifically, we leverage the normalizing flow library *Zuko*, which provides a unified framework and standard for building normalizing flow models. Thus, the constructed neural TM-SCM is effectively transformed into a normalizing flow.

Recall that the symbolic form of DNME is given by

$$f_{i,\theta}(\mathbf{v}_{\mathrm{pa}(i)}, \mathbf{u}_i) = \mathbf{b}_{i,\theta}(\mathbf{v}_{\mathrm{pa}(i)}) + \mathbf{a}_{i,\theta}(\mathbf{v}_{\mathrm{pa}(i)}) \odot \mathbf{u}_i.$$

To transform a DNME-type neural TM-SCM into a normalizing flow, we interpret the causal mechanism $f_{i,\theta}$ as a discrete transformation within a normalizing flow. This can be represented using a coupling transformation $\boldsymbol{c}$ combined with an affine transformation $\boldsymbol{a}$:

$$f_{i,\theta}(\mathbf{v}_{\mathcal{I}\setminus\{i\}}, \mathbf{u}_i) = \boldsymbol{c}(\mathbf{v}_{\mathcal{I}\setminus\{i\}}, \mathbf{u}_i; \boldsymbol{a}(\cdot; m_\theta(\mathbf{v}_{\mathrm{pa}(i)}))),$$
$$\boldsymbol{c}(\mathbf{x}, \mathbf{y}; f) = (\mathbf{x}, f(\mathbf{y})),$$
$$\boldsymbol{a}(\mathbf{x}; \mathbf{a}, \mathbf{b}) = \mathbf{b} + \exp(\mathbf{a}) \odot \mathbf{x},$$

where the affine transformation satisfies the strict positivity requirement of $\mathbf{a}_{i,\theta}(\mathbf{v}_{\mathrm{pa}(i)})$, and the coupling transformation ensures that $f_{i,\theta}$ depends only on $\mathbf{v}_{\mathrm{pa}(i)}$ and $\mathbf{u}_i$, thereby constraining the causal structure. The function $m_\theta$ is a multi-layer perceptron (MLP) that provides the parameters required for the affine transformation.

Correspondingly, the inverse of the causal mechanism, $(f_{i,\theta}(\mathbf{v}_{\mathcal{I}\setminus\{i\}}))^{-1}(\mathbf{v}_i)$, can be expressed as:

$$f_{i,\theta}(\mathbf{v}_{\mathcal{I}\setminus\{i\}}, \mathbf{v}_i) = \boldsymbol{c}^{-1}(\mathbf{v}_{\mathcal{I}\setminus\{i\}}, \mathbf{v}_i; \boldsymbol{a}(\cdot; m_\theta(\mathbf{v}_{\mathrm{pa}(i)}))),$$
$$\boldsymbol{c}^{-1}(\mathbf{x}, \mathbf{y}; f) = (\mathbf{x}, f^{-1}(\mathbf{y})),$$
$$\boldsymbol{a}^{-1}(\mathbf{x}; \mathbf{a}, \mathbf{b}) = (\mathbf{x} - \mathbf{b}) \oslash \exp(\mathbf{a}),$$

where $\oslash$ denotes component-wise division. The log absolute Jacobian determinants for the coupling and affine transformations are, respectively:

$$\log|\det \mathbf{J}_{\boldsymbol{c}}(\mathbf{x}, \mathbf{y}; f)| = \log|\det \mathbf{J}_f(\mathbf{y})|,$$
$$\log|\det \mathbf{J}_{\boldsymbol{a}}(\mathbf{x}; \mathbf{a}, \mathbf{b})| = \sum_{j \in 1:d_i} a_j,$$

where $a_j$ is the $j$-th component of $\mathbf{a}$.

The solution mapping $\boldsymbol{\Gamma}_\theta$ is then composed according to the causal order $\leq$ as $f_{s,\theta} \circ \cdots \circ f_{r,\theta}$, where $\mathrm{pr}(s) = 0$ and $\mathrm{pr}^*(r) = |\mathcal{I}|$. This forms a discrete normalizing flow, and since each transformation supports forward, inverse derivations, and log absolute Jacobian determinant calculations, $\boldsymbol{\Gamma}_\theta$ inherits these properties. Consequently, the log-likelihood of endogenous value $\mathbf{v}^{(i)}$ is computed as:

$$p_{\mathbf{V}_\theta}(\mathbf{v}^{(i)}) = \log p_{\mathbf{U}_\theta}(\boldsymbol{\Gamma}_\theta^{-1}(\mathbf{v}^{(i)})) + \log\left|\det \mathbf{J}_{\boldsymbol{\Gamma}_\theta^{-1}}(\mathbf{v}^{(i)})\right|,$$

supporting a maximum likelihood optimization process. Here, $\log p_{\mathbf{U}_\theta}$ is the log-likelihood function of the exogenous distribution, discussed in Appendix C.2.

### C.4. TNME

The TNME improves upon the DNME, with its formal representation defined as

$$f_{i,\theta}(\mathbf{v}_{\mathrm{pa}(i)}, \mathbf{u}_i) = \mathbf{b}_{i,\theta}(\mathbf{v}_{\mathrm{pa}(i)}) + \left(\mathbf{A}_{i,\theta}(\mathbf{v}_{\mathrm{pa}(i)})\right)\mathbf{u}_i^\mathsf{T}.$$

Specifically, the component-wise affine transformation $\boldsymbol{a}$ is replaced by a more sophisticated lower triangular affine transformation $\mathcal{A}$, such that

$$f_{i,\theta}(\mathbf{v}_{\mathcal{I}\setminus\{i\}}, \mathbf{u}_i) = \boldsymbol{c}(\mathbf{v}_{\mathcal{I}\setminus\{i\}}, \mathbf{u}_i; \mathcal{A}(\cdot; m_\theta(\mathbf{v}_{\mathrm{pa}(i)}))),$$
$$\mathcal{A}(\mathbf{x}; \mathbf{A}, \mathbf{b}) = \mathbf{b} + \mathrm{tril}(\exp(\mathbf{A} + \epsilon))\mathbf{x}^\mathsf{T},$$

where the lower triangular affine transformation takes a matrix $\mathbf{A}$ and a vector $\mathbf{b}$ as parameters. The $\exp$ function constrains the matrix to be positive, ensuring monotonicity, while the tril function and a small positive scalar $\epsilon$ enforce the matrix to be a full-rank lower triangular matrix, guaranteeing invertibility. The MLP $m_\theta$ provides the parameters required for the lower triangular affine transformation.

Similar to DNME, the inverse of the lower triangular affine transformation is given by

$$\mathcal{A}^{-1}(\mathbf{x}; \mathbf{a}, \mathbf{b}) = (\mathrm{tril}(\exp(\mathbf{A} + \epsilon)))^{-1}(\mathbf{x} - \mathbf{b})^\mathsf{T},$$

where $(\mathrm{tril}(\exp(\mathbf{A} + \epsilon)))^{-1}$ is the inverse of $\mathrm{tril}(\exp(\mathbf{A} + \epsilon))$. During implementation, instead of explicitly computing the matrix inverse, the inverse operation is treated as solving the linear system

$$\mathrm{tril}(\exp(\mathbf{A} + \epsilon))\,\mathcal{A}^{-1}(\mathbf{x}; \mathbf{a}, \mathbf{b}) = (\mathbf{x} - \mathbf{b})^\mathsf{T},$$

where $\mathcal{A}^{-1}(\mathbf{x}; \mathbf{a}, \mathbf{b})$ is treated as the unknown.

Due to the properties of lower triangular matrices, the log-determinant of the Jacobian of the lower triangular affine transformation is straightforward to compute:

$$\log|\det \mathbf{J}_{\boldsymbol{a}}(\mathbf{x}; \mathbf{a}, \mathbf{b})| = \sum_{j \in 1:d_i} (a_{j,j} + \epsilon),$$

where $a_{j,j}$ is the $(j, j)$-th element of the matrix $\mathbf{A}$. Subsequently, the construction of the mapping $\boldsymbol{\Gamma}_\theta$ follows the same principles as in DNME.

## C.5. CMSM

The CMSM models the re-indexed solution mapping $\mathbf{P}_\iota \circ \boldsymbol{\Gamma}_\theta$ as a sequence of discrete transformations,

$$\mathbf{P}_\iota \circ \boldsymbol{\Gamma}_\theta = \mathbf{T}_{1,\theta} \circ \cdots \circ \mathbf{T}_{n,\theta},$$

where $n$ is required to be no less than the diameter of the causal graph, as stated in (Javaloy et al., 2023). Each $\mathbf{T}_{j,\theta}$ is modeled as an autoregressive transformation $\boldsymbol{t}$, with an affine transformation $\boldsymbol{a}$ as the univariate transformation, such that

$$\mathbf{x}_j = \boldsymbol{t}(\mathbf{x}_{j-1}; \boldsymbol{a}),$$

where $\mathbf{x}_0 = \mathbf{P}_\iota(\mathbf{u})$ represents the vectorized exogenous values, and $\mathbf{x}_n = \mathbf{P}_\iota(\mathbf{v})$ represents the vectorized endogenous values. The autoregressive transformation $\boldsymbol{t}$ ensures that each component $x_{j,i}$ of $\mathbf{x}_j$ is expressed as

$$\mathbf{x}_{j,i} = \boldsymbol{a}(x_{j-1,i}; m_{j,\theta}(\mathbf{x}_{j-1,1:i-1})),$$

where the MLP $m_{j,\theta}$ provides the parameters required for the affine transformation. The autoregressive transformation $\boldsymbol{t}$ is a triangular transformation, and its inverse is computed in the same manner as triangular transformations, as detailed in Lemma A.15. Furthermore, its log absolute Jacobian determinant is given by

$$\log|\det \mathbf{J}_{\boldsymbol{t}}(\mathbf{x}; f)| = \sum_{i \in 1:\sum_{j \in \mathcal{I}} d_i} \log|\det \mathbf{J}_f(x_i)|,$$

as the Jacobian matrix of a triangular transformation is lower triangular. Consequently, the construction of the re-indexed solution mapping $\mathbf{P}_\iota \circ \boldsymbol{\Gamma}_\theta$ follows the same principles as DNME.

## C.6. TVSM

For the ODE

$$\begin{cases} \mathrm{d}\boldsymbol{x}(t) = \boldsymbol{v}(\boldsymbol{x}(t), t) \, \mathrm{d}t, \\ \boldsymbol{x}(0) = \mathbf{x}_0, \end{cases}$$

we denote the flow as $\mathcal{T}(\mathbf{x}_0, t) = \boldsymbol{x}(t)$, where $\boldsymbol{v} : \mathbb{R}^d \times [0, 1] \to \mathbb{R}^d$ is a time-dependent velocity field, and $\boldsymbol{x} : [0, 1] \to \mathbb{R}^d$ is the solution to the ODE. A flow $\mathcal{T}(\cdot, t)$ pushes forward the distribution at $t = 0$ to that at time $t$, such that $P_{t,\mathbf{x}} = (\mathcal{T}(\cdot, t))_\sharp P_{0,\mathbf{x}}$. According to Lemma A.21, if the velocity field $\boldsymbol{v}$ is a Lipschitz continuous triangular mapping for any $t$, then the flow $\mathcal{T}(\cdot, t)$ is always a TMI mapping.

In implementation, we adopt a method similar to (Chen et al., 2018) for modeling velocity fields, introducing a mask to ensure that $\boldsymbol{v}$ is a triangular mapping:

$$\boldsymbol{v}(\mathbf{x}, t) = \tanh((\mathbf{W} \odot \mathbf{M})\mathbf{x}^\mathsf{T} + \mathbf{b})(\mathbf{U} \odot \mathrm{sigmoid}(\mathbf{G}) \odot \mathbf{M})^\mathsf{T},$$

where

$$(\mathbf{W}, \mathbf{b}, \mathbf{U}, \mathbf{G}) = m_\theta(t),$$

$\mathbf{W}, \mathbf{U}, \mathbf{G}$ are matrices, and $\mathbf{b}$ is a vector, all parameterized by an MLP with $t$ as input. $\mathbf{M}$ is a lower triangular mask.

Unlike previous approaches that model the re-indexed solution mapping using composite discrete transforms, we model $\mathbf{P}_\iota \circ \boldsymbol{\Gamma}_\theta = \mathrm{odeint}(\boldsymbol{v}, 0, 1)$, and its inverse $\boldsymbol{\Gamma}_\theta^{-1} \circ (\mathbf{P}_\iota)^{-1} = \mathrm{odeint}(\boldsymbol{v}, 1, 0)$, where odeint represents an IVP solver from the

---

**Algorithm 1** Pseudo Potential Response for TM-SCM

---

**Input:** a TM-SCM $\mathcal{M}$, exogenous value $\mathbf{u}$, intervened value $\mathbf{x}$, vectorization $\iota$ under a causal order $\leq$ of $\mathcal{M}$.
**Output:** potential response $\mathbf{V}_{\mathcal{M}_{[\mathbf{x}]}}(\mathbf{u})$
$\mathbf{u}^* \leftarrow \mathbf{u}, \mathbf{v}^* \leftarrow \boldsymbol{\Gamma}(\mathbf{u}), D \leftarrow \sum_{i \in \mathcal{I}} d_i$ {Initialize potential response}
**for** $k = 1$ **to** $D$ **do**
   $(i, j) \leftarrow \iota^{-1}(k), \mathbf{v}' \leftarrow \mathbf{v}^*$
   **if** $i \in I_{\mathbf{x}}$ **then**
      $(\mathbf{v}')_{i,j} \leftarrow (\mathbf{x})_{i,j}$ {Do-intervention}
   **end if**
   $(\mathbf{u}^*)_{\iota^{-1}[1:k]} \leftarrow (\boldsymbol{\Gamma}^{-1}(\mathbf{v}'))_{\iota^{-1}[1:k]}$ {Find exogenous value for fully explaining the prefix part}
   $(\mathbf{v}^*)_{\iota^{-1}[k:D]} \leftarrow (\boldsymbol{\Gamma}(\mathbf{u}^*))_{\iota^{-1}[k:D]}$ {Assume the suffix part is not intervened, and update the potential response}
**end for**
**return** $\mathbf{v}^*$

---

*torchdiffeq* library (Chen, 2018). For log-likelihood computation, we employ the unbiased log-likelihood estimation method in (Grathwohl et al., 2019):

$$\log p_{\mathbf{V}_\theta}(\mathbf{v}^{(i)}) = \log p_{\mathbf{U}_\theta}((\boldsymbol{\Gamma}_\theta^{-1} \circ (\mathbf{P}_\iota)^{-1})(\mathbf{v}^{(i)})) - \mathbb{E}_{\epsilon \sim P_\epsilon}\left[\int_0^1 \epsilon^{\mathsf{T}}\left(\partial \boldsymbol{v}_{s,\theta}/\partial(\boldsymbol{\Gamma}_{s,\theta}^{-1} \circ (\mathbf{P}_\iota)^{-1})\right)\epsilon\, \mathrm{d}s\right],$$

where $\boldsymbol{\Gamma}_{s,\theta}^{-1} \circ (\mathbf{P}_\iota)^{-1}$ corresponds to $\mathrm{odeint}(\boldsymbol{v}, 1, s)$, and $P_\epsilon$ is the standard normal distribution.

### C.7. Inference

In the experiments, the trained neural TM-SCM infers counterfactual outcomes through a process referred to as *pseudo potential response* (Algorithm 1). Direct intervention on causal mechanisms is challenging for TM-SCMs constructed from solution mappings (e.g., CMSM and TVSM), as they lack explicit causal mechanisms and thus cannot directly fix the values of causal parent variables.

The term *pseudo potential response* arises from the characteristic of Algorithm 1 performing indirect interventions. TM-SCM, as a BSCM, possesses a fully supported exogenous variable space. Combined with TM-SCM's well-defined iteration order, the algorithm aims to inversely identify an exogenous value that perfectly explains the intervention and the ancestors already determined, rather than directly intervening on endogenous variables. This extends the intervention algorithm for single intervened variables in (Javaloy et al., 2023) to support arbitrary intervened variables and random vectors.

To ensure correctness, we prove that the output of the algorithm is consistent with the potential response $\mathbf{V}_{\mathcal{M}_{[\mathbf{x}]}}(\mathbf{u})$:

**Theorem C.1** (Correctness of Pseudo Potential Response)**.** *The result returned by the* pseudo potential response *algorithm,* $\mathbf{v}^*$*, equals* $\mathbf{V}_{\mathcal{M}_{[\mathbf{x}]}}(\mathbf{u})$*.*

*Proof.* For any $t \in 1{:}D$, where $D = \sum_{i \in \mathcal{I}} d_i$, suppose $t \in l_i{:}l_i + d_i$. By the definition of the potential response, we have

$$(\mathbf{V}_{\mathcal{M}_{[\mathbf{x}]}}(\mathbf{u}))_{\iota^{-1}(t)} = (\boldsymbol{\Gamma}_{[\mathbf{x}]}(\mathbf{u}))_{\iota^{-1}(t)} \overset{\text{Lemma A.2}}{=\!=\!=\!=} (\tilde{f}_{i[\mathbf{x}]}(\mathbf{u}_{\mathrm{an}^*(i)}))_{\iota^{-1}(t)}$$

$$\overset{\text{Definition A.1}}{=\!=\!=\!=} \begin{cases} (f_i((\tilde{f}_{k[\mathbf{x}]}(\mathbf{u}_{\mathrm{an}^*(k)}))_{k \in \mathrm{pa}(i)}, u_i))_{t-l_i} & i \notin I_{\mathbf{x}}, \\ (\mathbf{x})_{\iota^{-1}(t)} & i \in I_{\mathbf{x}}. \end{cases}$$

Suppose $(\mathbf{v}^*)_{\iota^{-1}[1:t-1]} = (\tilde{f}_{i[\mathbf{x}]}(\mathbf{u}_{\mathrm{an}^*(i)}))_{\iota^{-1}[1:t-1]}$. Consider the $t$-th iteration. Let $\mathbf{v}' = \mathbf{v}^*$ initially. If $i \in I_{\mathbf{x}}$, then $(\mathbf{v}')_{\iota^{-1}(t)} = \mathbf{x}_{\iota^{-1}(t)}$. By the property of $\boldsymbol{\Gamma}$ as a TM mapping, the component at index $d$ remains unchanged:

$$(\mathbf{v}^*)_{\iota^{-1}(t)} = (\mathbf{v}')_{\iota^{-1}(t)} = \mathbf{x}_{\iota^{-1}(t)} = (\tilde{f}_{i[\mathbf{x}]}(\mathbf{u}_{\mathrm{an}^*(i)}))_{\iota^{-1}(t)}.$$

If $i \notin I_{\mathbf{x}}$, then by the $(t-1)$-th iteration:

$$(\mathbf{v}^*)_{\iota^{-1}(t)} = (\boldsymbol{\Gamma}(\mathbf{u}^*))_{\iota^{-1}(t)},$$

where for each $k \in 1{:}t-1$:

$$(\tilde{f}_j(\mathbf{u}_{\mathrm{an}^*(j)}^*))_{\iota^{-1}(k)} = (\tilde{f}_{j[\mathbf{x}]}(\mathbf{u}_{\mathrm{an}^*(j)}))_{\iota^{-1}(k)},$$

assuming $k \in l_j : l_j + d_j$. Thus,

$$(\mathbf{v}^*)_{\iota^{-1}[t]} = (f_i((\tilde{f}_k(\mathbf{u}^*_{\mathrm{an}^*(k)}))_{k \in \mathrm{pa}(i)}, u_i))_{t-l_i} = (f_i((\tilde{f}_{k[\mathbf{x}]}(\mathbf{u}_{\mathrm{an}^*(k)}))_{k \in \mathrm{pa}(i)}, u_i))_{t-l_i} = (\tilde{f}_{i[\mathbf{x}]}(\mathbf{u}_{\mathrm{an}^*(i)}))_{\iota^{-1}(t)}.$$

Hence, after the $t$-th iteration, $(\mathbf{v}^*)_{\iota^{-1}[t]} = (\tilde{f}_{i[\mathbf{x}]}(\mathbf{u}_{\mathrm{an}^*(i)}))_{\iota^{-1}[t]}$ holds universally. Since $\mathbf{v}^*_{\iota[1:t-1]}$ remains unchanged,

$$(\mathbf{v}^*)_{\iota^{-1}[1:t]} = (\tilde{f}_{i[\mathbf{x}]}(\mathbf{u}_{\mathrm{an}^*(i)}))_{\iota^{-1}[1:t]}.$$

This holds for any $t \in 1:D$. Thus, at $t = D$,

$$\mathbf{v}^* = \tilde{f}_{i[\mathbf{x}]}(\mathbf{u}_{\mathrm{an}^*(i)})$$

where $\mathrm{an}^*(i) = \mathcal{I}$, implying $\mathbf{v}^* = \mathbf{V}_{\mathcal{M}_{[\mathbf{x}]}}(\mathbf{u})$. $\qquad\square$

# D. Experiments

## D.1. Datasets

**TM-SCM-SYM** The dataset collection includes four small synthetic datasets: BARBELL, STAIR, FORK, and BACKDOOR. Each causal variable is represented as a random vector with dimensionality ranging from 1 to 8, with up to four causal variables in total. The exogenous distributions are either mutually independent standard normal distributions or multivariate normal distributions with a Markov structure. The causal mechanisms between variables are defined as manually designed symbolic TM mappings. The specific generation mechanisms for these four synthetic datasets are detailed as follows.

- **BARBELL**: Includes 2 causal variables, $\mathbf{x}$ and $\mathbf{y}$, each with 8 dimensions. Their causal relationship follows the simplest binary cause-effect structure, $\mathbf{x} \to \mathbf{y}$, with the causal mechanisms represented as:

$$x_i = (f_x(\mathbf{u}_x))_i = \begin{cases} s(1.8\, u_{x,i}) - 1 & i = 1, \\ l\left(\sum_{j=1}^{i-1} x_j, u_{x,i}\right) & 1 < i \le 5, \\ 0.3\, u_{x,i} + s\left(\sum_{j=1}^{i-1} x_j + 1\right) - 1 & 5 < i \le 7, \\ \mathrm{CDF}^{-1}\left(-s\left(\left(1.3\left(\sum_{j=1}^{i-2} x_j\right) + \sum_{j=1}^{i-1} x_j\right)/3 + 1\right) + 2, 0.6, u_{x,i}\right) & i = 8. \end{cases}$$

$$y_i = (f_y(\mathbf{x}, \mathbf{u}_y))_i = \begin{cases} l\left(s(1.8\, u_{y,i}) - 1, x_i\right) & i = 1, \\ l\left(l\left(\sum_{j=1}^{i-1} y_j, u_{x,i}\right), x_i\right) & 1 < i \le 5, \\ \mathrm{CDF}^{-1}\left(-s\left(\left(1.3\left(\sum_{j=1}^{i-1} y_j\right) + l\left(\sum_{j=1}^{i-1} y_j, x_i\right)\right)/3 + 1\right) + 2, 0.6, u_{y,i}\right) & 5 < i \le 7, \\ 0.3\, u_{y,i} - 0.5\left(\sum_{j=1}^{i-1} y_j\right) + s\left(\sum_{j=1}^{i-2} y_j + 1\right) - 1 & i = 8. \end{cases}$$

The function $s$ represents the softplus function defined as $s(x) = \log(1 + \exp(x))$. The binary function $l$ is defined as $l(x, y) = s(x+1) + s(0.5+y) - 3$. The function $\mathrm{CDF}^{-1}(\mu, b, x)$ denotes the quantile function of the Laplace distribution at $x$, with location $\mu$ and scale $b$. The exogenous distribution follows a standard normal distribution.

- **STAIR** Consider 3 causal variables $\mathbf{x}$, $\mathbf{y}$, and $\mathbf{z}$ with dimensions 1, 2, and 3, respectively. Their causal structure follows a chain: $\mathbf{x} \to \mathbf{y} \to \mathbf{z}$. All causal mechanisms are linear and are represented as:

$$x_i = (f_x(\mathbf{u}_x))_i = 0.75\, u_{x,i}.$$

$$y_i = (f_y(\mathbf{x}, \mathbf{u}_y))_i = \begin{cases} 10\, x_i + u_{y,i} & i = 1, \\ 0.25\, x_{i-1} - 0.5\, u_{y,i} + 2 & i = 2. \end{cases}$$

$$z_i = (f_z(\mathbf{x}, \mathbf{u}_z))_i = \begin{cases} -0.5\, y_i + u_{z,i} - 4 & i = 1, \\ 5\, y_i - 1.5\, u_{z,i} & i = 2, \\ y_{i-1} + 2\, u_{z,i} - 0.5 & i = 3. \end{cases}$$

and the exogenous distribution follows a standard normal distribution.

- **FORK** Consider 4 causal variables $\mathbf{x}$, $\mathbf{y}$, $\mathbf{z}$, and $\mathbf{w}$, each of dimension 2. The causal structure among them includes a collider structure $\mathbf{x} \to \mathbf{z} \leftarrow \mathbf{y}$ as well as a direct causal relationship $\mathbf{z} \to \mathbf{w}$. All causal mechanisms are additive and can

be expressed as:

$$x_i = (f_x(\mathbf{u}_x))_i = \begin{cases} u_{x,i} & i = 1, \\ u_{x,i-1} + 0.25\,u_{x,i} & i = 2. \end{cases}$$

$$y_i = (f_y(\mathbf{x}, \mathbf{u}_y))_i = \begin{cases} u_{y,i} & i = 1, \\ -0.5\,u_{y,i-1} + u_{y,i} & i = 2. \end{cases}$$

$$z_i = (f_z(\mathbf{x}, \mathbf{y}, \mathbf{u}_z))_i = \begin{cases} \dfrac{1}{1 + \exp(-x_1 - y_2)} - y_1^2 + 0.5\,u_{z,i} & i = 1, \\ \dfrac{1}{1 + \exp(-y_1 - x_2)} - x_1^2 + 0.5(u_{z,i-1} + u_{z,i}) & i = 2. \end{cases}$$

$$w_i = (f_w(\mathbf{z}, \mathbf{u}_w))_i = \begin{cases} \dfrac{20}{1 + \exp(0.5\,z_1^2 - z_2)} + u_{w,i} & i = 1, \\ \dfrac{20}{1 + \exp(0.5\,z_2^2 - z_1)} + 0.25\,u_{w,i-1} - u_{w,i} & i = 2. \end{cases}$$

and the exogenous distribution also follows a standard normal distribution.

- **BACKDOOR** The causal graph involves 4 causal variables, $\mathbf{x}$, $\mathbf{y}$, $\mathbf{z}$, and $\mathbf{w}$, each of dimension 4. The causal structure includes a direct causal link $\mathbf{y} \to \mathbf{w}$, an indirect causal path $\mathbf{y} \to \mathbf{z} \to \mathbf{w}$, and a backdoor path $\mathbf{y} \leftarrow \mathbf{x} \to \mathbf{w}$. The causal mechanisms are represented as:

$$x_i = (f_x(\mathbf{u}_x))_i = \begin{cases} \tanh(u_{x,i}) & i = 1, \\ \mathrm{sigmoid}(-u_{x,i}) & i = 2, \\ x_{i-1} - \tanh(u_{x,i}) & i = 3, \\ 1 + \mathrm{sigmoid}(u_{x,i}) & i = 4. \end{cases}$$

$$y_i = (f_y(\mathbf{x}, \mathbf{u}_y))_i = \begin{cases} \tanh(x_1 + u_{y,i}) & i = 1, \\ \mathrm{sigmoid}(x_2 - u_{y,i}) & i = 2, \\ y_{i-1} - \tanh(u_{y,i}) & i = 3, \\ \mathbf{x}_{1:i-1} \cdot (\mathrm{softmax}(\mathbf{y}_{1:i-1}))^{\mathsf{T}} + \mathrm{sigmoid}(u_{y,i}) & i = 4. \end{cases}$$

$$z_i = (f_z(\mathbf{y}, \mathbf{u}_z))_i = \begin{cases} \mathrm{elu}(y_1 - u_{z,i}) & i = 1, \\ \mathrm{leaky\_relu}(y_2 + u_{z,i}) & i = 2, \\ z_{i-1} + \mathrm{elu}(u_{z,i}) & i = 3, \\ \mathbf{y}_{1:i-1} \cdot (\mathrm{softmin}(\mathbf{z}_{1:i-1}))^{\mathsf{T}} + \mathrm{leaky\_relu}(u_{z,i}) & i = 4. \end{cases}$$

$$w_i = (f_w(\mathbf{x}, \mathbf{y}, \mathbf{z}, \mathbf{u}_w))_i = \begin{cases} (x_i \oplus y_i \oplus z_i) \cdot (\mathrm{softmax}(x_i \oplus y_i \oplus z_i))^{\mathsf{T}} + \mathrm{elu}(z_i - u_{w,i}) & i = 1, \\ (w_{i-1} \oplus u_{w,i-1}) \cdot (\mathrm{softmin}(w_{i-1} \oplus u_{w,i-1}))^{\mathsf{T}} + \mathrm{leaky\_relu}(y_i + u_{w,i}) & i = 2, \\ (\mathbf{x}_{1:i} \oplus \mathbf{y}_{1:i} \oplus \mathbf{z}_{1:i}) \cdot (\mathrm{softmin}(\mathbf{x}_{1:i} \oplus \mathbf{y}_{1:i} \oplus \mathbf{z}_{1:i}))^{\mathsf{T}} + x_{i-1} + \mathrm{elu}(u_{w,i}) & i = 3, \\ (\mathbf{w}_{1:i-1} \oplus \mathbf{u}_{w,1:i-1}) \cdot (\mathrm{softmax}(\mathbf{w}_{1:i-1} \oplus \mathbf{u}_{w,1:i-1}))^{\mathsf{T}} + \mathrm{leaky\_relu}(u_{w,i}) & i = 4. \end{cases}$$

where tanh, sigmoid, elu, and leaky_relu are widely used activation functions, ensuring monotonicity and continuity. The exogenous distribution follows a multivariate normal distribution, with location $\mathbf{0}$ and covariance matrix:

$$\begin{bmatrix} 1 & 0.8 & 0.3 & 0.2 \\ 0.8 & 1 & 0.4 & 0.1 \\ 0.3 & 0.4 & 1 & 0.6 \\ 0.2 & 0.1 & 0.6 & 1 \end{bmatrix}.$$

**ER-DIAG-50 and ER-TRIL-50**  Each dataset collection consists of 50 randomly synthesized datasets, where each dataset contains 3–8 causal variables, and the dimensionality of each causal variable is randomly chosen between 1 and 8. The causal structure among the variables is modeled as an Erdős-Rényi graph, the exogenous distribution is randomly generated as a multivariate normal distribution, and the causal mechanisms are constructed using randomly generated TM mappings. The details for generating these components are described as follows.

- **Variables** The number of causal variables, $n$, is sampled from a discrete uniform distribution Unif$\{3, 8\}$, and these variables are denoted as $\mathbf{x}_1, \ldots, \mathbf{x}_n$.
- **Dimensions** The dimension $d_i$ of a causal variable $\mathbf{x}_i$ is sampled from a discrete uniform distribution Unif$\{1, 8\}$, with the $j$-th component denoted as $x_{i,j}$.
- **Graphs** An Erdős-Rényi graph is generated over the $n$ causal variables with a probability of $0.5$ for each edge, ensuring the graph is directed and acyclic. Specifically, we construct an $n \times n$ adjacency matrix $\mathbf{A}$, where each element $a_{i,j}$ is sampled from a continuous uniform distribution Unif$(0, 1)$. If $a_{i,j} > 0.5$ and $i < j$, the edge $\mathbf{x}_i \to \mathbf{x}_j$ is included.
- **Exogenous distributions** For each causal variable, a multivariate normal distribution of dimension $m$ is randomly generated to represent its corresponding exogenous distribution. By Markovianity, exogenous samples are formed by independently sampling from these distributions. Specifically, the parameters of the multivariate normal distribution are generated by sampling a mean vector $\boldsymbol{\mu}$ and a matrix $\mathbf{C}$ from the standard normal distribution. The covariance matrix is computed as $\boldsymbol{\Sigma} = \mathbf{C}^\mathsf{T}\mathbf{C} + \mathbf{I}$, ensuring positive definiteness, where $\mathbf{I}$ is the identity matrix.
- **Causal mechanisms** The randomly generated causal mechanisms are ensured to be TM mappings. For ER-DIAG-50, the causal mechanism is represented as

$$(f_i(\mathbf{x}_{\mathrm{pa}(i)}, \mathbf{u}_i))_j = h_{i,j}(g_i(\mathbf{x}_{\mathrm{pa}(i)}), u_{i,j}),$$

for $j \in 1\!:\!d_i$. For ER-TRIL-50, it is expressed as

$$(f_i(\mathbf{x}_{\mathrm{pa}(i)}, \mathbf{u}_i))_j = h_{i,j}((f_i(\mathbf{x}_{\mathrm{pa}(i)}, \mathbf{u}_i))_{j-1}, u_{i,j}),$$

where for $j = 1$, $(f_i(\mathbf{x}_{\mathrm{pa}(i)}, \mathbf{u}_i))_{j-1} = g_i(\mathbf{x}_{\mathrm{pa}(i)})$.

The function $g_i$ is a randomly generated continuous function $\mathbb{R}^{\sum_{k \in \mathrm{pa}(i)} d_k} \to \mathbb{R}$. To enhance its complexity, it is expressed as a combination of sinusoidal and segmented terms:

$$g(\mathbf{x}) = \underbrace{\sum_{i=1}^{m} c_i \cdot \sin(\langle \mathbf{w}_i, \mathbf{x} \rangle + \phi_i)}_{\text{sinusoidal term}} + \underbrace{s_{j^*(\mathbf{x})} \cdot \|\mathbf{x} - \boldsymbol{\mu}_{j^*(\mathbf{x})}\|}_{\text{segmented term}},$$

where $j^*(\mathbf{x}) = \arg\min_{1 \le j \le n} \|\mathbf{x} - \boldsymbol{\mu}_j\|$, and $c_i, \mathbf{w}_i, \phi_i, s_j, \boldsymbol{\mu}_j$ are random parameters. The number of sinusoidal and segmented terms is $n = m = 3$.

The function $h_{i,j}$ is a randomly generated monotonic function $\mathbb{R}^2 \to \mathbb{R}$, ensuring monotonicity with respect to the second dimension. It is represented in a linear form:

$$h_{i,j}(x_1, x_2) = s_1 \cdot x_1 + s_2 \cdot x_2 + b,$$

where the bias term $b$ and scaling coefficients $s_1, s_2$ are random parameters. The signs of $s_1, s_2$ are not constrained to be positive, indicating that the resulting mapping is not necessarily a TMI mapping but rather a more general TM mapping.

## D.2. Metrics

We employed three metrics to evaluate the performance of the trained models.

**OBS$_{\mathbf{WD}}$** This metric measures the Wasserstein distance between the observational distribution derived from the model and the ground truth, evaluating how well the trained model fits the observational distribution. The Wasserstein distance is computed using the *geomloss* library (Feydy et al., 2019), which calculates the un-biased Sinkhorn divergence with a blur parameter of $0.05$ and a cost function of $C(x, y) = \frac{1}{2}\|x - y\|_2^2$.

**CTF$_{\mathbf{RMSE}}$** This metric measures the error between the counterfactual results inferred by the model and the ground truth, assessing the accuracy of the trained model in counterfactual reasoning and reflecting the model's consistency in answering counterfactual queries. We use the root mean square error (RMSE), which retains the same unit as the original values, making it more interpretable for evaluating the accuracy of counterfactual reasoning.

**CTF$_{\mathbf{WD}}$** This is a metric unreported in the main text but included as a supplementary evaluation in the appendix. It measures the Wasserstein distance between the interventional distribution and the ground truth in an average sense. Since

the observed values in the ground truth of counterfactual datasets are sampled from the observational distribution, these samples, when considering only the counterfactual outcomes, correspond to the distribution expressed as

$$\mathbb{E}_{\mathbf{x} \sim P_{\mathbf{X}}} \left[ \int_{\Omega_{\mathbf{V}}} P_{\mathbf{V}_{[\mathbf{x}]} | \mathbf{V}}(\cdot, \mathbf{v}) P_{\mathbf{V}}(\mathrm{d}\mathbf{v}) \right] = \mathbb{E}_{\mathbf{x} \sim P_{\mathbf{X}}} \left[ P_{\mathbf{V}_{[\mathbf{x}]}} \right],$$

where $P_{\mathbf{X}}$ denotes the prior distribution of intervention values provided by the counterfactual dataset, and $P_{\mathbf{V}_{[\mathbf{x}]}}$ represents the interventional distribution under intervention $\mathbf{x}$.

### D.3. Execution

This subsection details the preprocessing, training, and testing procedures of the experiments to ensure reproducibility.

**Preprocessing**   During the initial execution, following the settings described in Appendix D.1, each synthetic dataset is divided into three splits: training, validation, and test datasets. The training dataset comprises observational data with a sample size of 20,000, directly sampled from the exogenous distribution, with exogenous samples propagated through the synthesized TM-SCM to derive endogenous observational values. The validation and test datasets, each containing 2,000 samples, consist of counterfactual data, providing observations, interventions, and counterfactual outcomes. These counterfactual datasets are synthesized in three steps: (i) Similar to the observational dataset, exogenous values and their corresponding endogenous observations are sampled. (ii) Intervened values are sampled anew, where the intervened variables $\mathbf{X}$ satisfy certain conditions termed interventionally meaningful, and the intervened values adhere to the observational distribution. (iii) The counterfactual outcomes are derived using the exogenous values from (i) and the intervened values from (ii) to compute the potential responses.

The notion of *interventionally meaningful* refers to the following three conditions of an intervention set $\mathbf{X}$: (i) For any $X \in \mathbf{X}$, $X$ must have at least one child. (ii) If $X, Y \in \mathbf{X}$ and $X$ is a parent of $Y$, then $X$ and $Y$ must have at least one common child. (iii) $\mathbf{X}$ is non-empty. In implementation, indices $I_{\mathbf{X}}$ are sampled to satisfy these conditions, and a mask is used to indicate whether a variable has been intervened upon.

Finally, all dimensions in the datasets are standardized. To ensure consistency, the mean and variance from the observational dataset are used for standardization, including for the intervened values and counterfactual outcomes in the counterfactual datasets. Since standardization merely involves component-wise scaling and shifting, the triangular monotonicity property of TM-SCM is preserved.

**Training**   All neural network components in the models, as reported in Appendix C, are configured as MLPs with 2 hidden layers and a width of 128. The outputs of these MLPs include an additional dimension representing the encoding length, and the parameters subsequently derived are averaged over this encoding. This design enhances the expressiveness of the MLPs while maintaining fairness in model parameters. Specifically, before each run, we perform binary search to find an appropriate encoding length such that the total number of model parameters does not exceed 1M. All models are constructed according to this standard, ensuring that the parameter counts are comparable across models.

For experiments on TM-SCM-SYM, we train for 100 epochs with a batch size of 64. Validation is performed every $k$ training steps, where $k$ grows exponentially such that the interval after the $t$-th validation is $k = \lfloor \gamma^t \rfloor$. We set $\gamma = 1.25$. The validation results are used to plot the scatter relationship between $\mathrm{OBS_{WD}}$ and $\mathrm{CTF_{RMSE}}$. All experiments under this configuration are conducted with 10 different random seeds.

For experiments on ER-DIAG-50 and ER-TRIL-50, we train for 50 epochs with a batch size of 64, performing validation after every epoch. All experiments are run once on their respective 50 pre-generated random datasets.

For all the experiments mentioned above, the optimizer used is Adam with a learning rate of 0.001 and weight decay of 0.01 as a regularization term. The model weights corresponding to the epoch with the lowest $\mathrm{CTF_{RMSE}}$ on the validation set are saved for testing.

**Testing**   The test set is used to evaluate the model performance reported in the tables. Specifically, for $\mathrm{OBS_{WD}}$, we utilize the part of the test set labeled as observed values. According to the description in the preprocessing section, these samples satisfy the ground truth observational distribution. For $\mathrm{CTF_{RMSE}}$ and $\mathrm{CTF_{WD}}$, we use the parts of the test set labeled as observed values, intervened values, and counterfactual outcomes. The observed and intervened values are input into

Algorithm 1 to estimate the counterfactual outcomes, and the ground truth counterfactual outcomes are used to compute these counterfactual metrics. During validation and testing, in cases where certain model configurations exhibit instability at specific positions, leading to numerical overflow, the corresponding test cases are masked and ignored.

## D.4. Full Results

**Ablation on TM-SCM-SYM** We report below the complete ablation study results on the TM-SCM-SYM dataset as discussed in the main text. First, we present the full version of the scatter plots shown in the main text, as illustrated in Figure 4, which are obtained from the validation results of experiments conducted under 10 different random seeds.

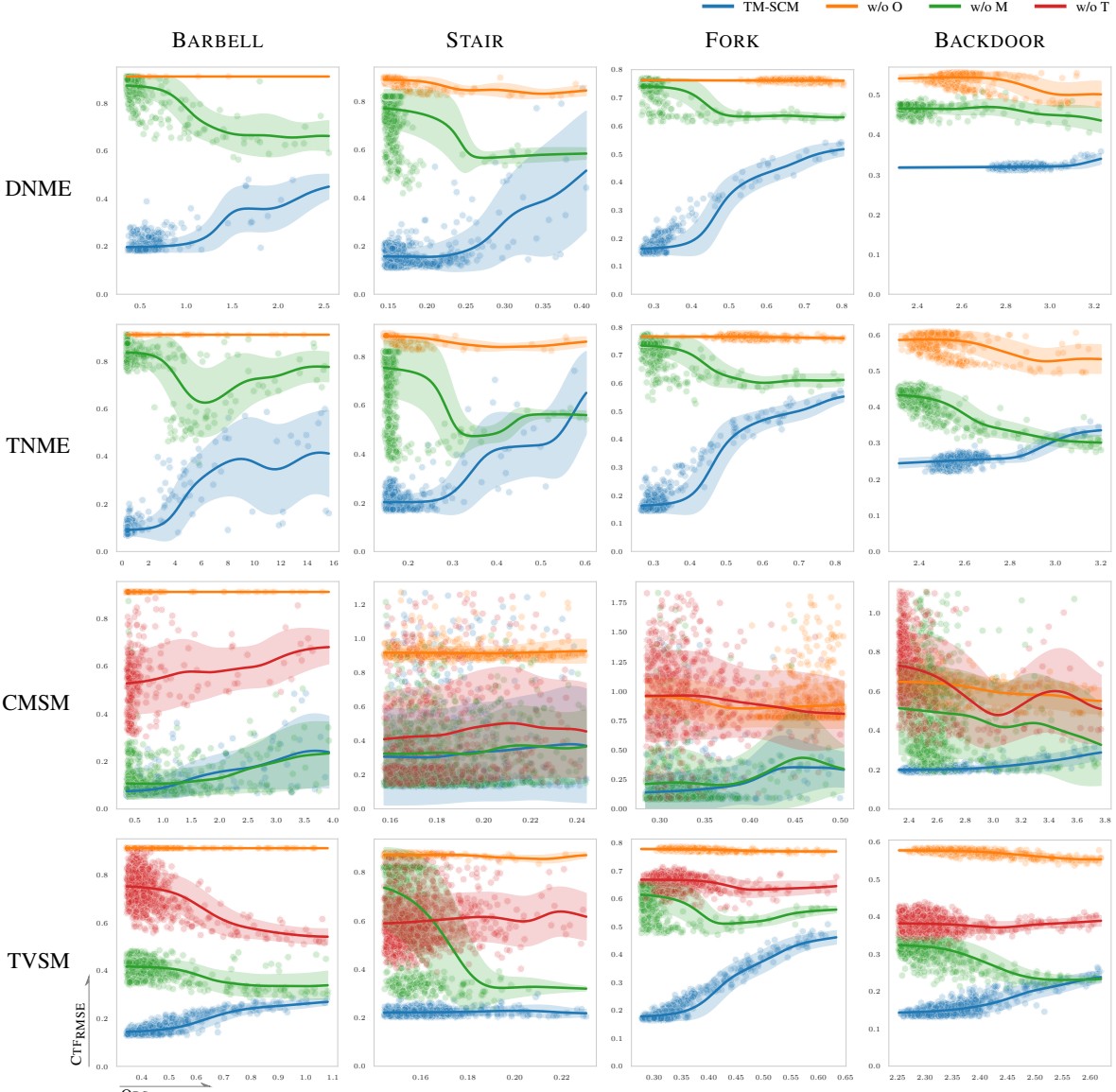

*Figure 4.* Ablation results of neural TM-SCMs on TM-SCM-SYM. Colored curves depict sliding-window regressions, with shaded areas showing 95% CI. To improve the readability of the plot scales, outliers below the 0.01 quantile and above the 0.99 quantile were removed. Rows represent models (DNME, TNME, CMSM, TVSM), and columns represent datasets (BARBELL, STAIR, FORK, BACKDOOR).

Across all combinations of models and datasets, the non-ablated neural TM-SCM (blue) consistently achieves lower $\text{CTF}_{\text{RMSE}}$, demonstrating superior performance in counterfactual consistency. Specifically, as training progresses, the horizontal axis $\text{OBS}_{\text{WD}}$ gradually decreases, indicating an improved fit of the model to the observed distribution. It can be observed that, except for certain settings where the model lacks sufficient expressiveness, $\text{OBS}_{\text{WD}}$ is reduced to a similar

*Table 3.* Ablation results of neural TM-SCM on BARBELL and STAIR. The values shown are the means of 10 experiments with different random seeds, with the subscript representing the 95% CI. The best-performing results are highlighted in bold.

| METHOD | | BARBELL | | | STAIR | | |
|---|---|---|---|---|---|---|---|
| | | $\text{OBS}_{\text{WD}}$ | $\text{CTF}_{\text{RMSE}}$ | $\text{CTF}_{\text{WD}}$ | $\text{OBS}_{\text{WD}}$ | $\text{CTF}_{\text{RMSE}}$ | $\text{CTF}_{\text{WD}}$ |
| DNME | - | $\mathbf{0.42}_{\pm\mathbf{0.03}}$ | $\mathbf{0.18}_{\pm\mathbf{0.00}}$ | $\mathbf{0.20}_{\pm\mathbf{0.01}}$ | $0.18_{\pm0.03}$ | $\mathbf{0.10}_{\pm\mathbf{0.02}}$ | $\mathbf{0.03}_{\pm\mathbf{0.01}}$ |
| | w/o O | $4.44_{\pm2.74}$ | $0.94_{\pm0.00}$ | $3.89_{\pm0.00}$ | $31.54_{\pm52.74}$ | $0.80_{\pm0.00}$ | $0.61_{\pm0.00}$ |
| | w/o M | $5.38_{\pm0.12}$ | $0.50_{\pm0.02}$ | $1.53_{\pm0.10}$ | $\mathbf{0.17}_{\pm\mathbf{0.01}}$ | $0.47_{\pm0.03}$ | $0.48_{\pm0.04}$ |
| TNME | - | $\mathbf{0.39}_{\pm\mathbf{0.03}}$ | $\mathbf{0.07}_{\pm\mathbf{0.00}}$ | $\mathbf{0.04}_{\pm\mathbf{0.00}}$ | $\mathbf{0.16}_{\pm\mathbf{0.01}}$ | $0.16_{\pm0.01}$ | $\mathbf{0.07}_{\pm\mathbf{0.01}}$ |
| | w/o O | $18.26_{\pm1.02}$ | $0.94_{\pm0.00}$ | $3.89_{\pm0.00}$ | $32.65_{\pm52.99}$ | $0.80_{\pm0.00}$ | $0.61_{\pm0.00}$ |
| | w/o M | $7.50_{\pm6.76}$ | $0.52_{\pm0.05}$ | $1.55_{\pm0.23}$ | $0.18_{\pm0.01}$ | $0.39_{\pm0.02}$ | $0.35_{\pm0.04}$ |
| CMSM | - | $0.41_{\pm0.02}$ | $\mathbf{0.05}_{\pm\mathbf{0.01}}$ | $\mathbf{0.02}_{\pm\mathbf{0.01}}$ | $\mathbf{0.17}_{\pm\mathbf{0.01}}$ | $0.24_{\pm0.23}$ | $0.30_{\pm0.53}$ |
| | w/o O | $\mathbf{0.39}_{\pm\mathbf{0.04}}$ | $0.94_{\pm0.00}$ | $3.89_{\pm0.00}$ | $15.05_{\pm38.19}$ | $0.83_{\pm0.02}$ | $0.60_{\pm0.03}$ |
| | w/o M | $1.01_{\pm0.46}$ | $0.07_{\pm0.01}$ | $0.04_{\pm0.01}$ | $\mathbf{0.17}_{\pm\mathbf{0.01}}$ | $\mathbf{0.13}_{\pm\mathbf{0.04}}$ | $\mathbf{0.05}_{\pm\mathbf{0.03}}$ |
| | w/o T | $1.82_{\pm2.12}$ | $0.98_{\pm1.22}$ | $25.38_{\pm55.39}$ | $0.19_{\pm0.03}$ | $14.67_{\pm32.28}$ | $0.51_{\pm0.70}$ |
| TVSM | - | $\mathbf{0.37}_{\pm\mathbf{0.02}}$ | $\mathbf{0.10}_{\pm\mathbf{0.00}}$ | $\mathbf{0.03}_{\pm\mathbf{0.01}}$ | $\mathbf{0.17}_{\pm\mathbf{0.02}}$ | $0.16_{\pm0.01}$ | $\mathbf{0.08}_{\pm\mathbf{0.01}}$ |
| | w/o O | $0.48_{\pm0.15}$ | $0.94_{\pm0.00}$ | $3.89_{\pm0.00}$ | $32.15_{\pm53.73}$ | $0.80_{\pm0.00}$ | $0.61_{\pm0.00}$ |
| | w/o M | $1.15_{\pm0.21}$ | $0.29_{\pm0.01}$ | $0.48_{\pm0.03}$ | $\mathbf{0.17}_{\pm\mathbf{0.01}}$ | $0.28_{\pm0.01}$ | $0.21_{\pm0.02}$ |
| | w/o T | $1.33_{\pm0.21}$ | $0.51_{\pm0.01}$ | $1.59_{\pm0.07}$ | $20.56_{\pm46.17}$ | $0.43_{\pm0.02}$ | $0.44_{\pm0.05}$ |

*Table 4.* Ablation results of neural TM-SCM on FORK and BACKDOOR. The values shown are the means of 10 experiments with different random seeds, with the subscript representing the 95% CI. The best-performing results are highlighted in bold.

| METHOD | | FORK | | | BACKDOOR | | |
|---|---|---|---|---|---|---|---|
| | | $\text{OBS}_{\text{WD}}$ | $\text{CTF}_{\text{RMSE}}$ | $\text{CTF}_{\text{WD}}$ | $\text{OBS}_{\text{WD}}$ | $\text{CTF}_{\text{RMSE}}$ | $\text{CTF}_{\text{WD}}$ |
| DNME | - | $\mathbf{0.28}_{\pm\mathbf{0.01}}$ | $\mathbf{0.14}_{\pm\mathbf{0.00}}$ | $\mathbf{0.08}_{\pm\mathbf{0.00}}$ | $\mathbf{2.83}_{\pm\mathbf{0.03}}$ | $\mathbf{0.27}_{\pm\mathbf{0.00}}$ | $\mathbf{0.59}_{\pm\mathbf{0.02}}$ |
| | w/o O | $2.15_{\pm0.10}$ | $0.74_{\pm0.00}$ | $0.57_{\pm0.00}$ | $4.79_{\pm0.26}$ | $0.46_{\pm0.00}$ | $1.45_{\pm0.00}$ |
| | w/o M | $1.61_{\pm2.86}$ | $0.62_{\pm0.01}$ | $0.47_{\pm0.01}$ | $4.36_{\pm0.12}$ | $0.34_{\pm0.01}$ | $0.88_{\pm0.05}$ |
| TNME | - | $\mathbf{0.27}_{\pm\mathbf{0.01}}$ | $\mathbf{0.14}_{\pm\mathbf{0.00}}$ | $\mathbf{0.07}_{\pm\mathbf{0.00}}$ | $\mathbf{2.47}_{\pm\mathbf{0.04}}$ | $\mathbf{0.16}_{\pm\mathbf{0.00}}$ | $\mathbf{0.20}_{\pm\mathbf{0.01}}$ |
| | w/o O | $3.88_{\pm0.24}$ | $0.74_{\pm0.00}$ | $0.57_{\pm0.00}$ | $9.20_{\pm0.24}$ | $0.46_{\pm0.00}$ | $1.45_{\pm0.00}$ |
| | w/o M | $15.31_{\pm32.54}$ | $0.58_{\pm0.02}$ | $0.48_{\pm0.01}$ | $3.20_{\pm0.04}$ | $0.28_{\pm0.01}$ | $0.63_{\pm0.03}$ |
| CMSM | - | $0.30_{\pm0.02}$ | $\mathbf{0.08}_{\pm\mathbf{0.01}}$ | $\mathbf{0.02}_{\pm\mathbf{0.00}}$ | $\mathbf{2.53}_{\pm\mathbf{0.09}}$ | $\mathbf{0.15}_{\pm\mathbf{0.00}}$ | $\mathbf{0.19}_{\pm\mathbf{0.01}}$ |
| | w/o O | $1.56_{\pm2.43}$ | $0.80_{\pm0.05}$ | $0.71_{\pm0.26}$ | $6.35_{\pm3.18}$ | $0.47_{\pm0.00}$ | $1.50_{\pm0.01}$ |
| | w/o M | $\mathbf{0.30}_{\pm\mathbf{0.02}}$ | $0.08_{\pm0.00}$ | $0.02_{\pm0.00}$ | $2.68_{\pm0.09}$ | $0.17_{\pm0.02}$ | $0.23_{\pm0.05}$ |
| | w/o T | $0.37_{\pm0.06}$ | $0.63_{\pm0.24}$ | $1.02_{\pm0.56}$ | $7.91_{\pm11.03}$ | $0.37_{\pm0.02}$ | $1.04_{\pm0.11}$ |
| TVSM | - | $\mathbf{0.29}_{\pm\mathbf{0.01}}$ | $\mathbf{0.16}_{\pm\mathbf{0.00}}$ | $\mathbf{0.08}_{\pm\mathbf{0.00}}$ | $\mathbf{2.29}_{\pm\mathbf{0.02}}$ | $\mathbf{0.12}_{\pm\mathbf{0.01}}$ | $\mathbf{0.11}_{\pm\mathbf{0.01}}$ |
| | w/o O | $2.10_{\pm0.08}$ | $0.74_{\pm0.00}$ | $0.57_{\pm0.00}$ | $5.79_{\pm0.05}$ | $0.46_{\pm0.00}$ | $1.45_{\pm0.00}$ |
| | w/o M | $0.32_{\pm0.03}$ | $0.48_{\pm0.01}$ | $0.31_{\pm0.01}$ | $2.53_{\pm0.07}$ | $0.21_{\pm0.01}$ | $0.36_{\pm0.03}$ |
| | w/o T | $0.46_{\pm0.03}$ | $0.61_{\pm0.02}$ | $0.43_{\pm0.02}$ | $2.64_{\pm0.44}$ | $0.34_{\pm0.01}$ | $0.87_{\pm0.05}$ |

level across most configurations. However, the final performance on the vertical axis $\text{CTF}_{\text{RMSE}}$ varies across different ablation modes, highlighting the negative impact of the ablated components on counterfactual consistency.

In these ablation modes, w/o O (yellow) represents the case where the causal order is reversed. Although $\text{OBS}_{\text{WD}}$ decreases in almost all settings, $\text{CTF}_{\text{RMSE}}$ remains nearly unchanged. This highlights the significant role of the causal order assumption $\mathcal{A}_{\leq}$ in ensuring counterfactual consistency. w/o M (green) represents the case where the exogenous distribution is non-Markovian. In most settings, as $\text{OBS}_{\text{WD}}$ decreases, $\text{CTF}_{\text{RMSE}}$ instead increases. This indicates that counterfactual consistency cannot converge to a lower level as expected under non-Markovian conditions, underscoring the necessity of the Markovian assumption $\mathcal{A}_{\text{M}}$ for counterfactual consistency. w/o T (red) represents the case where the solution mapping of the constructed SCM is not triangular (allowed only for CMSM and TVSM). In these settings, as $\text{OBS}_{\text{WD}}$ decreases, $\text{CTF}_{\text{RMSE}}$ diverges. This demonstrates the critical impact of the assumption that SCMs are TM-SCMs, $\mathcal{A}_{\text{TM-SCM}}$, on counterfactual consistency. Only in the no ablation mode (blue), where all three assumptions in Corollary 5.4—$\mathcal{A}_{\text{TM-SCM}}$, $\mathcal{A}_{\text{M}}$, and $\mathcal{A}_{\leq}$—are satisfied, does counterfactual consistency improve (as evidenced by decreasing $\text{CTF}_{\text{RMSE}}$) as $\mathcal{A}_{P_V}$ is gradually satisfied (indicated by

*Table 5.* Ablation results of neural TM-SCM on TM-SCM-SYM for exogenous distributions. The values shown are the means of 10 experiments with different random seeds, with the subscript representing the 95% CI.

| METHOD | DIST | BARBELL $\text{OBS}_{\text{WD}}$ | BARBELL $\text{CTF}_{\text{RMSE}}$ | STAIR $\text{OBS}_{\text{WD}}$ | STAIR $\text{CTF}_{\text{RMSE}}$ | FORK $\text{OBS}_{\text{WD}}$ | FORK $\text{CTF}_{\text{RMSE}}$ | BACKDOOR $\text{OBS}_{\text{WD}}$ | BACKDOOR $\text{CTF}_{\text{RMSE}}$ |
|---|---|---|---|---|---|---|---|---|---|
| DNME | N | $6.20_{\pm0.06}$ | $0.19_{\pm0.00}$ | $0.17_{\pm0.00}$ | $0.11_{\pm0.00}$ | $1.14_{\pm0.03}$ | $0.29_{\pm0.00}$ | $4.57_{\pm0.02}$ | $0.20_{\pm0.00}$ |
|  | GMM | $1.94_{\pm0.43}$ | $0.19_{\pm0.01}$ | $0.16_{\pm0.01}$ | $0.12_{\pm0.01}$ | $0.37_{\pm0.04}$ | $0.17_{\pm0.01}$ | $3.41_{\pm0.23}$ | $0.22_{\pm0.01}$ |
|  | NF | $0.42_{\pm0.03}$ | $0.18_{\pm0.00}$ | $0.18_{\pm0.03}$ | $0.10_{\pm0.02}$ | $0.28_{\pm0.01}$ | $0.14_{\pm0.00}$ | $2.83_{\pm0.03}$ | $0.27_{\pm0.00}$ |
| TNME | N | $5.88_{\pm0.07}$ | $0.10_{\pm0.00}$ | $0.17_{\pm0.00}$ | $0.10_{\pm0.00}$ | $1.12_{\pm0.04}$ | $0.26_{\pm0.00}$ | $3.12_{\pm0.02}$ | $0.15_{\pm0.00}$ |
|  | GMM | $3.65_{\pm3.46}$ | $0.09_{\pm0.01}$ | $0.17_{\pm0.02}$ | $0.14_{\pm0.01}$ | $0.34_{\pm0.03}$ | $0.16_{\pm0.01}$ | $2.82_{\pm0.22}$ | $0.15_{\pm0.00}$ |
|  | NF | $0.39_{\pm0.03}$ | $0.07_{\pm0.00}$ | $0.16_{\pm0.01}$ | $0.16_{\pm0.01}$ | $0.27_{\pm0.01}$ | $0.14_{\pm0.00}$ | $2.47_{\pm0.04}$ | $0.16_{\pm0.00}$ |
| CMSM | N | $2.96_{\pm2.68}$ | $0.06_{\pm0.01}$ | $0.17_{\pm0.02}$ | $0.12_{\pm0.02}$ | $0.32_{\pm0.02}$ | $0.07_{\pm0.01}$ | $2.86_{\pm0.41}$ | $0.13_{\pm0.00}$ |
|  | GMM | $1.99_{\pm2.13}$ | $0.06_{\pm0.00}$ | $0.19_{\pm0.02}$ | $0.11_{\pm0.02}$ | $0.30_{\pm0.02}$ | $0.08_{\pm0.00}$ | $15.93_{\pm27.12}$ | $0.13_{\pm0.00}$ |
|  | NF | $0.41_{\pm0.02}$ | $0.05_{\pm0.01}$ | $0.17_{\pm0.01}$ | $0.24_{\pm0.23}$ | $0.30_{\pm0.02}$ | $0.08_{\pm0.01}$ | $2.53_{\pm0.09}$ | $0.15_{\pm0.00}$ |
| TVSM | N | $0.34_{\pm0.01}$ | $0.09_{\pm0.00}$ | $0.16_{\pm0.00}$ | $0.14_{\pm0.00}$ | $0.31_{\pm0.01}$ | $0.17_{\pm0.00}$ | $2.29_{\pm0.01}$ | $0.10_{\pm0.00}$ |
|  | GMM | $0.53_{\pm0.10}$ | $0.09_{\pm0.00}$ | $0.16_{\pm0.01}$ | $0.15_{\pm0.01}$ | $0.29_{\pm0.01}$ | $0.15_{\pm0.00}$ | $2.27_{\pm0.01}$ | $0.10_{\pm0.01}$ |
|  | NF | $0.37_{\pm0.02}$ | $0.10_{\pm0.00}$ | $0.17_{\pm0.02}$ | $0.16_{\pm0.01}$ | $0.29_{\pm0.01}$ | $0.16_{\pm0.00}$ | $2.29_{\pm0.02}$ | $0.12_{\pm0.01}$ |

decreasing $\text{OBS}_{\text{WD}}$).

Furthermore, we can analyze the neural TM-SCM methods under different constructions (without ablations) row by row. It can be observed that DNME and TNME exhibit similar performance but perform slightly worse on the BACKDOOR dataset. This may be attributed to their limited expressive capacity for the observed distribution, as reflected by the $\text{OBS}_{\text{WD}}$, which did not decrease as expected to the same level as other models. CMSM appears to be less stable during training, as evidenced by the widely dispersed scatter points across all settings. This could imply that the parameters of CMSM undergo significant changes during training, resulting in unstable performance. In contrast, TVSM demonstrates more concentrated scatter points, indicating higher stability. Moreover, it achieves a lower $\text{OBS}_{\text{WD}}$ on the BACKDOOR dataset compared to DNME and TNME, suggesting stronger expressive capacity. However, due to its reliance on IVP solvers, its inference speed is relatively slow.

We report the final performance of the above experiments on the test set in Table 3 and Table 4.

It can be observed that, apart from certain anomalies exhibited by CMSM under the non-Markovian ablation on the STAIR dataset (as also reflected in Figure 4), the non-ablation settings consistently achieve lower $\text{CTF}_{\text{RMSE}}$. This further highlights the critical role of the three assumptions in Corollary 5.4, namely $\mathcal{A}_{\text{TM-SCM}}$, $\mathcal{A}_{\text{M}}$, and $\mathcal{A}_{\leq}$, in ensuring counterfactual consistency.

Finally, we conducted an ablation study on exogenous distributions by constructing new models using different exogenous distributions as described in Appendix C.2. This analysis demonstrates that Corollary 5.4 and the proposed neural TM-SCM are insensitive to the type of exogenous distribution. The results, shown in Table 5, indicate that while different models may exhibit some variation in the fit of the observational distribution (i.e., $\text{OBS}_{\text{WD}}$), their counterfactual consistency (i.e., $\text{CTF}_{\text{RMSE}}$) remains largely similar. Furthermore, as asserted in Appendix C.2, using a standard normal distribution as the exogenous distribution may not provide sufficient expressive power, as evidenced by the higher $\text{OBS}_{\text{WD}}$ values in DNME and TNME models in Table 5.

**Ablation on ER-DIAG-50 and ER-TRIL-50**   We report below the complete results of the ablation study on the ER-DIAG-50 and ER-TRIL-50 datasets as presented in the main text. The ER-DIAG-50 and ER-TRIL-50 datasets encompass diverse TM-SCM configurations, including various scales, structures, and parameter settings. Compared to the four symbolically constructed datasets in TM-SCM-SYM, they provide more comprehensive validation and testing for our framework.

The complete version of the ablation results table from the main text, as shown in Table 6, presents the final results on the test set. The table additionally includes the metrics $\text{OBS}_{\text{WD}}$ and $\text{CTF}_{\text{WD}}$, which measure the average fit to the observational distribution and the prediction for the interventional distribution, respectively.

The results demonstrate that models without ablations achieve the best performance on the final counterfactual consistency

*Table 6.* Ablation results of neural TM-SCM on ER-Diag-50 and ER-Tril-50. The values shown are the means of 50 experiments, with the subscript representing the 95% CI. The best-performing results are highlighted in bold.

| METHOD | | ER-Diag-50 | | | ER-Tril-50 | | |
|---|---|---|---|---|---|---|---|
| | | $\text{OBS}_\text{WD}$ | $\text{CTF}_\text{RMSE}$ | $\text{CTF}_\text{WD}$ | $\text{OBS}_\text{WD}$ | $\text{CTF}_\text{RMSE}$ | $\text{CTF}_\text{WD}$ |
| DNME | - | $\mathbf{3.47}_{\pm\mathbf{0.53}}$ | $\mathbf{0.53}_{\pm\mathbf{0.05}}$ | $\mathbf{1.50}_{\pm\mathbf{0.28}}$ | $21.28_{\pm22.09}$ | $\mathbf{0.51}_{\pm\mathbf{0.12}}$ | $\mathbf{3.37}_{\pm\mathbf{3.05}}$ |
| | w/o O | $5.06_{\pm1.75}$ | $0.78_{\pm0.05}$ | $2.88_{\pm0.43}$ | $25.42_{\pm21.27}$ | $0.89_{\pm0.10}$ | $4.79_{\pm3.03}$ |
| | w/o M | $4.35_{\pm0.77}$ | $0.62_{\pm0.04}$ | $1.98_{\pm0.30}$ | $\mathbf{18.53}_{\pm\mathbf{24.04}}$ | $0.58_{\pm0.10}$ | $3.60_{\pm3.04}$ |
| TNME | - | $4.65_{\pm2.93}$ | $\mathbf{0.47}_{\pm\mathbf{0.05}}$ | $\mathbf{1.32}_{\pm\mathbf{0.29}}$ | $\mathbf{3.34}_{\pm\mathbf{3.17}}$ | $\mathbf{0.55}_{\pm\mathbf{0.12}}$ | $\mathbf{3.49}_{\pm\mathbf{3.05}}$ |
| | w/o O | $5.40_{\pm2.55}$ | $11.24_{\pm20.98}$ | $3.47_{\pm1.08}$ | $23.75_{\pm29.53}$ | $6.41_{\pm9.84}$ | $37.11_{\pm45.91}$ |
| | w/o M | $\mathbf{3.88}_{\pm\mathbf{0.65}}$ | $0.62_{\pm0.04}$ | $2.08_{\pm0.33}$ | $5.57_{\pm3.27}$ | $0.73_{\pm0.21}$ | $13.60_{\pm19.84}$ |
| CMSM | - | $15.70_{\pm23.20}$ | $\mathbf{0.37}_{\pm\mathbf{0.05}}$ | $\mathbf{0.96}_{\pm\mathbf{0.23}}$ | $9.78_{\pm9.87}$ | $\mathbf{0.42}_{\pm\mathbf{0.12}}$ | $3.17_{\pm3.07}$ |
| | w/o O | $41.74_{\pm45.13}$ | $2.64_{\pm3.72}$ | $2.91_{\pm0.44}$ | $36.65_{\pm30.28}$ | $2.12_{\pm2.49}$ | $4.69_{\pm3.08}$ |
| | w/o M | $8.00_{\pm9.88}$ | $1.69_{\pm2.60}$ | $1.20_{\pm0.35}$ | $10.39_{\pm10.56}$ | $0.75_{\pm0.49}$ | $7.79_{\pm8.34}$ |
| | w/o T | $\mathbf{3.74}_{\pm\mathbf{1.09}}$ | $0.64_{\pm0.05}$ | $2.47_{\pm0.51}$ | $23.36_{\pm36.32}$ | $1.25_{\pm1.29}$ | $\mathbf{2.89}_{\pm\mathbf{1.33}}$ |
| TVSM | - | $3.66_{\pm1.05}$ | $\mathbf{0.46}_{\pm\mathbf{0.05}}$ | $\mathbf{1.29}_{\pm\mathbf{0.28}}$ | $3.40_{\pm2.01}$ | $\mathbf{0.50}_{\pm\mathbf{0.12}}$ | $3.35_{\pm2.99}$ |
| | w/o O | $5.25_{\pm2.31}$ | $0.79_{\pm0.04}$ | $2.89_{\pm0.43}$ | $11.79_{\pm10.36}$ | $0.88_{\pm0.10}$ | $4.68_{\pm3.03}$ |
| | w/o M | $\mathbf{3.54}_{\pm\mathbf{0.53}}$ | $0.53_{\pm0.05}$ | $1.63_{\pm0.29}$ | $4.25_{\pm4.82}$ | $0.53_{\pm0.11}$ | $\mathbf{3.33}_{\pm\mathbf{3.00}}$ |
| | w/o T | $22.22_{\pm36.74}$ | $0.67_{\pm0.05}$ | $2.43_{\pm0.49}$ | $14.30_{\pm15.34}$ | $0.78_{\pm0.12}$ | $6.63_{\pm3.87}$ |

metric, $\text{CTF}_\text{RMSE}$. This highlights the universality of the three assumptions in Corollary 5.4, namely $\mathcal{A}_\text{TM-SCM}$, $\mathcal{A}_\text{M}$, and $\mathcal{A}_\leq$, in ensuring counterfactual consistency. Furthermore, these findings empirically reinforce the validity of Corollary 5.4.

Beyond providing empirical support for counterfactual consistency, these results also reveal several characteristics of the four different models. For instance, DNME performs significantly better on ER-Diag-50 compared to ER-Tril-50, as DNME is specifically designed based on causal mechanisms where exogenous variables interact in a diagonal manner. The performance drop on ER-Tril-50 indicates that DNME lacks sufficient expressive power in more general settings. TNME addresses this limitation of DNME through its design. CMSM may exhibit instability due to its potentially poor fit to the observed distribution. Furthermore, as only numerically stable test results are recorded in the experiments, many omitted cases in CMSM's results may contribute to its lower $\text{CTF}_\text{WD}$ values. In contrast, TVSM demonstrates higher accuracy in counterfactual inference compared to DNME and TNME, while also achieving a better fit to the observational distribution than CMSM.

