# OpenReview forum: "Exogenous Isomorphism for Counterfactual Identifiability"
_ICML.cc/2025/Conference — ICML 2025 spotlightposter_

### Official Review · Reviewer_CJgk · 2025-03-10

**Overall Recommendation:** 4

**Summary:**

This paper analyses L3 identifiability, showing that full recovery of exogenous variables in SCMs is not required to achieve it. It also unifies the existing theories of bijective SCMs and TM-SCMs, implementing neural TM-SCM for the experimental section, thus showing the practical applicability of the developed ideas.

## Update after rebuttal
Authors carefully addressed my concerns about novelty. I believe this work is sound and rigurous and, in light of the arguments in the response, not only makes valuable connections among existing theories but also extends them. I am now more confident in changing my initial score from 3: Weak accept to 4: Accept.

**Claims And Evidence:**

The claims are well supported.

**Essential References Not Discussed:**

All relevant related papers are properly cited.

**Experimental Designs Or Analyses:**

I did not see any important issues regarding experimental designs.

**Methods And Evaluation Criteria:**

Benchmarks and datasets are quite complete. No external evaluation is performed, but this is not required due to the nature of the paper.

**Other Comments Or Suggestions:**

-

**Other Strengths And Weaknesses:**

As a strength, the paper is generally well written.

As weakness, the paper does not seem to present great novelties. The fact that full recovery of exogenous variables in SCMs is not required to achieve counterfactual identifiability has been previously demonstrated in works like (Nasr-Esfahany et al., 2023) or (Javaloy et al., 2023). Neural TM-SCM are not new either. This paper has some value as it collects the state-of-the-art ideas about counterfactual identifiability and makes and effort in unifying the ideas of bijective SCMs and of TM-SCM, but I am not completely sure if it has an important enough novelty for ICML. I would appreciate if authors could address this issue.

**Questions For Authors:**

Could the authors specify the degree of novelty of the paper? Is it a compilation and formalization of state-of-the-art ideas or does it bring some novel idea?

**Relation To Broader Scientific Literature:**

All relevant related papers are properly cited.

**Theoretical Claims:**

Theoretical claims are well supported.

---

> ### Author Rebuttal · Authors · 2025-03-31
>
> We appreciate the reviewers for raising questions regarding the novelty of this paper. Below are clarifications to address these misunderstandings:
>
> > **Question 1**: Specify the degree of novelty of the paper
>
> The main result of this paper is that if causal mechanisms are aligned via **exogenous isomorphism**, then they yield **consistent $\mathcal{L}_3$ distribution** (Theorem 3.2). The former serves as a target in representation learning, while the latter is formulated by PCH. In this way, our primary contribution resides in establishing a novel connection between these two distinct perspectives.
>
> Theorem 3.2 applies to **any recursive SCM**. BSCM and TM-SCM then introduce additional parametric assumptions, and they are special cases identified in our effort to relate our work to previous literature.
>
> From the perspective of BSCM or TM-SCM, our paper further extends existing counterfactual identification theories (e.g., Nasr-Esfahany et al., 2023) by:
> 1. Addressing not only counterfactual outcomes but all $\mathcal{L}_3$ distributions defined by PCH;
> 2. Extending the framework to vector spaces, where each causal variable take values in $\mathbb{R}^d$;
> 3. Presenting our derivations from a novel perspective of counterfactual transport.
>
> Regarding model design works (e.g., Javaloy et al., 2023), our contribution is to help formalize proofs that model indeed possess counterfactual identifiability (previously only demonstrated to achieve component-wise bijective identifiability in representation learning). This aspect has been rarely discussed or is only empirically illustrated in those studies.
>
> These theoretical and practical connections are further detailed in the related work section in Appendix B (see page 25). In the revised version, we plan to add a dedicated related work section in the main text to briefly highlight the novelty of our contributions.
>
> > **Question 2**: Formalization of state-of-the-art ideas or bringing in some novel idea?
>
> As suggested by the title, the novel perspective of exogenous isomorphism is the primary contribution of our paper. The reviewer’s concerns might arise for the following reasons:
> - Experimentally and in terms of conclusions, our work may appear similar to previous studies—partly because we discuss extensive material on BSCM and TM-SCM. However, these are used as bridges to connect the idea of exogenous isomorphism with prior work and to design ablation experiments that validate our theoretical findings. Hence, referring to their results is necessary.
> - In addition, the imprecise wording in the abstract may have led readers to mistakenly believe that proving that full recovery of exogenous variables in SCMs is not required to achieve counterfactual identifiability is our primary contribution. In fact, our focus is on clarifying the degree of representational identifiability that leads to complete counterfactual identifiability. We will update the relevant text accordingly.
>
> We appreciate the reviewer’s valuable feedback, which will help us enhance the clarity and impact of our revised manuscript.

---

> > ### Comment · Reviewer_CJgk · 2025-04-03
> >
> > I would like to thank the authors for carefully addressing my concerns about novelty. I think that this work is sound and rigorous and, by the arguments in the response, it seems that it not only makes valuable connections among existing theories but also extends those theories. I cannot fully evaluate the extent to which the paper presents novelty, so I defer this aspect to other reviewers, but I maintain my inclination towards accept.

---

### Official Review · Reviewer_PYFF · 2025-03-14

**Overall Recommendation:** 4

**Summary:**

This paper explores ∼L3-identifiability, aiming to ensure that all Structural Causal Models (SCMs) satisfying given assumptions provide consistent answers to causal questions. The authors introduce exogenous isomorphism and propose ∼EI-identifiability, showing that full recovery of exogenous variables is not needed for ∼L3-consistency. They explore this in two special SCM classes: Bijective SCMs (BSCMs) and Triangular Monotonic SCMs (TM-SCMs), offering new methods to achieve identifiability. The paper also leverages neural TM-SCMs for counterfactual reasoning, with experiments validating the effectiveness of the proposed approach. Key contributions include the novel concept of exogenous isomorphism and the unified framework for achieving counterfactual identifiability.

**Claims And Evidence:**

Yes, the claims made in the paper are well-supported by theoretical proofs and empirical evidence:
Theoretical Proofs: The authors provide rigorous mathematical proofs for their key claims, such as Theorem 3.2 (which shows that exogenous isomorphism implies ∼L3-identifiability) and Theorem 4.6 (which demonstrates ∼EI-identifiability for Bijective SCMs). These proofs are detailed and logically sound, establishing a strong theoretical foundation.2.
Empirical Validation: The paper includes experiments on synthetic datasets that demonstrate the effectiveness of the proposed methods. The results show that neural TM-SCMs achieve higher counterfactual consistency (measured by CTFRMSE) compared to ablations, validating the necessity of the assumptions made for ∼EI-identifiability.

**Essential References Not Discussed:**

No, there are no essential related works that are missing from the citations or discussions in the paper.

**Experimental Designs Or Analyses:**

Yes, I checked the soundness of the experimental designs and analyses. The authors conducted experiments on synthetic datasets to validate the correctness of the theory and the effectiveness of the proposed method. The experimental design considered different model structures (such as DNME, TNME, CMSM, and TVSM) and various datasets (such as TM-SCM-SYM, ER-DIAG-50, and ER-TRIL-50), which are representative to some extent. In the experimental analysis, the authors provided a detailed discussion on the impact of each assumption on model performance and used ablation studies to verify the necessity of these assumptions.

**Methods And Evaluation Criteria:**

Yes, the methods and evaluation criteria used in the paper are appropriate for the problem of counterfactual reasoning. The authors propose a novel approach to achieve∼EI-identifiability by focusing on specific classes of SCMs(BSCMs and TM-SCMs)and demonstrate its effectiveness through neural network parameterizations. The evaluation metrics, including OBSWD(Wasserstein distance for observational distribution)and CTFRMSE(root mean square error for counterfactual outcomes),are well-suited to assess the model's performance in fitting the observational data and generating consistent counterfactual results.

**Other Comments Or Suggestions:**

1. Scalability Analysis: The authors should provide a detailed analysis of the computational complexity and scalability of the proposed neural TM-SCMs, especially for large-scale datasets. This would help address potential limitations in practical applications.
2. Comparison with Other Methods: A more detailed comparison with other state-of-the-art methods for counterfactual reasoning would strengthen the paper. This could include both theoretical and empirical comparisons, highlighting the advantages and trade-offs of the proposed approach.

**Other Strengths And Weaknesses:**

Strengths:
1. Theoretical Innovation: The paper introduces the concept of exogenous isomorphism and proposes ∼EI-identifiability based on this, providing new theoretical support for the problem of causal identifiability.
2. Methodological Practicality: The paper not only presents theoretical insights but also applies the theory to practical problems through neural TM-SCMs, demonstrating its effectiveness in counterfactual reasoning.
3. Experimental Validation: The paper validates the correctness of the theory and the effectiveness of the method through experiments on synthetic datasets, offering strong support for the reliability of both the theory and the approach.
Weaknesses:
1. Limitations of Assumptions: Although the assumptions proposed in the paper are theoretically sound, they may be difficult to satisfy in practical applications. For example, the assumption that the causal model is Markovian may not hold in some complex scenarios.
2. Computational Complexity: The neural TM-SCMs proposed in the paper may have high computational complexity, especially when dealing with large-scale datasets. This could limit their scalability in practical applications.

**Questions For Authors:**

1. Real-World Applications: Can the authors provide examples of real-world applications where the proposed method could be particularly beneficial? How would the method handle the complexities and noise typically found in real-world data?
2. Comparison with Other Methods: Could the authors provide a more detailed comparison with other causal representation learning methods, especially those that also aim to improve counterfactual consistency? What are the key differences and advantages of the proposed approach?

**Relation To Broader Scientific Literature:**

The paper situates its contributions within the broader literature on causal inference and machine learning,:
In the field of causal inference, counterfactual identifiability is an important research direction. This paper proposes a new identifiability target—achieving identifiability across the entire counterfactual hierarchy—which offers new ideas and methods for research in causal inference. Additionally, the concept of exogenous isomorphism introduced in the paper and the ∼EI-identifiability based on it provide a new perspective for the theoretical study of causal models.
In the field of machine learning, causal inference and causal representation learning are current research hotspots. The research findings of this paper not only provide theoretical support for research in these areas but also offer valuable references for practical applications.

**Theoretical Claims:**

Yes, I checked the correctness of the proofs. The authors introduce several theorems and lemmas to establish the connection between exogenous isomorphism and∼L3-identifiability. For example, Theorem 3.2 shows that exogenous isomorphism implies∼L3-identifiability, providing a solid theoretical foundation for the proposed method. The proofs are detailed and logically sound, supporting the claims made in the paper.

---

> ### Author Rebuttal · Authors · 2025-03-31
>
> We thank the reviewer for their careful reading and for pointing out some weaknesses and suggestions. Below are our responses to these points.
>
> > **Weakness 1**: Limitations of Assumptions
>
> It is acknowledged that the strength of the assumptions is a common drawback in identification tasks; however, these assumptions are necessary to guarantee identifiability. For instance, the Markovianity assumption is routinely employed in causal inference, causal discovery, and causal representation learning, especially in situations where only observational data is available. This is also the primary scenario studied in our work with BSCM and TM-SCM.
>
> Although the assumptions in our current results for BSCM and TM-SCM are relatively strong, we remain optimistic because the theory regarding exogenous isomorphism applies to **any recursive SCM** without imposing additional assumptions about distributions or mechanisms. This lays the groundwork for future analysis of $\mathcal{L}_3$-identifiability in imperfect real-world scenarios.
>
> > **Weakness 2 & Suggestion 1**: Complexity When Dealing with Large-Scale Datasets
>
> In terms of model construction, TM-SCM represents a class of models that includes additive noise models, causal flows, and others (see Appendix B.2, page 25). These models are closely related to current deep generative models, which provides potential for future applications on high-dimensional and large-scale data. Since our paper mainly focuses on proof-of-concept experiments aimed at empirically validating the correctness of the related theory, we have not yet explored applications on large-scale datasets.
>
> Regarding inference, especially for counterfactual inference with invertible SCMs, we present the Pseudo Potential Response algorithm in detail in Appendix C.7 (see page 33). This algorithm requires $n$ iterations, where $n$ is the total dimensionality of the variables, with each iteration involving one inverse operation and one forward operation. Although the algorithm iterates over the dimensions, the algorithm extends the approach in (Javaloy et al., 2023) to handle vector spaces and multiple interventions. We also note that (Le et al., 2024) and (Almodóvar et al., 2025) have proposed approximate methods that learn a encoder fitting the inverse operation, which can accelerate counterfactual inference on invertible SCMs. We will clarify and expand on these points in the appendix.
>
> > **Question 1**: Real-World Applications
>
> Counterfactual reasoning has already been widely applied in real-world tasks (see lines 38-42). This work focuses on the identifiability of counterfactual reasoning, and its impact lies in theoretically ensuring the reliability of related applications.
>
> At present, this work remains in the early stages of theoretical development and synthetic data experiments. We primarily use proof-of-concept experiments to demonstrate the practicality of addressing counterfactual consistency issues. We have not yet ventured into real-world applications, and therefore have not focused on the complexities and noise involved therein. Nonetheless, the results from these synthetic experiments offer preliminary empirical support for the theoretical guarantees of counterfactual identifiability established in several related works.
>
> > **Suggestion 2 & Question 2**: Comparison with Other Methods
>
> The theoretical comparisons have already been listed in the related work section in Appendix B (see page 25), and we will also include the literature mentioned by reviewer BQ84.
>
> In practice, since we have not proposed a specific model, a direct comparison with other methods is not necessary. Instead, we have refined four prototypes from previous literature (simplified implementations of several state-of-the-art methods) to validate the theory in the ablation experiments. These prototypes were built and run in all ablation experiments, so the ablation studies inherently serve as a comparison among different methods, as detailed in Appendix D.4 (see page 37).
>
> In the revised version, we will include a brief version of the related work in the main text to provide a more detailed comparison, which will highlight the advantages.

---

> > ### Comment · Reviewer_PYFF · 2025-04-08
> >
> > Thank you for your response. My concerns have been addressed. This is a solid paper as a great contribution to causality community. I am willing to increase my score to 4.

---

### Official Review · Reviewer_BQ84 · 2025-03-17

**Overall Recommendation:** 4

**Summary:**

The paper introduces the notation of exogenenous isomorphism (EI) between SCMs, a suffient relationship (Theorem 3.2) that ensuring counterfactual equivlance. In section 4 and section 5, the paper focus on two special types of SCMs: Bijective SCMs (BSCMs) and Triangular Monotonic SCMs (TM-SCMs). The paper first introduces the sufficient and necessary condition of achieving EI identification is that there exists a bijection $h_i$ between the exogenous variable $\textbf{u}\_i^{(1)}$ and  $\textbf{u}\_i^{(2)}$ of two SCMs $\mathcal{M}^{(1)}$ and $\mathcal{M}^{(2)}$ (Theorem 4.3). Further, the paper proves EI identifiability under assumption set $\mathbf{\mathcal{A}}$ from counterfactual transport (Theorem 4.6) and KR Transport (Theorem 4.8). Similar result applies to Triangular Monotonic SCM (Proposition 5.3).

Finally, the paper shows how these theoretical results can inform practical modeling strategies. By designing “neural” versions of TM-SCMs—where monotonic and triangular structures are imposed on neural networks—the authors illustrate how to fit a model solely to observational data while still supporting consistent answers for counterfactual queries. Empirical evaluations on synthetic data demonstrate that the learned models reproduce ground-truth counterfactual outcomes accurately, underlining the paper’s main claim that consistent counterfactual inference can be achieved without recovering the full latent structure.

**Claims And Evidence:**

The paper claims EI identification results on Bijection SCMs and Triangular Monotonic SCMs supported by various theorems. The paper further proposes Neural TM-SCM, constructing the neural network parameterized SCMs to satisfy the assumptions the theorem requires.

**Essential References Not Discussed:**

Here are some paper talking about bijective or invertible SCM in the literature, and similar (partial) results in latent causal model counterfactual identification or causal representation learning. For example, some of the work here talks about TR transformation on identification (upto point-wise bijection and a re-ordering operator due to the nature of latent causal model) [3,4,5].

[1] Nasr-Esfahany, A., & Kiciman, E. (2023). Counterfactual (non-) identifiability of learned structural causal models. arXiv preprint arXiv:2301.09031.

[2] Nasr-Esfahany, A., Alizadeh, M., & Shah, D. (2023, July). Counterfactual identifiability of bijective causal models. In International Conference on Machine Learning (pp. 25733-25754). PMLR.

[3] Brehmer, J., De Haan, P., Lippe, P., & Cohen, T. S. (2022). Weakly supervised causal representation learning. Advances in Neural Information Processing Systems, 35, 38319-38331.

[4] Wu, P., Li, H., Zheng, C., Zeng, Y., Chen, J., Liu, Y., ... & Zhang, K. (2025). Learning Counterfactual Outcomes Under Rank Preservation. arXiv preprint arXiv:2502.06398.

[5] Zhou, Z., Bai, R., Kulinski, S., Kocaoglu, M., & Inouye, D. I. (2023). Towards characterizing domain counterfactuals for invertible latent causal models. arXiv preprint arXiv:2306.11281.

**Experimental Designs Or Analyses:**

The experiments reflects the theorem, showing the constructrion constraints attemping to satisfy $\mathbf{\mathcal{A}}$ achieves lower coutnerfactual error (CTF RMSE).

**Methods And Evaluation Criteria:**

The methods and evaluation criteria reflects the problem.

**Other Comments Or Suggestions:**

Typos: Title of Seciton 5, should be triangular.

**Other Strengths And Weaknesses:**

The paper writes clearly and well-structured. Though notation heavy, it is understandable since the paper contains heavy theorem contributions.

**Questions For Authors:**

1. You show that two SCMs are exogenously isomorphic if their causal mechanisms align via component-wise bijections and they induce the same L3 distributions. In real-world scenarios (where full knowledge of mechanisms is rarely available), do you envision practical approaches to test whether two learned SCMs might be exogenously isomorphic or close to it?

**Relation To Broader Scientific Literature:**

Thie paper extends and generalize the existed counterfacutal identification results. Like it extends [1]'s theroem and generalize [2] (bijective, proposition 6.2) )and [3] (bijective, markovian, Guassian exogenous noise in Theorem 1) to a more generalize scenario.

**Theoretical Claims:**

I did not check the correctness of proofs.

---

> ### Author Rebuttal · Authors · 2025-03-31
>
> We thank the reviewer for pointing out the typo and have added three references relevant to our work:
>
> - (Brehmer et al., 2022): In this paper, Theorem 1 proves that the latent causal model is identifiable from weak supervision up to graph isomorphisms and elementwise diffeomorphisms, where the latter is a prerequisite for exogenous isomorphism. Our results further show that if such a model is $\sim_{\text{EI}}$-identifiable, then the counterfactual distribution is consistent. This answers the question raised in the paper regarding whether a redefinition of causal variables preserves the counterfactual distribution.
>
> - (Wu et al., 2025): This paper focuses on relaxing the strictly monotonicity assumption to a rank preservation assumption to prove the identifiability of counterfactual outcomes. The corresponding SCM can be regarded as a generalization of the TM-SCM, so the results of this paper indeed fall within the discussion framework of $\sim_{\text{EI}}$-identifiability.
>
> - (Zhou et al., 2023): This paper studies the invertible latent causal model, which is a version of the BSCM with latent causal variables, and is therefore related to the subject of our study. The paper mainly focuses on what is referred to as the domain counterfactual, defines the concept of domain counterfactual equivalence, and concentrates on deriving the counterfactual error bound.
>
> We will refine these points in our revision.
>
> > **Question 1**: In real-world scenarios (where full knowledge of mechanisms is rarely available), do you envision practical approaches to test whether two learned SCMs might be exogenously isomorphic or close to it?
>
> If we have full knowledge of the mechanisms (for example, if we have learned proxy SCMs and have access to their mechanisms, as discussed in (Zhou et al., 2023)), we can directly verify whether two SCMs are exogenously isomorphic according to the definition. However, in real-world scenarios, SCMs learned under incomplete knowledge are unlikely to happen to fall into the equivalence class of exogenous isomorphism, which corresponds to the identification task.
>
> In identification tasks, rather than using statistical tests, we typically introduce assumptions to prove identifiability; this is a means to ensure favorable properties a priori, with the connection to reality mainly lying in the strength of these assumptions. Our paper later discusses sets of assumptions that guarantee exogenous isomorphism for two specific types of SCMs: BSCM and TM-SCM.
>
> Of course, there are also statistical methods that can indicate whether a learned SCM is close to being exogenously isomorphic, such as the counterfactual error bound presented in (Zhou et al., 2023) or the partially identifiable aspects that have recently received attention in counterfactual reasoning. However, this is somewhat beyond the scope of our discussion and can be explored in future work.

---

> > ### Comment · Reviewer_BQ84 · 2025-04-05
> >
> > Thanks for answering my questions! It's a great paper. I will keep my score.

---

### Decision · Program_Chairs · 2025-05-01

**Decision:**

Accept (spotlight poster)

**Comment:**

All reviews are unanimous about this paper. Authors did clarify aspects relating to novelty and comparisons to existing works during the discussion which seems to have satisfied the reviewers.

The paper proposes exogenous isomorphism - an equivalence relation between Structural Causal Models (SCMs) such that for all SCMs that are equivalent under this relation, all the counterfactual queries are answered uniquely. This identifies all queries in the three layers of Pearl's Causal Hierarchy Uniquely.  Authors then argue that if this is satisfied, it is not necessary to estimate the true exogenous variables. Authors also discuss sub-classes of SCMs where such an equivalence relation holds and also propose Neural architectures that can utilize the theory to answer counterfactual queries.

The paper addresses a fundamental problem and makes counterfactual identification possible for a class of SCMs satisfying some equivalence relation. Solid contribution. I recommend acceptance.